# Enhancing LLM Training via Spectral Clipping

Xiaowen Jiang [1 2]   Andrei Semenov [3]   Sebastian U. Stich [1]

## Abstract

While spectral-based optimizers like Muon operate directly on the spectrum of updates, standard adaptive methods such as AdamW do not account for the spectral structure of weights and gradients, leaving them vulnerable to two empirical issues in large language model (LLM) training: (i) the optimizer updates can have large spectral norms, potentially destabilizing training and degrading generalization; (ii) stochastic gradient noise can exhibit sparse spectral spikes, with a few dominant singular values much larger than the rest. We propose *SPECTRA*, a general framework addressing these by (i) *post*-spectral clipping of updates to enforce spectral-norm constraints (ii) optional *pre*-spectral clipping of gradients to suppress spectral noise spikes. We prove that post-clipping constitutes a Composite Frank-Wolfe method with spectral-norm constraints and weight regularization. We further analyze how pre-clipping mitigates sparse spectral spikes. We propose efficient soft spectral clipping via Newton-Schulz iterations, avoiding expensive SVD. Experiments on LLM pretraining show SPECTRA uniformly improves validation loss for various optimizers, including AdamW, Signum, Mars, and AdEMAMix, with the best-performing variants achieving state-of-the-art results. Models trained with SPECTRA exhibit smaller weight norms, confirming the link between spectral clipping and regularization.

## 1. Introduction

The optimization of Large Language Models (LLMs) has increasingly shifted focus from simple gradient-based updates to more advanced preconditioning and normalization techniques. While standard optimizers like AdamW (Loshchilov & Hutter, 2019) rely on coordinate-wise normalization to handle ill-conditioned curvature, a growing body of work suggests that controlling the spectral properties of weight matrices and updates can be beneficial for training stability and generalization (Miyato et al., 2018; Bartlett et al., 2017; Pethick et al., 2025; Davis & Drusvyatskiy, 2025). Consequently, spectral-based optimizers such as Shampoo (Gupta et al., 2018), Spectral Descent (Carlson et al., 2015b) and Muon (Jordan et al., 2024; Liu et al., 2025)—which operate directly on the spectrum of update matrices—have emerged as theoretically grounded paths toward faster convergence.

However, the transition to purely spectral optimizers is not yet ubiquitous. Recent optimizer benchmarks reveal that advanced coordinate-wise methods, such as AdEMAMix (Pagliardini et al., 2025) and Mars (Yuan et al., 2025), often match or exceed the performance of current spectral approaches (Semenov et al., 2025). Despite their effectiveness, these coordinate-wise algorithms do not account for the global spectral structure of weights and gradients, leaving them vulnerable to two fundamental issues.

First, the spectral norms of optimizer updates are often uncontrolled. As detailed in Section 2, popular optimizers like Signum (Bernstein et al., 2018) or AdamW (Loshchilov & Hutter, 2019) can produce update directions with excessive spectral norms, potentially destabilizing training and degrading generalization. Second, stochastic gradient noise frequently exhibits "sparse spectral spikes", by which we mean low-rank components with singular values orders of magnitude larger than the signal. Standard coordinate-wise or global clipping fails to distinguish these artifacts, often suppressing the informative signal alongside the noise.

**Contributions.** To bridge the gap between the empirical strength of coordinate-wise methods and the theoretical benefits of spectral control, we introduce *SPECTRA* (SPEctral Clipping for TRaining Acceleration). SPECTRA is a general, optimizer-agnostic framework that: (i) applies *post-spectral clipping* to the update matrix to strictly enforce spectral constraints, and (ii) optionally applies *pre-spectral clipping* to raw gradients to filter spectral noise spikes. Our main contributions are as follows:

**1) Algorithmic Framework.** We propose SPECTRA, a flexible wrapper applicable to any base optimizer with de-

[1]CISPA Helmholtz Center for Information Security, Saarbrücken, Germany [2]Universität des Saarlandes, Saarbrücken, Germany [3]EPFL. Correspondence to: Xiaowen Jiang <xiaowen.jiang@cispa.de>, Andrei Semenov <andrii.semenov@epfl.ch>, Sebastian U. Stich <stich@cispa.de>.

*Proceedings of the 43rd International Conference on Machine Learning*, Seoul, South Korea. PMLR 306, 2026. Copyright 2026 by the author(s).

coupled weight decay (e.g., AdamW, Signum, Mars, AdE-MAMix). We develop a GPU-efficient approximation of spectral clipping via Newton-Schulz iterations based solely on fast square matrix–matrix multiplications.

**2) Implicit Regularization & Spectral Norm Constraints.** We prove that applying post-spectral clipping to the standard update rule with decoupled weight decay is equivalent to a Composite Frank-Wolfe algorithm. This reformulation reveals that SPECTRA minimizes a spectral-norm constrained objective with implicit weight regularization, simultaneously enhancing stability and promoting low-complexity solutions. Furthermore, we provide an explicit geometric interpretation of the hyperparameters, showing how they directly control the radius of the spectral constraint ball and the strength of the regularization.

**3) Robustness to Spectral Spikes.** We empirically demonstrate the existence of spectral noise spikes in stochastic gradients during LLM training. We then provide theoretical insights on why spectral clipping can help mitigate the adverse effects of such noise structures during optimization and can be significantly better than global norm clipping.

**4) Empirical SOTA.** We evaluate SPECTRA on both synthetic and LLM pretraining tasks ranging from 124M to 1.5B parameters. We demonstrate that SPECTRA consistently improves the validation loss of base optimizers and enables the use of larger learning rates. Applying SPECTRA to coordinate-wise methods achieves the best performance among state-of-the-art methods. Additionally, models trained with SPECTRA exhibit smaller weight norms, confirming the theoretical link to weight regularization. The code is available at https://github.com/mlolab/llm-spectral-clipping.

**Notations.** We denote the set of integers $\{1, \ldots, n\}$ by $[n]$. For a matrix $\mathbf{X}$, we let $\sigma_i(\mathbf{X})$ be its $i$-th largest singular value. We use $\|\mathbf{X}\|_2$ and $\|\mathbf{X}\|_F$ to denote the spectral and Frobenius norms of $\mathbf{X}$, respectively, and $\|\mathbf{x}\|_2$ for the Euclidean norm of a vector $\mathbf{x}$. For both matrices and vectors, we use $\|\mathbf{X}\|_\infty = \max_{i,j}\{|\mathbf{X}_{i,j}|\}$ to denote the $\ell_\infty$ norm. Other notations are discussed as needed within the context.

We discuss the closely related work in the main text and defer the reader to Appendix A for additional references and broader context.

## 2. Preliminaries and Setup

The architecture of transformer-based LLMs consists of a large number of linear transformations, parameterized by weight matrices such as the query, key, value, and output projections in attention modules, as well as matrices in feed-forward layers. Without loss of generality, we focus on the training of a single such weight matrix, denoted by

$\mathbf{X} \in \mathbb{R}^{m \times n}$ and initialized at $\mathbf{X}_0$, using stochastic gradient methods. The widely-used update rule with decoupled weight decay is given by:

$$\mathbf{X}_{k+1} = (1 - \lambda\eta_k)\mathbf{X}_k - \eta_k\mathbf{U}_k, \quad k = 0, 1, 2 \ldots, \quad (1)$$

where $\lambda \geq 0$ is the weight decay factor, $\eta_k > 0$ is the learning rate, and $\mathbf{U}_k$ denotes the update matrix computed by the specific optimizer using the stochastic gradient.

This formulation contains various popular optimizers used in LLM training, including AdamW (Loshchilov & Hutter, 2019), AdEMAMix (Pagliardini et al., 2025), Signum (Bernstein et al., 2018), Mars (Yuan et al., 2025) and Muon (Jordan et al., 2024; Liu et al., 2025). The distinction lies in the computation of the update matrix $\mathbf{U}_k$. For instance, SGD-M uses $\mathbf{U}_K = \mathbf{M}_k$ where $\mathbf{M}_k$ is a momentum buffer accumulated via Polyak momentum (Polyak, 1964), or Nesterov meomentum (Nesterov, 2018), or other recursive variants (Cutkosky & Orabona, 2019; Jiang et al., 2025). Signum (Sign-SGD with momentum) sets $\mathbf{U}_k = \text{sign}(\mathbf{M}_k)$ where sign is the entry-wise sign operator. For AdamW, the update is the momentum normalized by the square root of the second-moment estimates.

**Phenomenon I: Excessive Spectral Norm of Update Matrices.** Controlling the spectral norm of weight matrices is known to be crucial for both stable training (Miyato et al., 2018) and improvement of generalization (Bartlett et al., 2017), particularly for long runs (Pethick et al., 2025). Specifically, for any input vector with bounded norm, the Euclidean norm of the output after the linear transformation is controlled by the spectral norm of the weight matrix, effectively limiting the propagation of perturbations through the network. On the other hand, the weight norm is determined by the accumulated updates during training. Using update rule (1) with $\eta_k \equiv \eta$, we have for any $k \geq 1$:

$$\|\mathbf{X}_k\|_2 \leq (1-\lambda\eta)^k\|\mathbf{X}_0\|_2 + \frac{1 - (1 - \lambda\eta)^k}{\lambda} \max_{i<k}\{\|\mathbf{U}_i\|_2\} .$$

This exposes a vulnerability: if the spectral norms of the updates $\|\mathbf{U}_i\|_2$ are not controlled, the weight matrix can grow rapidly. It has been observed that standard adaptive optimizers like AdamW can produce updates with excessive magnitudes, particularly in the early phases of training (Liu et al., 2019) or prior to loss spikes (Wortsman et al., 2024). For sign-based methods (e.g., Signum), the spectral norm of the update $\mathbf{U}_k = \text{sign}(\mathbf{M}_k)$ is at least $\sqrt{\max(m,n)}$ and can be as large as $\sqrt{mn}$ (since for any $\mathbf{X} \in \mathbb{R}^{m \times n}$, we have $\frac{\|\text{sign}(\mathbf{X})\|_F}{\sqrt{\min(m,n)}} \leq \|\text{sign}(\mathbf{X})\|_2 \leq \|\text{sign}(\mathbf{X})\|_F$). Consequently, there exists risks of training instability and excessive final weight norms. Mitigating this risk often requires the use of small learning rates or extended warm-up periods to prevent divergence (You et al., 2019; Gilmer et al., 2022), which might slow down the convergence.

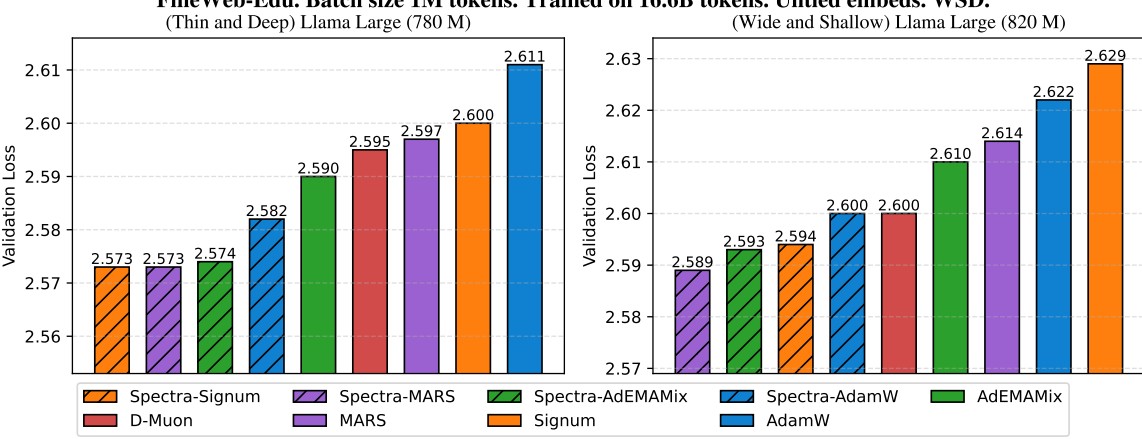

*Figure 1.* Final validation loss comparison for large Llama-style models trained for the Chinchilla optimal horizon. 'Thin and deep' models have smaller embedding dimensions but have more layers than 'wide and shallow' ones. The running time comparisons w/wo using SPECTRA are provided in Figure F.7. We use a total batchsize of 1012 and 992 for 780M and 820M model respectively with 1024 sequence length. The hyperparameters such as learning rate used for each method are reported in the tables in Section F.3.

**Phenomenon II: Sparse Spectral Spikes in Stochastic Gradients.** Recent studies identify gradient and loss spikes as a predominant source of training instability (Huang et al., 2025). In this work, we deepen this understanding by analyzing LLM training dynamics, revealing that the singular value spectrum of raw stochastic gradients is often heavy-tailed—specifically, a few singular values ("spectral outliers") are orders of magnitude larger than the rest. This aligns with findings characterizing Transformer gradient noise as highly non-Gaussian (Simsekli et al., 2019; Zhang et al., 2020). By decomposing the stochastic gradient into a signal component (approximated by large-batch gradients) and a residual noise component, we trace this phenomenon primarily to the noise. We observe that the noise often contains low-rank structures—*sparse spectral spikes*—whose magnitudes far exceed that of the signal. Furthermore, the subspace spanned by these spikes is typically nearly orthogonal to the principal signal directions, suggesting these high-magnitude components represent pure noise. Detailed empirical evidence is provided in Section 4.1.

**Remedy: Spectral Clipping Operator.** To mitigate the phenomena of excessive update norms and sparse noise spikes, we propose to apply a spectral clipping operator to the update matrices $\mathbf{U}_k$ (and optionally to the raw stochastic gradients). Formally, let $\mathbf{X} \in \mathbb{R}^{m \times n}$ be a fixed matrix with compact SVD decomposition $\mathbf{X} = \mathbf{U}\mathbf{S}\mathbf{V}^T$, where $\mathbf{U} \in \mathbb{R}^{m \times q}$, $\mathbf{S} \in \mathbb{R}^{q \times q}$ and $\mathbf{V} \in \mathbb{R}^{q \times n}$ with $q := \min(m, n)$. We define the spectral clipping operator as:

$$\text{clip}_c^{\text{sp}}(\mathbf{X}) := \mathbf{U} \, \text{diag}\big(\text{clip}_c(\mathbf{S}_{11}), \ldots, \text{clip}_c(\mathbf{S}_{qq})\big)\mathbf{V}^T \,,$$

where $c > 0$ is the clipping threshold, and $\text{clip}_c(x) := \text{sign}(x)\min(|x|, c)$ is the standard scalar clipping function.

By construction, this operator ensures $\|\text{clip}_c^{\text{sp}}(\mathbf{U})\|_2 \leq c$, which strictly controls the spectral norm of the updates. (The

update norm comparisons between AdamW and Spectra-AdamW are shown in Figure F.10 and F.12.) In the presence of sparse spike noise, spectral clipping can effectively remove those large noise spikes while preserving the scale and structure of the remaining signal components. (Rigorous arguments can be found in Section 4.2.)

Another two commonly used clipping operators are coordinate-wise clipping: $\text{clip}_c^{\text{cw}}(\mathbf{X})_{i,j} := \text{clip}_c(\mathbf{X}_{i,j})$ and global clipping: $\text{clip}_c^{\text{g}}(\mathbf{X}) := \min(1, c/\|\mathbf{X}\|_F)\mathbf{X}$. Coordinate-wise clipping ignores correlations between entries and fails to effectively limit the spectral norm, as discussed earlier. While global clipping can control the spectral norm, it achieves this by simultaneously suppressing the useful signal components along with the noise. We discuss the theoretical advantages of spectral clipping over global clipping in Section 4.2.

## 3. Post-Spectral Clipping: Spectral Norm Constrained Optimization with Implicit Weight Regularization

In this section, we formally introduce *Post-Spectral Clipping* and provide its theoretical justification. We show that adding this simple operation to the update can be mathematically equivalent to a Composite Frank-Wolfe algorithm solving a spectral-norm constrained optimization problem with weight norm regularization. This dual nature ensures that the optimizer simultaneously enforces strict spectral norm constraints while promoting low-complexity solutions (via the regularization), thereby potentially enhancing both training stability and generalization performance.

Specifically, we propose to use the following modified update rule. For $k = 0, 1, 2, \ldots$,

$$\mathbf{X}_{k+1} = (1 - \lambda\eta_k)\mathbf{X}_k - \alpha\eta_k \, \text{clip}_{c_k}^{\text{sp}}(\mathbf{U}_k) \,, \qquad (2)$$

where $\alpha > 0$ is a scaling factor. Compared to (1), the effective update direction $\alpha\,\text{clip}^{\text{sp}}_{c_k}(\mathbf{U}_k)$ has spectral norm bounded by $\alpha c_k$.

We now discuss its connection to the composite Frank-Wolfe method (Harchaoui et al., 2015; Yurtsever et al., 2018). Consider the following general composite constrained optimization problem:

$$\min_{\mathbf{X}\in Q}\big\{F(\mathbf{X}) := f(\mathbf{X}) + \psi(\mathbf{X})\big\}\,, \qquad (3)$$

where $f : \mathbb{R}^{m\times n} \to \mathbb{R}$ is the loss function (e.g. cross-entropy loss), $\psi : \mathbb{R}^{m\times n} \to \mathbb{R}\cup\{+\infty\}$ is a proper closed convex function and $Q \subseteq \mathbb{R}^{m\times n}$ is a compact convex set. We now analyze the instantiation of (2) using the standard polyak momentum update:

$$\mathbf{X}_{k+1} = (1-\lambda\eta_k)\mathbf{X}_k - \alpha\eta_k\,\text{clip}^{\text{sp}}_{c_k}(\mathbf{M}_k)\,, \qquad (4)$$

where $\mathbf{M}_{k+1} = (1-\beta_k)\mathbf{M}_k + \beta_k\mathbf{g}(\mathbf{X}_{k+1})$, $\beta_k \in (0,1]$ is the momentum parameter, and $\mathbf{g}(\mathbf{X}_{k+1})$ is the stochastic gradient of $f$ at $\mathbf{X}_{k+1}$. In what follows, we denote the spectral norm ball with radius $D_2 > 0$ as:

$$Q_2 := \{\mathbf{X} \in \mathbb{R}^{m\times n} : \|\mathbf{X}\|_2 \le D_2\}\,.$$

**Proposition 3.1.** *The update rule* (4) *can be equivalently reformulated as the following stochastic composite Frank-Wolfe method:*

$$\begin{cases} \mathbf{V}_{k+1} \approx \underset{\mathbf{X}\in Q}{\arg\min}\,\{\langle \mathbf{M}_k, \mathbf{X}\rangle + \psi(\mathbf{X})\} \\ \mathbf{X}_{k+1} = (1-\gamma_k)\,\mathbf{X}_k + \gamma_k\,\mathbf{V}_{k+1} \end{cases}, \qquad (5)$$

*by choosing* $\gamma_k = \lambda\eta_k$, $c_k \equiv \frac{\lambda D_2}{\alpha}$ *and* $\psi(\mathbf{X}) = \frac{\lambda}{2\alpha}\|\mathbf{X}\|_F^2$, *provided that* $Q = Q_2$ *and each subproblem for computing* $\mathbf{V}_{k+1}$ *is solved exactly.*

From the equivalent formulation, we can immediately derive standard convergence guarantees for (4).

**Assumption 3.2.** At any point $\mathbf{X} \in \mathbb{R}^{m\times n}$, the stochastic gradient oracle returns an independent estimator $\mathbf{g}(\mathbf{X})$ such that $\mathbb{E}[\mathbf{g}(\mathbf{X})] = \nabla f(\mathbf{X})$ and $\mathbb{E}[\|\mathbf{g}(\mathbf{X}) - \nabla f(\mathbf{X})\|_F^2] \le \sigma^2$.

**Assumption 3.3.** There exists $L_F > 0$ such that for any $\mathbf{X}, \mathbf{Y} \in \mathbb{R}^{m\times n}$, $\|\nabla f(\mathbf{X}) - \nabla f(\mathbf{Y})\|_F \le L_F\|\mathbf{X} - \mathbf{Y}\|_F$.

**Theorem 3.4.** *Let SGD-M with spectral clipping* (4) *be applied to problem* (3) *with* $Q = Q_2$ *and* $\psi(\mathbf{X}) := \frac{b}{2}\|\mathbf{X}\|_F^2$ *under convexity of* $f$ *and Assumption 3.2. Suppose we use a batch size of $B$ for computing the stochastic gradient* $\mathbf{g}(\mathbf{X}_k)$. *By choosing* $c_k \equiv bD_2$, $\lambda\eta_k = \frac{2}{k+2}$, $\alpha = \frac{\lambda}{b}$, *and* $\beta_k \equiv 1$, *we have for any* $K \ge 1$:

$$\mathbb{E}[F(\mathbf{X}_K) - F^\star] \le \frac{2C_f}{K+1} + \frac{D_F\sigma}{\sqrt{B}}\,,$$

*where* $C_f := \sup_{\mathbf{X},\mathbf{S}\in Q_2,\gamma\in[0,1],\mathbf{Y}=\mathbf{X}+\gamma(\mathbf{S}-\mathbf{X})}\{\frac{2}{\gamma^2}(f(\mathbf{Y}) - f(\mathbf{X}) - \langle\nabla f(\mathbf{X}), \mathbf{Y} - \mathbf{X}\rangle)\}$ *and* $D_F := \sup_{\mathbf{X},\mathbf{Y}\in Q}\{\|\mathbf{X} - $

$\mathbf{Y}\|_F\}$. *If Assumption 3.3 additionaly holds, then by choosing* $\beta_k = \frac{1}{(k+1)^{2/3}}$ *and* $\mathbf{M}_0 = \mathbf{g}(\mathbf{X}_0)$, *we have for any* $K \ge 1$:

$$\mathbb{E}[F(\mathbf{X}_K) - F^\star] \le \frac{2C_f}{K+1} + \frac{12.4L_F D_F^2 + 6.2D_F\sigma/\sqrt{B}}{K^{1/3}}\,.$$

The curvature constant $C_f$ is the standard quantity in Frank-Wolfe analysis (Jaggi, 2013), which measures the maximum non-linearity within the set $Q$. In the presence of noise, the rate of $1/K^{1/3}$ matches the standard stochastic Frank-Wolfe with momentum on a single function (Mokhtari et al., 2020).

*Remark* 3.5. In Theorem 3.4, we first fix the problem constraints $(D_2, b)$ and then set the parameters $(c, \lambda, \alpha)$. In practice, one can choose $c$, $\alpha$ and $\lambda$ to determine the size of $D_2$ and the level of regularization. Then the resulting algorithm is implicitly solving problem (3) with $\psi(\mathbf{X}) := \frac{\lambda}{2\alpha}\|\mathbf{X}\|_F^2$ and $Q := \{\mathbf{X} : \|\mathbf{X}\|_2 \le \frac{\alpha c}{\lambda}\}$.

**Connection to Muon.** Muon (Jordan et al., 2024) applies the orthogonalization operator to the momentum by normalizing all the singular values to 1. Clearly, $\frac{1}{c}\text{clip}^{\text{sp}}_c(\mathbf{X}) = \text{orth}(\mathbf{X}) := \mathbf{U}_X\mathbf{V}_X^T$ for any $c \le \sigma_{\min}(\mathbf{X})$. Thus, if we let $\alpha \to \infty$ and choose $c = \frac{1}{\alpha}$, then method (4) recovers Muon (Jordan et al., 2024) as a special case, which solves the constrained problem without regularization ($b = 0$).

### 3.1. Extensions: A Family of Spectral Optimizers

Previously, we have focused on the quadratic regularizer $\psi(\mathbf{X}) = \frac{b}{2}\|\mathbf{X}\|_F^2$ which leads to a simple spectral clipping operation. However, our framework is flexible: Different choices of the regularizer yield different algorithmic update rules. This allows us to characterize a broad family of *spectral optimizers*. Consider the general subproblem:

$$\mathbf{X}^\star = \underset{\mathbf{X}\in Q_2}{\arg\min}\big\{\langle\mathbf{G}, \mathbf{X}\rangle + \psi(\mathbf{X})\big\}\,, \qquad (6)$$

**Rotational Invariant Regularizers.** Consider the case where $\psi$ is separable and acts solely on the singular values of $\mathbf{X}$. In this setting, the optimal update takes the form $\mathbf{X}^\star = -\mathbf{U}_G\mathbf{S}^\star\mathbf{V}_G^T$, where $\mathbf{U}_G$ and $\mathbf{V}_G$ are the left and right singular vectors of $\mathbf{G}$, and $\mathbf{S}^\star$ is a diagonal matrix with entries $\mathbf{S}_{ii}^\star = \sigma_i^\star \ge 0$. (See Section E.3 for full derivations for general unitarily invariant $\psi$.) This recovers several known operators as special cases.

1) Nuclear Norm: $\psi(\mathbf{X}) = b\sum_{i=1}^q \sigma_i(\mathbf{X})$. This acts as a soft-thresholding operator: $\sigma_i^\star = \begin{cases} 0 & \text{if } \sigma_i(\mathbf{G}) \le b \\ D_2 & \text{if } \sigma_i(\mathbf{G}) > b\end{cases}$.

2) Schatten $p$-norm ($p > 1$): $\psi(\mathbf{X}) = \frac{b}{p}\|\mathbf{X}\|_p^p = \frac{b}{p}\sum_{i=1}^q \sigma_i(\mathbf{X})^p$ and $\sigma_i^\star = \min\big((\frac{\sigma_i(\mathbf{G})}{b})^{\frac{1}{p-1}}, D_2\big)$.

3) Unnormalized Matrix Entropy (Tsuda et al., 2005): $\psi(\mathbf{X}) = b\sum_{i=1}^q \sigma_i(\mathbf{X})\log(\sigma_i(\mathbf{X})) - \sigma_i(\mathbf{X})$. Let

$\psi_i(x) = bx\log(x) - bx$ in (E.3). We have: $\sigma_i^\star = \min\left(\exp(\frac{\sigma_i(\mathbf{G})}{b}), D_2\right)$.

**$\ell_\infty$-norm Regularization and its Connection to Spectrally Clipped Signum.** We next discuss the choice of $\psi(\mathbf{X}) = \frac{b}{2}\|\mathbf{X}\|_\infty^2 := \frac{b}{2}\max_{i,j}\{|\mathbf{X}_{i,j}|\}^2$. For the original constrained problem (6), there exists an optimal dual variable $\mu^\star$ such that: $\mathbf{X}^\star = \arg\min_{\mathbf{X}}\{\langle\mathbf{G},\mathbf{X}\rangle + \frac{b}{2}\|\mathbf{X}\|_\infty^2 + \mu^\star(\|\mathbf{X}\|_2 - D_2)\}$. Using the first-order optimality condition, we have: $\mathbf{0} \in \mathbf{G} + b\|\mathbf{X}^\star\|_\infty\partial(\|\mathbf{X}^\star\|_\infty) + \mu^\star\partial(\|\mathbf{X}^\star\|_2)$ . If $\mu^\star = 0$, then $\mathbf{X}^\star$ is equal to the unconstrained solution:

$$\tilde{\mathbf{X}}^\star := -\frac{\|\mathbf{G}\|_1}{b}\operatorname{sign}(\mathbf{G}) \in \underset{\mathbf{X}\in\mathbb{R}^{m\times n}}{\arg\min}\left\{\langle\mathbf{G},\mathbf{X}\rangle + \frac{b}{2}\|\mathbf{X}\|_\infty^2\right\},$$

where $\|\mathbf{X}\|_1 = \sum_{i,j}|\mathbf{X}_{i,j}|$ denote the entrywise $\ell_1$-norm. Otherwise, it does not admit a simple closed-form solution. We can try to solve it by using standard convex optimization methods such as projected gradient descent. On the other hand, a natural approximation is to project the unconstrained solution onto the feasible set:

$$\tilde{\mathbf{U}} := \operatorname{proj}_{Q_2}(\tilde{\mathbf{X}}^\star) = -\frac{\|\mathbf{G}\|_1}{b}\operatorname{clip}_{\frac{bD_2}{\|\mathbf{G}\|_1}}^{\mathrm{sp}}\left(\operatorname{sign}(\mathbf{G})\right).$$

When $\|\operatorname{sign}(\mathbf{G})\|_2 \leq \frac{bD_2}{\|\mathbf{G}\|_1}$, then $\mathbf{X}^\star = \tilde{\mathbf{U}}$. If we substitute this approximate solution into our update rule (2) with $\mathbf{G}$ chosen as the momentum, we recover *Signum with post-spectral clipping*. This suggests that spectrally clipped Signum can be viewed as an approximate Frank-Wolfe step for minimizing a spectrally constrained loss regularized by the $\ell_\infty$-norm (limiting the magnitude of each coordinate).

# 4. Robustness to Sparse Spectral Spikes

In this section, we first provide empirical evidence of the existence of sparse spectral spikes in LLM training. Then we give some theoretical insights on why spectral clipping can help mitigate the adverse effects of such noise structures during optimization.

## 4.1. Empirical Observations

We first record the singular value distributions of stochastic gradients and update directions in a 124M parameter LLaMA-style transformer trained with AdamW/Signum. We observe from Figure F.9 that raw stochastic gradients exhibit heavy-tailed spectrum with high condition numbers across layers. While the AdamW/Signum updates show a more uniform spectral distribution, they fail to impose a hard ceiling (Figure F.10 and F.11). We next investigate the noise structure by decomposing gradients into signal and noise components ($\mathbf{N} = \mathbf{g} - \mathbf{G}$). We observe that stochastic noise manifests as "spectral spikes" whose top singular values frequently exceed the signal norm. Moreover, the dominant noise spikes are nearly orthogonal to the principal

directions of the signal (Figure F.14). The detailed setup and results can be found in Section F.2.

These findings provide a empirical justification for spectral clipping. Since high-magnitude noise components are geometrically separated from the signal, spectral clipping effectively truncates the noise spikes without significantly distorting the underlying signal direction.

## 4.2. Theoretical Insights

Let $\mathbf{G} \in \mathbb{R}^{m\times n}$ be the 'true' signal. Suppose we receive a stochastic signal $\mathbf{g} = \mathbf{G} + \mathbf{N}$ where $\mathbf{N} = \ell\mathbf{U}_N\mathbf{V}_N^T$ is a low-rank noise with zero-mean such that $\ell \gg \|\mathbf{G}\|_2$ (spike). We show that the spectrally clipped matrix largely preserves the signal with controlled variance.

**Definition 4.1.** Let $\mathbf{U} \in \mathbb{R}^{d\times r}$ be a zero-mean random matrix with orthonormal column, i.e., $\mathbb{E}[\mathbf{U}] = \mathbf{0}$ and $\mathbf{U}^T\mathbf{U} = \mathbf{I}_r$. We say that $\mathbf{U}$ has bounded anisotropy with parameter $\kappa > 0$, if $\mathbb{E}[\mathbf{U}\mathbf{U}^T] \preceq \frac{\kappa r}{d}\mathbf{I}_d$ , where $\mathbf{I}_d$ is the identity matrix in $\mathbb{R}^{d\times d}$.

**Lemma 4.2.** *Let $\mathbf{G} \in \mathbb{R}^{m\times n}$ be a fixed matrix with $q := \min(m,n) \gg r$. Let $\mathbf{g} = \mathbf{G}+\mathbf{N}$ where $\mathbf{N} = \ell\mathbf{U}_N\mathbf{V}_N^T$, $\ell > 0$, and $\mathbf{U}_N \in \mathbb{R}^{m\times r}$ and $\mathbf{V}_N \in \mathbb{R}^{n\times r}$ satisfy Definition 4.1 with parameter $\kappa \leq \frac{q}{25r^2}$. Let $\tilde{\mathbf{g}} = \operatorname{clip}_c^{\mathrm{sp}}(\mathbf{g})$. Suppose that $\ell \geq 9\sqrt{r}\|\mathbf{G}\|_2$. Then for any $c \geq \|\mathbf{G}\|_2$, it holds that: $\mathbb{E}_\mathbf{N}[\langle\mathbf{G},\tilde{\mathbf{g}}\rangle] \geq \frac{1}{3}\|\mathbf{G}\|_F^2$ and $\mathbb{E}_\mathbf{N}[\|\tilde{\mathbf{g}}\|_F^2] \leq r\min(c,\ell+\|\mathbf{G}\|_2)^2 + \|\mathbf{G}\|_F^2$.*

Intuitively, we can treat $\mathbf{G}$ as a perturbation to the large spike noise $\mathbf{N}$. By the standard matrix perturbation theory, the top singular values of $\mathbf{g}$ are close to those of $\mathbf{N}$ (dominated by $\ell$), and the remaining singular values are close to those of $\mathbf{G}$. Thus, we have $\operatorname{clip}_c^{\mathrm{sp}}(\mathbf{G} + \ell\mathbf{U}_N\mathbf{V}_N^T) \approx \mathbf{G} + c\mathbf{U}_N\mathbf{V}_N^T$. The proof can be found in Section D.1.

**Comparisons.** Compared with the raw stochastic signal $\mathbf{g}$ which satisfies $\mathbb{E}[\|\mathbf{g}\|_F^2] = \|\mathbf{G}\|_F^2 + r\ell^2$. Spectral clipping effectively reduces the variance term from $r\ell^2$ to $rc^2$. We next show that global clipping cannot preserve the signal as effectively as spectral clipping.

**Lemma 4.3.** *Consider the same setting as in Lemma 4.2. Let $\tilde{\mathbf{g}} = \operatorname{clip}_c^{\mathrm{g}}(\mathbf{g})$. Suppose that $\ell \geq 3\|\mathbf{G}\|_F$. If $c \leq \sqrt{r}(\ell - \|\mathbf{G}\|_2)$, then it holds that: $\frac{4c}{5\sqrt{r}\ell}\|\mathbf{G}\|_F^2 \leq \mathbb{E}_\mathbf{N}[\langle\mathbf{G},\tilde{\mathbf{g}}\rangle] \leq \frac{c}{\sqrt{r}\ell}\|\mathbf{G}\|_F^2$ and $\mathbb{E}_\mathbf{N}[\|\tilde{\mathbf{g}}\|_F^2] = c^2$. If $\sqrt{r}(\ell - \|\mathbf{G}\|_2) \leq c \leq \ell + \|\mathbf{G}\|_2$, then we have: $\mathbb{E}_\mathbf{N}[\langle\mathbf{G},\tilde{\mathbf{g}}\rangle] \leq \|\mathbf{G}\|_F^2$ and $\frac{4}{9}r\ell^2 \leq \mathbb{E}_\mathbf{N}[\|\tilde{\mathbf{g}}\|_F^2] \leq c^2$. Finally, if $c \geq \ell + \|\mathbf{G}\|_2$, then it holds that: $\mathbb{E}_\mathbf{N}[\langle\mathbf{G},\tilde{\mathbf{g}}\rangle] = \|\mathbf{G}\|_F^2$ and $\mathbb{E}_\mathbf{N}[\|\tilde{\mathbf{g}}\|_F^2] = \|\mathbf{G}\|_F^2 + r\ell^2$.*

The previous lemma shows that for any $c > 0$, global clipping either have small signal preservation, $\mathbb{E}[\langle\mathbf{G},\tilde{\mathbf{g}}\rangle] \simeq \frac{c}{\ell}\|\mathbf{G}\|_F^2$ which decreases linearly as $\ell$ increases, or have large variance scaling as $\ell^2$. This is because global clipping scales down the entire matrix uniformly, including the useful signal components.

From the previous lemmas, we can immediately derive convergence guarantees for SGD with spectral clipping in the presence of sparse spectral spikes. Consider the minimization problem:

$$\min_{\mathbf{X} \in \mathbb{R}^{m \times n}} f(\mathbf{X}) \,. \tag{7}$$

We assume that the objective function $f$ is $L_{\|\cdot\|_F}$-smooth and has bounded gradient.

**Assumption 4.4.** There exists $M, B > 0$ such that for any $\mathbf{X} \in \mathbb{R}^{m \times n}$, we have:

$$\|\nabla f(\mathbf{X})\|_2 \le M, \quad \|\nabla f(\mathbf{X})\|_F \le B \,. \tag{8}$$

**Assumption 4.5.** At any point $\mathbf{X} \in \mathbb{R}^{m \times n}$, the stochastic gradient oracle returns $\mathbf{g}(\mathbf{X}) = \nabla f(\mathbf{X}) + N(\mathbf{X})$, where $N(\mathbf{X}) = \ell \mathbf{U}_X \mathbf{V}_X^T$, $\ell > 0$, and $\mathbf{U}_X \in \mathbb{R}^{m \times r}$ and $\mathbf{V}_X \in \mathbb{R}^{n \times r}$ are independent random matrices that satisfy Definition 4.1 with parameter $r^2 \kappa \ll q := \min(m, n)$.

Assumption 4.5 implies that for any $\mathbf{X} \in \mathbb{R}^{m \times n}$, the stochastic gradient is unbiased: $\mathbb{E}[\mathbf{g}(\mathbf{X})] = \nabla f(\mathbf{X})$, and its variance is bounded: $\mathbb{E}[\|\mathbf{g}(\mathbf{X}) - \nabla f(\mathbf{X})\|_F^2] = r\ell^2$. Therefore, to reach $\mathbb{E}[\|\nabla f(\hat{\mathbf{X}}_K)\|_F]^2 \le \epsilon^2$, Vanilla SGD requires at most $K = \mathcal{O}\left(\frac{L_F F^0}{\epsilon^2} + r\ell^2 \frac{L_F F^0}{\epsilon^4}\right)$ iterations, where $F^0 = f(\mathbf{X}_0) - f^*$ and $\hat{\mathbf{X}}_K$ is uniformly sampled from $\{\mathbf{X}_0, \dots, \mathbf{X}_{K-1}\}$. We next compare the convergence rates of the following two methods:

$$\mathbf{X}_{k+1} = \mathbf{X}_k - \eta \, \mathrm{clip}_c^{\mathrm{sp}}(\mathbf{g}(\mathbf{X}_k)) \,, \tag{9}$$

$$\mathbf{X}_{k+1} = \mathbf{X}_k - \eta \, \mathrm{clip}_c^{\mathrm{g}}(\mathbf{g}(\mathbf{X}_k)) \,, \tag{10}$$

**Theorem 4.6.** *Let SGD with spectral clipping* (9) *be applied to problem* (7) *under Assumptions 3.3, 4.4, and 4.5 with $25\kappa r^2 \le q := \min(m, n)$. For any $\ell > 0$, by choosing $\eta = 1/(\frac{6rL_F \min(10\sqrt{r}M, 10\ell/9)}{\epsilon^2} + 3L_F)$ and $c = 10\sqrt{r}M$, to reach $\mathbb{E}[\|\nabla f(\hat{\mathbf{X}}_K)\|_F^2] \le \epsilon^2$, we need at most*

$$K \le \frac{36 L_F F^0}{\epsilon^2} + 72 \min(10\sqrt{r}M, 10\ell/9)^2 \frac{r L_F F^0}{\epsilon^4}$$

$$\simeq \frac{L_F F^0}{\epsilon^2} + \min(\sqrt{r}M, \ell)^2 \frac{r L_F F^0}{\epsilon^4}$$

*iterations, where $F^0 := f(\mathbf{X}_0) - f^*$ and $\hat{\mathbf{X}}_K$ is uniformly sampled from $\{\mathbf{X}_0, \dots, \mathbf{X}_{K-1}\}$.*

**Theorem 4.7.** *Consider SGD with global clipping* (10) *under the same setting and notations as in Theorem 4.6. For any $\ell \ge 3B$, by choosing $c \le \frac{2}{3}\sqrt{r}\ell$ and $\eta = \sqrt{\frac{2F^0}{L_F K c^2}}$, to reach $\mathbb{E}[\|\nabla f(\hat{\mathbf{X}}_K)\|_F^2] \le \epsilon^2$, we need at most $K \le \frac{4r\ell^2 L_F F^0}{\epsilon^4}$ iterations.*

From Theorem 4.6, we see that SGD with spectral clipping achieves an improvement of dependence from $r\ell^2$ to $r \min(\sqrt{r}M, \ell)^2$. Meanwhile, it is robust to any spike noise level $\ell$ by choosing the clipping threshold $c \simeq \sqrt{r}M$. We discuss the performance of SGD-M with both pre and post-spectral clipping under Assumption 4.5 in Section E.2.

## 5. *SPECTRA*: SPEctral Clipping for LLM TRaining Acceleration

In this section, we formally present *SPECTRA*, a new broad class of optimizers that leverage spectral clipping to enhance the training of LLMs.

### 5.1. Soft Spectral Clipping

The computation of the exact spectral clipping is costly as it requires full SVD decomposition. To overcome this bottleneck, we propose *Soft Spectral Clipping* (SSC), a new simple approximation of the spectral clipping operator, which only requires square matrix–matrix multiplications, making it efficient on GPU hardware. Let $\mathrm{clip}_c(x) := \mathrm{sign}(x)\min(|x|, c)$ be the scalar clipping function. Consider the smooth approximation of $\mathrm{clip}_c(x)$: $h_c(x) = \frac{x}{\sqrt{1 + x^2/c^2}}$. As illustrated in Figure E.1, $h_c(x)$ behaves linearly for small $|x| < c$ and saturates smoothly to $\mathrm{sign}(x)c$ for large $|x| > c$. For any $x$, it satisfies $|h_c(x)| \le |\mathrm{clip}_c(x)|$. The error bound can be found in Lemma C.14. Similarly, for any matrix $\mathbf{X} \in \mathbb{R}^{m \times n}$ (assuming $m \le n$ w.l.o.g.), applying $h_c$ to the singular values of $\mathbf{X}$ yields the SSC operator (which is easy to be verified):

$$H_c(\mathbf{X}) := (\mathbf{I} + \mathbf{X}\mathbf{X}^T/c^2)^{-\frac{1}{2}} \mathbf{X} \,.$$

To compute the term $(\mathbf{I} + \mathbf{X}\mathbf{X}^T/c^2)^{-\frac{1}{2}}$ efficiently, we utilize the classic Newton-Schulz iteration (Algorithm 3), a well-established method for finding the inverse square root of a positive definite matrix (Schulz, 1933; Higham, 2008; Song et al., 2022). Since $\mathbf{I} + \mathbf{X}\mathbf{X}^T/c^2$ has eigenvalues strictly bounded below by 1, it is well-conditioned, ensuring the fast convergence of the Newton method. The full procedure is detailed in Algorithm 1. As shown in Figure E.2 and Table F.2, the computational overhead is moderate and its fraction of total per-step time shrinks dramatically as scale grows —making it comparable to other scalable operations like those in Muon (Jordan et al., 2024).

### 5.2. Algorithm Description

We provide the complete pseudocode for *SPECTRA* in Algorithm 2. The framework is designed to be optimizer-agnostic, accepting any base optimizer that follows update rule (1). While we discuss $\mathbf{X}$ mostly as a matrix, it naturally extends to 1D parameters. The key components are the same as discussed before: 1) Post-Spectral Clipping applied to the update and 2) An optional pre-processing step for raw gradients, which is effective for removing 'sparse spike' noise before it corrupts the momentum buffer states, typically useful with small batchsize.

**Recommended Hyperparameter Configurations.**

1. **Scaling Factor** $\alpha$. To ensure consistent optimization

**Algorithm 1** Soft Spectral Clipping $\approx \mathrm{clip}_c^{\mathrm{sp}}(\mathbf{X})$

1: **Input:** $\mathbf{X} \in \mathbb{R}^{m \times n}, c > 0, N_s > 0$.
2: **Signature:** $\mathrm{SSC}_c(\mathbf{X}, N_s)$.
3: Set $s_{\max}^2 = \min(\|\mathbf{X}\mathbf{X}^T\|_F, \max_i \sum_{j=1}^n |(\mathbf{X}\mathbf{X}^T)_{ij}|)$.[1]
4: **if** $s_{\max} \leq c$ **then**
5:    **Return X**
6: **else**
7:    Set $\alpha = 1 + (s_{\max}/c)^2$.
8:    Compute $\mathbf{A} = \mathrm{MISR}(\mathbf{I} + \mathbf{X}\mathbf{X}^T/c^2, \alpha, N_s)$.[2]
9:    **Return: AX**.
10: **end if**

---

[1]$s_{\max}^2$ is an upper bound on the largest singular value of $\mathbf{X}\mathbf{X}^T$. By the Gershgorin circle theorem, $\sigma_1(\mathbf{X}\mathbf{X}^T)$ is bounded by the largest 1-norm of any of its rows.
[2]MISR can be found in Algorithm 3.

---

**Algorithm 2** *SPECTRA*

1: **Input:** Base optimizer: $\mathcal{BO}$, learning rate schedule $\mathcal{S}_{lr}$, clipping schedule $\mathcal{S}_c$, Pre-SPC $\in \{\mathrm{True}, \mathrm{False}\}$, $c_{\mathrm{pre}} > 0, c, \eta, \lambda, \alpha, N_s > 0, \mathbf{X}_0 \in \mathbb{R}^{m \times n}$.
2: **for** $k = 0, 1, 2 \ldots, K - 1$ **do**
3:    Compute stochastic gradient $\mathbf{g}(\mathbf{X}_k)$.
4:    **if** Pre-SPC = True **then**
5:       $\tilde{\mathbf{g}}_k = \mathrm{SSC}_{c_{\mathrm{pre}}}(\mathbf{g}(\mathbf{X}_k), N_s)$.
6:    **else**
7:       $\tilde{\mathbf{g}}_k = \mathbf{g}(\mathbf{X}_k)$.
8:    **end if**
9:    Compute $c_k = \mathcal{S}_c(c, k)$ and $\eta_k = \mathcal{S}_{lr}(\eta, k)$.
10:    Compute $\mathbf{U}_k$ using $\tilde{\mathbf{g}}_k$ according to $\mathcal{BO}$.
11:    $\mathbf{X}_{k+1} = (1 - \lambda \eta_k)\mathbf{X}_k - \alpha \eta_k \mathrm{SSC}_{c_k}(\mathbf{U}_k, N_s)$.
12: **end for**

---

behavior across layers with varying dimensions, we align our spectral constraint with the RMS norm, defined as $\|\mathbf{A}\|_{\mathrm{rms} \to \mathrm{rms}} = \sup_{\|z\|_{\mathrm{rms}}=1} \|\mathbf{A}z\|_{\mathrm{rms}} = \sqrt{\frac{n}{m}}\|\mathbf{A}\|_2$, for $\mathbf{A} \in \mathbb{R}^{m \times n}$, where $\|x\|_{\mathrm{rms}} = \frac{1}{\sqrt{d}}\|x\|_2$ for $x \in \mathbb{R}^n$. A good practical is to allow $D_{\mathrm{rms}} \propto \max(1, \sqrt{n/m})$ to have conservative capacity scaling (Jordan et al., 2024; Pethick et al., 2025). If we target a constraint radius $D_{\mathrm{rms}}$, the equivalent spectral radius is $D_2 = \sqrt{\frac{m}{n}}D_{\mathrm{rms}}$. According to Theorem 3.4, we get $\alpha = \frac{\lambda}{b} = \frac{\lambda D_2}{c}$. By choosing $D_2 = \max(\sqrt{\frac{m}{n}}, 1)\frac{c}{\lambda}$, we obtain $D_{\mathrm{rms}} = \max(\sqrt{\frac{n}{m}}, 1)\frac{c}{\lambda}$ and $\alpha = \max(\sqrt{\frac{m}{n}}, 1)$.

2. $c$**,** $\lambda$ **and** $N_s$. With $\alpha$ fixed, the ratio $\frac{c}{\lambda}$ controls the radius of the spectral norm ball ($D_2 \propto \frac{c}{\lambda}$) and $\lambda$ determines the regularization level $b = \lambda/\alpha$. A common and robust choice is $\lambda \approx 0.1$ and $c \approx 10$. Finally, we find that $N_s = 10$ provides a consistent performance for all the experiments.

3. **Dynamic Clipping Schedules.** During the initial learning rate warm-up phase ($k < K_{\mathrm{warm}}$), we recommend main-

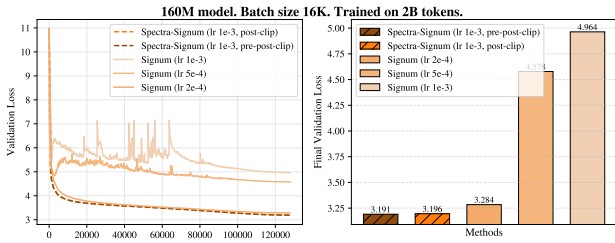

*Figure 2.* Validation loss comparison under small batch size training. Spectra-Signum maintains stability at the large learning rate, whereas Signum exhibits training instability (Signum typically prefers smaller LR than Adam-ish methods (Semenov et al., 2025)). Spectra-Signum with both pre and post clipping almost overlap with Spectra-Signum with post clipping.

taining a constant effective update capacity. Since the update magnitude is bounded $c_k \eta_k$, simply using a fixed small $c$ with a small $\eta_k$ would overly restrict the training. Instead, we scale inversely $c_k = \frac{c\eta}{\eta_k}$. For $k \geq K_{\mathrm{warm}}$, we set $c_k \equiv c$ by default across all the experiments.

## 6. Experiments

### 6.1. Synthetic Experiments.

We validate our theory on a synthetic matrix logistic regression task. Using the update rule (4) with hyperparameters from Theorem 3.4, Figure F.1 and F.4 confirm that our method correctly minimizes the composite objective $F(\mathbf{X})$ within the spectral norm ball, where appropriate regularization improves generalization (details in Section F.1.1).

Next, we test robustness against rank-$r$ spectral spike noise $\mathbf{N} = \ell \mathbf{U}\mathbf{V}^\top$. As shown in Figure F.5, spectral clipping maintains fast convergence regardless of the noise scale $\ell$ using fixed hyperparameters. In contrast, vanilla and globally clipped SGD require significantly reduced learning rates to prevent divergence. These benefits also extend to momentum-based updates (Figure F.6; see Section F.1.2).

### 6.2. LLM Pretraining Tasks.

We compare SPECTRA against state-of-the-art optimizers, including AdEMAMix (Pagliardini et al., 2025), AdamW (Loshchilov & Hutter, 2019), Signum (Bernstein et al., 2018), D-Muon (Liu et al., 2025), Mars (Yuan et al., 2025), and SOAP (Vyas et al., 2024), following the benchmark protocol in (Semenov et al., 2025). We apply SPECTRA to the first three optimizers to demonstrate its versatility across methods with different **memory requirements** (see Table F.3). We employ LLaMA3-based architectures (Grattafiori et al., 2024) with untied embeddings (details in Table F.1) trained on FineWeb-Edu dataset (Lozhkov et al., 2024). Hyperparameter choices are reported in Section F.3. Unless otherwise stated, we disable pre-spectral clipping for SPECTRA to isolate the contribution of post-

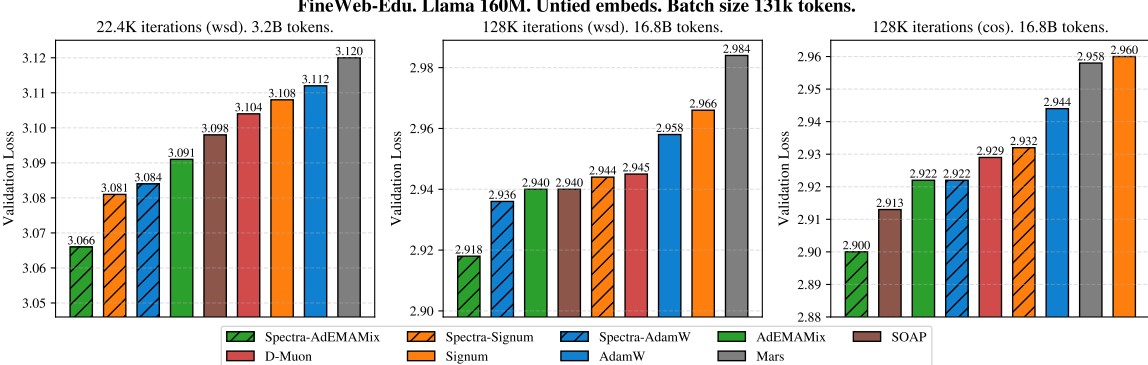

*Figure 3.* Final validation loss comparison for a small size Llama model trained with both cos and wsd learning rate schedule. The run time comparisons w/wo using SPECTRA are provided in Figure F.8. We use a total batchsize of 256 with 512 sequence length. The hyperparameters such as learning rate used for each method are reported in the tables in Section F.3.

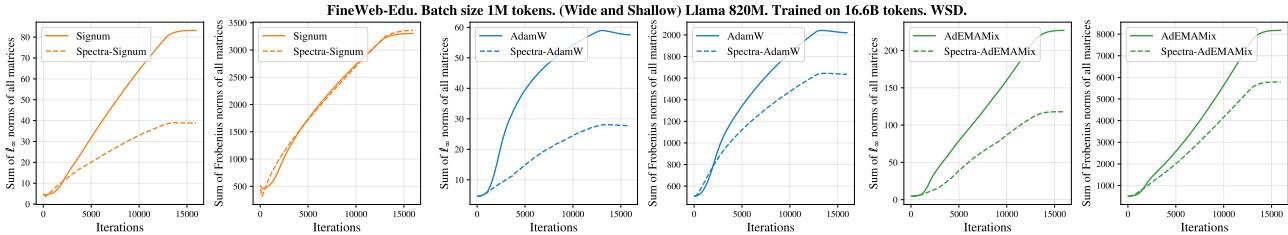

*Figure 4.* Evolution of the aggregated $\ell_\infty$ and Frobenius norms of all weight matrices during training. SPECTRA consistently maintains a lower $\ell_\infty$ norm compared to the base optimizer.

spectral clipping, which is the primary component of the method (see the discussion at the end of Section F.3). We mostly use the WSD schedule (Hägele et al., 2024) for its practical advantage of enabling resumption from any checkpoint during the stable phase (constant learning rate).

**SPECTRA Enables Large Learning Rate.** We begin by demonstrating the stability of SPECTRA under large learning rates. We train a 124M-parameter transformer using Signum with a small batch size of $16k$ tokens. Previous studies note that Signum is sensitive to gradient noise, often performing poorly in small-batch regimes (Kornilov et al., 2025; Tomihari & Sato, 2025). Consistent with this, Figure 2 shows that standard Signum exhibits training instability and poor validation loss when the learning rate is increased. In contrast, Spectra-Signum maintains stable convergence at a high learning rate, significantly outperforming the best Signum baseline (improving validation perplexity by 2.5). These results highlight the efficiency of SPECTRA in mitigating instability caused by gradient noise.

**160M-parameter Model.** We use a sequence length of 512 tokens and batch size of 256. We first consider the Chinchilla optimal duration (Hoffmann et al., 2022) for this scale (3.2B tokens, approx. 24.4K iterations). As shown in the leftmost plot of Figure 3, the three SPECTRA variants achieve the lowest validation losses. When increasing the training budget to 16B tokens (testing both WSD and Cosine

schedules), Spectra-AdEMAMix consistently outperforms other methods by a large margin. The run time comparison with and without SPECTRA is presented in Figure F.8. We also evaluate the **post-orthogonalization** operator; results in Table F.8 indicate that post-spectral clipping yields superior performance. The ablation study in terms of $c$ can be found in Table 3 where the wide plateau suggests that the method is not sensitive to $c$ in practice. We also ablate over $N_s \in \{5, 10, 15\}$ and results in Table 4 show almost no difference in final loss. We use $N_s = 10$ as a conservative default to ensure convergence across all training phases (e.g., early warmup where conditioning can be poor)

**820M Wide and Shallow Model.** Following (Semenov et al., 2025), we scale the model to 820M by increasing the embedding dimension while maintaining the number of layers. The Chinchilla optimal duration for this size is approximately 16.4B tokens. We increase the sequence length from 512 to 1K and batch size from 256 to 992 (1M tokens per batch). We run each method for 16K iterations; final validation losses are reported in the right plot of Figure 1. SPECTRA consistently improves upon the base optimizers. Training curves with respect to both iterations and runtime are presented in Figure F.7. Figure 4 plots the summation of the $\ell_\infty$ and Frobenius norms of all weight matrices during training; all three SPECTRA variants exhibit smaller $\ell_\infty$ norms, confirming the implicit $\ell_\infty$-regularization effect of sign-based methods.

*Table 1.* The evaluation results (accuracy $\times 100\%$) of the 820M model pretrained on Fineweb-Edu with 16.8B tokens (5-shot with lm-evaluation-harness (Gao et al., 2024)).

| Optimizer | ARC-c | ARC-e | BoolQ | PIQA | HellaSwag | WinoGrande | OBQA | $\lambda$-OpenAI | Sciq | MMLU | Average |
|---|---|---|---|---|---|---|---|---|---|---|---|
| Mars | 34.98 | 67.80 | 57.25 | 70.08 | 46.87 | 52.17 | 34.20 | **29.38** | 89.60 | 25.37 | 50.77 |
| D-Muon | 36.60 | 67.47 | 53.73 | 69.31 | 46.69 | 52.72 | 34.60 | 28.84 | **90.90** | 25.08 | 50.59 |
| Signum | 34.56 | 65.66 | 56.02 | 69.64 | 45.01 | 52.25 | 34.20 | 28.04 | 87.30 | 25.75 | 49.84 |
| Spectra-Signum | 35.58 | **68.81** | **58.10** | **70.46** | 47.04 | **54.46** | 35.60 | 29.36 | 90.30 | **26.44** | **51.62** |
| AdamW | 33.45 | 66.84 | 56.09 | 69.21 | 45.65 | 50.99 | 35.20 | 25.54 | 90.80 | 23.86 | 49.76 |
| Spectra-AdamW | 35.67 | 66.96 | 55.41 | 68.82 | 46.57 | 52.09 | **36.40** | 26.74 | 89.20 | 25.14 | 50.30 |
| AdEMAMix | 35.41 | 65.95 | 43.43 | 70.08 | 46.43 | 53.20 | 34.40 | 28.51 | 88.30 | 24.84 | 49.05 |
| Spectra-AdEMAMix | **36.86** | 68.81 | 53.94 | 69.48 | **47.09** | 53.43 | 36.40 | 28.26 | 89.40 | 26.23 | 50.99 |

*Table 2.* Fixed-time budget comparison at 780M scale. Baseline optimizers are given 5–20% more training steps to compare with SPECTRA's 2% per-step overhead. SPECTRA variants trained for 16000 steps are shown in the last column.

| Optimizer | Baseline (extended steps) | | | | SPECTRA |
|---|---|---|---|---|---|
| | 16000 | +5% (16800) | +10% (17600) | +20% (19200) | 16000 |
| AdamW | 2.608 | 2.605 | 2.600 | 2.590 | 2.582 |
| Signum | 2.600 | 2.594 | 2.589 | 2.580 | 2.573 |
| AdEMAMix | 2.591 | 2.585 | 2.580 | 2.571 | 2.574 |
| MARS | 2.598 | 2.592 | 2.587 | 2.579 | 2.573 |
| D-Muon | 2.595 | 2.589 | 2.584 | 2.574 | — |

*Table 3.* Sensitivity to clipping threshold $c$ (160M, WSD schedule, lr=$10^{-3}$). Baseline AdamW without SPECTRA: 2.958. Performance is flat from $c = 5$ to $c = 50$ (all within 0.004 of the optimum). Only $c = 1$ degrades due to over-aggressive clipping.

| $c$ | 1 | 5 | 10 | 20 | 50 |
|---|---|---|---|---|---|
| Val Loss | 2.975 | 2.940 | **2.936** | **2.936** | 2.940 |

We next evaluate the downstream performances of the pretrained models on 10 standard in-context learning tasks. Details are provided in Section F.4 and the results are reported in Table 1. All three SPECTRA variants improve average accuracy over the baselines, with Spectra-Signum achieving the best.

**780M Deep and Thin Model.** We next consider a modern deep architecture by increasing the number of layers and reducing the embedding dimension, a configuration known to be harder to optimize. The left plot in Figure 1 shows that the four SPECTRA variants rank top 4. Notably, Spectra-Signum performs well while requiring the least memory. Table 2 further shows that even with 20% more training steps, baseline methods remain on par with or worse than SPECTRA variants trained for the original number of steps. Since SPECTRA's per-step overhead is only around 2% at this scale (Table F.2), this demonstrates that SPECTRA is not only token-efficient but also time-efficient.

**1.5B Model with $\mu$P.** To evaluate SPECTRA at larger scale, we conduct short training runs (8B tokens, 8k steps) on

*Table 4.* Ablation of number of NS iterations for Spectra-AdamW (160M, WSD schedule, lr=$10^{-3}$). bfloat16 has only 7 bits of mantissa. Additional iterations beyond convergence don't improve accuracy–they accumulate round-off errors. Therefore, we use float32 for NS=15 instead.

| NS Iters | 5 (bf16) | 10 (bf16) | 15 (float32) |
|---|---|---|---|
| Val Loss | 2.939 | 2.936 | 2.937 |

a 1.5B-parameter model with $\mu$P (Yang et al., 2021). As shown in Table F.5, SPECTRA-AdEMAMix achieves the best validation loss among all considered methods.

**Summary and Discussion.** In general, Spectra-AdEMAMix reliably achieves the best validation losses across varying training durations and model sizes, while Spectra-Signum excels with large batch sizes and demonstrates strong performance on downstream tasks. Overall, we recommend Spectra-Signum as a strong and memory-efficient optimizer for standard LLm pretraining when the batchsize is sufficiently large.

# 7. Conclusion.

We proposed SPECTRA, a novel spectral optimization framework designed for efficient LLM training. By explicitly constraining the spectral norm of the update matrix, SPECTRA enhances convergence stability and final validation performance compared to standard baselines, while avoiding additional memory overhead. Future directions include: 1) scaling the experiments up to models of larger sizes, extending training durations, and evaluating performance on broad downstream tasks; 2) theoretical analysis of spectral clipping under more general noise assumptions; 3) empirical comparions of using other spectral clipping implementations such as (Cesista, 2025) and the currently used soft spectral clipping; 4) exploring efficient distributed implementations of spectral clipping for multi-gpu training especially under tensor parallelism; 5) investigating the effect of explicitly using various weight regularization techniques; and 6) exploring the potential of spectral clipping in other optimization contexts, such as reinforcement learning.

## Acknowledgments

This work is funded by the Deutsche Forschungsgemeinschaft (DFG, German Research Foundation) – Project number 553954458. The authors thank Anton Rodomanov and Nikita Doikov for helpful discussions regarding the paper. This work was also supported by the Helmholtz Association's Initiative and Networking Fund on the HAICORE@FZJ partition.

## Impact Statement

This paper presents work that aims to improve the training of large language models. Our work contributes to the efficiency of deep learning research. This has potential positive impacts regarding the environmental footprint of training large-scale models and lowering the hardware barrier for research participation. We do not foresee specific negative societal consequences beyond the general risks associated with the advancement of LLM capabilities.

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

# Appendix

## A. More Related Work

**Clipping and Optimization.** Gradient clipping has evolved from a heuristic to prevent exploding gradients in recurrent networks (Pascanu et al., 2013) into a critical component of modern optimizer design. Broadly, global and coordinate-wise clipping are the most widely studied variants due to their computational efficiency. Recent theoretical analyses have rigorously justified the benefits of clipping beyond simple stability. Pan & Li (2023) demonstrate the effect of coordinate-wise clipping on sharpness reduction and speeding up the convergence. Zhang et al. (2019) highlight the advantages of global clipping and normalization under the generalized smoothness assumptions which appears in training deep neural networks. Furthermore, Zhang et al. (2020) establish that clipping significantly improves robustness to heavy-tailed noise, effectively mimicking the behavior of adaptive methods. Subsequently, more advanced theoretical results are established with refined convergence guarantees under various assumptions (Gorbunov et al., 2020; 2024; Koloskova et al., 2023) Recently, Chezhegov et al. (2025) further demonstrated the improvement of Adam-Norm and AdaGrad-Norm under heavy-tailed noise with global clipping.

**Connection to Robust and Byzantine Optimization**. The mechanics of gradient clipping share a strong theoretical foundation with robust statistics and Byzantine-tolerant optimization (Alistarh et al., 2018). In distributed learning scenarios where a subset of workers may be malicious, robust aggregation rules often employ clipping-like operations to filter out outliers (Karimireddy et al., 2021; Malinovsky et al., 2024). In this view, gradient clipping in standard training can be seen as a robust estimator that filters out "stochastic outliers" in the gradient estimation, thereby stabilizing the learning process.

**Spectral Clipping**. Global and coordinate-wise clipping treat all gradients as flat vectors, ignoring the structural information inherent in matrix parameters. Spectral clipping has recently emerged as a more principled alternative for matrix-based models. For instance, Guo et al. (2026) show that post-hoc spectral clipping can enforce stability in linear dynamical systems more effectively than constrained optimization. Boroojeny et al. (2024) apply spectral constraints to implicitly linear layers to improve generalization and adversarial robustness. Most recently, Cesista (2025) proposed efficient algorithms for computing exact spectral clipping using block-matrix iterations, framing the method as steepest descent on the spectral ball. We leave the detailed comparisons between these techniques and our currently used soft spectral clipping as interesting future explorations.

**Spectral Optimizer & Normalization.** The concept of optimizing in the spectral domain traces back to early works by Carlson et al. (2015b), who proposed stochastic spectral descent—performing steepest descent in the spectral norm—to accelerate the training of Restricted Boltzmann Machines and deep networks (Carlson et al., 2015a; 2016). Later, the Shampoo optimizer (Gupta et al., 2018) bridged the gap between spectral methods and adaptive preconditioning. Operating as a matrix-valued version of AdaGrad (Duchi et al., 2011), Shampoo approximates second-order information via Kronecker structures; recent analysis suggests it can be viewed as a form of spectral descent when the preconditioner accumulation is removed (Bernstein & Newhouse, 2024).

More recently, Muon (Jordan et al., 2024) has popularized spectral optimization by combining momentum with orthogonalized updates computed via Newton-Schultz (NS) iterations, achieving remarkable efficiency in nanoGPT benchmarks. Theoretical foundations for these methods have since emerged: Pethick et al. (2025) demonstrate that Muon with decoupled weight decay effectively performs Frank-Wolfe optimization within a spectral-norm ball. By generalizing the underlying norms, they proposed the Stochastic Conditional Gradient (SCG) framework, which exhibits favorable hyperparameter transfer properties. Further theoretical analyses regarding convergence and structural benefits have been established (Xie et al., 2025; Riabinin et al., 2025; Shen et al., 2025; Crawshaw et al., 2025). Geometric interpretations also link these methods to steepest descent in specific norms: just as sign-based methods exploit $\ell_\infty$ geometry (Xie et al., 2024; Yadav et al., 2025), removing weight decay from Muon corresponds to steepest descent in the spectral norm space. Recent work by (Gonon et al., 2026) provides novel theoretical insights into the Muon optimizer by studying its behavior on simple quadratics, revealing that polar-factor approximation errors can qualitatively change discrete-time optimization dynamics and improve finite-time performance — a phenomenon not captured by existing worst-case or single-step analyses.

Building on the success of Muon, numerous variants have been proposed to improve adaptivity and efficiency. These include hybrids like AdaMuon (Si et al., 2025) and AdaGrad-Muon (Zhang et al., 2025), as well as efficiency-focused methods like Drop-Muon (Gruntkowska et al., 2025) and ASGO (An et al., 2025). Parallel efforts have optimized the

core orthogonalization operation itself: Dion (Ahn et al., 2025) introduces a distributed algorithm using amortized power iterations to avoid full-matrix reconstruction, while Amsel et al. (2025) and Boumal & Gonon propose refined iterative schemes to better approximate the orthogonalization operator. Beyond matrix methods, normalization techniques have also been applied to coordinate-wise optimizers (Wang et al., 2025) and standard SGD (Ma et al., 2025). Most recently, Davis & Drusvyatskiy (2025) provided rigorous justifications for the conditions under which spectral gradient updates help in DL.

## B. Basic Definitions

**Definition B.1.** A differentiable function $f : \mathbb{R}^{m \times n} \to \mathbb{R}$ is called $L_{\|\cdot\|}$-smooth for some $L \geq 0$ if for all $\mathbf{X}, \mathbf{Y} \in \mathbb{R}^{m \times n}$,

$$\|\nabla f(\mathbf{X}) - \nabla f(\mathbf{Y})\|_* \leq L\|\mathbf{X} - \mathbf{Y}\| \,,$$

where $\|\cdot\|_*$ is the dual norm of $\|\cdot\|$. If $f$ is $L_{\|\cdot\|}$-smooth, then for any $\mathbf{X}, \mathbf{Y} \in \mathbb{R}^{m \times n}$, we have (Nesterov (2018), Lemma 1.2.3):

$$f(\mathbf{Y}) \leq f(\mathbf{X}) + \langle \nabla f(\mathbf{X}), \mathbf{Y} - \mathbf{X} \rangle + \frac{L}{2} \|\mathbf{Y} - \mathbf{X}\|^2 \,. \tag{B.1}$$

**Definition B.2** ((Golub & van Loan, 2013, Section 2.5.3)). Let $\mathbf{X}, \mathbf{Y} \in \mathbb{R}^{m \times r}$ be two matrices with orthonormal columns. The distance between the subspaces spanned by those columns is defined as:

$$\text{dist}(\mathbf{X}, \mathbf{Y}) := \|\mathbf{X}\mathbf{X}^T - \mathbf{Y}\mathbf{Y}^T\|_2 \,.$$

**Definition B.3** ((Åke Björck & Golub, 1973; Golub & van Loan, 2013)). Let $\mathbf{X}, \mathbf{Y} \in \mathbb{R}^{m \times r}$ be two matrices with orthonormal columns. Let $\mathbf{U} \, \text{diag}(\cos(\theta_1), \ldots, \cos(\theta_r))\mathbf{V}^T$ be the singular value decomposition of $\mathbf{X}^T\mathbf{Y}$. Then $\theta_1, \ldots, \theta_r \in [0, \pi/2]$ are called the principal angles between the subspaces spanned by the columns of $\mathbf{X}$ and $\mathbf{Y}$.

## C. Useful Lemmas

**Lemma C.1.** *For any two matrices* $\mathbf{X}, \mathbf{Y} \in \mathbb{R}^{m \times n}$ *and any* $\gamma > 0$*, we have:*

$$|\langle \mathbf{X}, \mathbf{Y} \rangle| \leq \frac{\gamma}{2} \|\mathbf{X}\|_F^2 + \frac{1}{2\gamma} \|\mathbf{Y}\|_F^2 \,, \tag{C.1}$$

$$\|\mathbf{X} + \mathbf{Y}\|_F^2 \leq (1 + \gamma) \|\mathbf{X}\|_F^2 + \left(1 + \frac{1}{\gamma}\right) \|\mathbf{Y}\|_F^2 \,. \tag{C.2}$$

**Lemma C.2.** *Let* $(\mathbf{A}_k)_{k=0}^{+\infty}$*,* $(\mathbf{B}_k)_{k=0}^{+\infty}$ *and be two sequences in* $\mathbb{R}^{m \times n}$ *and* $(a_k)_{k=0}^{+\infty}$ *be a non-negative sequence such that* $\mathbf{A}_{k+1} = a_k \mathbf{A}_k + \mathbf{B}_k$ *for any* $k \geq 0$*. Then it holds that, for any* $K \geq 1$*,*

$$\mathbf{A}_K = \Pi_{k=0}^{K-1} a_k \mathbf{A}_0 + \sum_{k=1}^{K} (\Pi_{i=k}^{K-1} a_i)\mathbf{B}_{k-1} \,,$$

*where we define* $\Pi_{i=k_1}^{k_2} a_i = 1$ *for any* $k_1 > k_2$*.*

*Proof.* If there exists $i$ such that $a_i = 0$, then the claim clearly holds. Now suppose $a_i > 0$ for any $i \geq 0$. Let $A_k = \Pi_{i=0}^{k-1} a_i$ for $k \geq 1$ and $A_0 = 1$. Dividing the main recurrence by $A_{k+1}$, we have for any $k \geq 0$:

$$\frac{\mathbf{A}_{k+1}}{A_{k+1}} = \frac{\mathbf{A}_k}{A_k} + \frac{\mathbf{B}_k}{A_{k+1}} \,.$$

Summing up from $k = 0$ to $k = K - 1$, we get, for any $K \geq 1$:

$$\frac{\mathbf{A}_K}{A_K} = \frac{\mathbf{A}_0}{A_0} + \sum_{k=0}^{K-1} \frac{\mathbf{B}_k}{A_{k+1}} \,.$$

Rearranging, we get the claim. $\qquad \square$

**Lemma C.3.** *Let $\gamma_k = \frac{2}{k+2}$ and $\beta_k = \frac{1}{(k+1)^{2/3}}$ for $k \geq 0$. Let $S_k = \sum_{i=1}^{k}\big(\Pi_{j=i}^{k-1}(1-\beta_j)\big)\gamma_{i-1}$ ($S_1 = \gamma_0 = 1$) and $B_k = \sum_{i=1}^{k}\big(\Pi_{j=i}^{k-1}(1-\beta_j)\big)^2\beta_{i-1}^2$ ($B_1 = \beta_0^2 = 1$) for $k \geq 1$. Then for any $k \in \mathbb{N}^+$, it holds that:*

$$S_k \leq \frac{3}{k^{1/3}} \qquad and \qquad B_k \leq \frac{1}{k^{2/3}} \ .$$

*Proof.* We prove both claims by induction. For any $k \geq 2$, we have:

$$S_k = \gamma_{k-1} + \sum_{i=1}^{k-1}\big(\Pi_{j=i}^{k-1}(1-\beta_j)\big)\gamma_{i-1} = \gamma_{k-1} + (1-\beta_{k-1})S_{k-1} = \Big(1 - \frac{1}{k^{2/3}}\Big)S_{k-1} + \frac{2}{k+1} \ .$$

For $k = 1$, we have $S_1 = 1 \leq 3$. For $k \geq 2$, suppose the claim holds for $k-1$. Then we have $S_{k-1} \leq \frac{3}{(k-1)^{1/3}}$ and we need to show that: $S_k \leq 3\big(1 - k^{-2/3}\big)(k-1)^{-1/3} + \frac{2}{k+1} \leq 3k^{-1/3}$. Note that

$$(k-1)^{-1/3} = k^{-1/3}(1-1/k)^{-1/3} = k^{-1/3}\Big(1 + \frac{1}{k-1}\Big)^{1/3} \leq k^{-1/3}\Big(1 + \frac{1}{3(k-1)}\Big) \ ,$$

where the last inequality follows from the concavity of the function $(1+x)^{1/3}$. Therefore, it suffices to show that:

$$3(1 - k^{-2/3})\Big(k^{-1/3} + \frac{k^{-1/3}}{3(k-1)}\Big) + \frac{2}{k+1} \leq 3k^{-1/3} \ .$$

The left hand side can be simplified as $3k^{-1/3} + k^{-1/3}/(k-1) - 3k^{-1} - k^{-1}/(k-1) + 2/(k+1)$. Dropping the term $-\frac{1}{k(k-1)}$, it is sufficient to show that $\frac{k^{-1/3}}{k-1} + \frac{2}{k+1} \leq 3k^{-1}$ or equivalently, $\frac{k^{2/3}}{k-1} + \frac{2k}{k+1} \leq 3$. The inequality holds for $k = 2$ and $k = 3$. For $k \geq 4$, we have $\frac{2k}{k+1} = 2(1 - \frac{1}{k+1}) \leq 2$ and $h(k) := \frac{k^{2/3}}{k-1} = \frac{1}{k^{1/3}-k^{-2/3}}$ is decreasing over $k > 1$ and $h(4) < 1$, which concludes the proof for the first claim. For the second claim, we have for any $k \geq 2$:

$$B_k = \beta_{k-1}^2 + \sum_{i=1}^{k-1}\big(\Pi_{j=i}^{k-1}(1-\beta_j)\big)^2\beta_{i-1}^2 = \beta_{k-1}^2 + (1-\beta_{k-1})^2 B_{k-1} = (1 - k^{-2/3})^2 B_{k-1} + k^{-4/3} \ .$$

For $k = 1$, we have $B_1 = \beta_0^2 = 1 \leq 1$. For $k \geq 2$, suppose the claim holds for $k-1$. Then we have $B_{k-1} \leq (k-1)^{-2/3}$ and we need to show that: $B_k \leq (1 - k^{-2/3})^2(k-1)^{-2/3} + k^{-4/3} \leq k^{-2/3}$, or equivalently, $(1 - k^{-2/3})\big(k/(k-1)\big)^{2/3} \leq 1$. Note that $\big(k/(k-1)\big)^{2/3} = (1 + \frac{1}{k-1})^{2/3} \leq 1 + \frac{2}{3(k-1)}$, it is sufficient to show that: $(1 - k^{-2/3})\big(1 + \frac{2}{3(k-1)}\big) \leq 1$. Expanding the left hand side, we need to prove that $\frac{2}{3} \leq \frac{k-1}{k^{2/3}} + \frac{2}{3k^{2/3}}$. This holds for $k = 2$. For $k \geq 3$, since $h(k) := \frac{k-1}{k^{2/3}}$ is increasing over $k \geq 1$ and $h(3) > \frac{2}{3}$, the inequality holds and the claim follows. $\square$

**Lemma C.4.** *Let $S_k = \sum_{i=1}^{k-1} \frac{i+1}{i^{1/3}}$ for $k \geq 2$. Then it holds that $S_k \leq 0.6k^{5/3} + 1.5k^{2/3} - 2.1 \leq 2.1k^{5/3}$.*

*Proof.* Let $h(i) := \frac{i+1}{i^{1/3}} = i^{2/3} + i^{-1/3}$. We have $h'(i) = \frac{2}{3}i^{-1/3} - \frac{1}{3}i^{-4/3} = \frac{1}{3}i^{-4/3}(2i-1) > 0$ for any $i \geq 1$. Thus, $h$ is increasing over $[1, +\infty)$ and we have: $h(i) \leq \int_i^{i+1} h(i)di$. Summing up from $i = 1$ to $i = k-1$, we get:

$$S_k = \sum_{i=1}^{k-1} h(i) \leq \sum_{i=1}^{k-1} \int_i^{i+1} h(i)di = \int_1^k i^{2/3} + i^{-1/3}di = \frac{3}{5}(k^{5/3} - 1) + \frac{3}{2}(k^{2/3} - 1) \ . \qquad \square$$

**Lemma C.5** ((Golub & van Loan, 2013, Algorithm 6.4.3)). *Let $\mathbf{X}, \mathbf{Y} \in \mathbb{R}^{m \times r}$ be two matrices with orthonormal columns. Then it holds that:*

$$\mathrm{dist}(\mathbf{X}, \mathbf{Y}) = \max\{\sin\theta_i : i = 1, \ldots, r\} \ ,$$

*where $\theta_1, \ldots, \theta_r$ are the principal angles between the subspaces spanned by the columns of $\mathbf{X}$ and $\mathbf{Y}$.*

**Lemma C.6** (Weyl's inequality (Horn & Johnson, 2013, Chapter 7)). *Let* $\mathbf{A}, \mathbf{B} \in \mathbb{R}^{m \times n}$ *be two matrices and let* $q = \min(m, n)$. *Let* $\sigma_1(\mathbf{A}) \geq \ldots \geq \sigma_q(\mathbf{A})$ *and* $\sigma_1(\mathbf{B}) \geq \ldots \geq \sigma_q(\mathbf{B})$ *denote the singular values of* $\mathbf{A}$ *and* $\mathbf{B}$, *respectively. Then for any indices* $1 \leq i, j \leq q$ *and* $i + j \leq q + 1$, *we have:*

$$\sigma_{i+j-1}(\mathbf{A} + \mathbf{B}) \leq \sigma_i(\mathbf{A}) + \sigma_j(\mathbf{B}) .$$

*Consequently, for any* $1 \leq i \leq q$, *it holds that:*

$$|\sigma_i(\mathbf{A} + \mathbf{B}) - \sigma_i(\mathbf{A})| \leq \sigma_1(\mathbf{B}) = \|\mathbf{B}\|_2, .$$

**Lemma C.7.** *Let* $\mathbf{A}, \mathbf{B} \in \mathbb{R}^{m \times m}$ *be two symmetric matrices and suppose* $\mathbf{A}$ *is positive semidefinite. Then it holds that:* $\mathrm{trace}(\mathbf{AB}) \leq \mathrm{trace}(\mathbf{A})\lambda_{\max}(\mathbf{B})$.

*Proof.* Let $\mathbf{A} = \sum_{i=1}^{m} \lambda_i(\mathbf{A})\mathbf{u}_i\mathbf{u}_i^T$ be the eigenvalue decomposition of $\mathbf{A}$. Then we have:

$$\mathrm{trace}(\mathbf{AB}) = \sum_{i=1}^{n} \lambda_i(\mathbf{A})\,\mathrm{trace}(\mathbf{u}_i\mathbf{u}_i^T\mathbf{B}) = \sum_{i=1}^{n} \lambda_i(\mathbf{A})\mathbf{u}_i^T\mathbf{B}\mathbf{u}_i \leq \sum_{i=1}^{n} \lambda_i(\mathbf{A})\lambda_{\max}(\mathbf{B}) = \mathrm{trace}(\mathbf{A})\lambda_{\max}(\mathbf{B}) . \qquad \square$$

**Lemma C.8** (Wedin $\sin\Theta$ Theorem (Wedin, 1972; Stewart & guang Sun, 1990)). *Let* $\mathbf{A}, \mathbf{E} \in \mathbb{R}^{m \times n}$ *be two matrices and let* $\hat{\mathbf{A}} = \mathbf{A} + \mathbf{E}$. *Let* $q = \min(m, n)$. *Let* $\sigma_1(\mathbf{A}) \geq \ldots \geq \sigma_q(\mathbf{A})$ *and* $\sigma_1(\hat{\mathbf{A}}) \geq \ldots \geq \sigma_q(\hat{\mathbf{A}})$ *denote the singular values of* $\mathbf{A}$ *and* $\hat{\mathbf{A}}$, *respectively. Suppose that there exists an index* $r \in [1, q)$ *such that* $\sigma_r(\mathbf{A}) - \sigma_{r+1}(\hat{\mathbf{A}}) > 0$. *Then, it holds that:*

$$\max\{\mathrm{dist}(\mathbf{U}, \hat{\mathbf{U}}), \mathrm{dist}(\mathbf{V}, \hat{\mathbf{V}})\} \leq \frac{\max\{\|\mathbf{EV}\|_2, \|\mathbf{E}^T\mathbf{U}\|_2\}}{\sigma_r(\mathbf{A}) - \sigma_{r+1}(\hat{\mathbf{A}})} \leq \frac{\|\mathbf{E}\|_2}{\sigma_r(\mathbf{A}) - \sigma_{r+1}(\hat{\mathbf{A}})} ,$$

*where* $\mathbf{U}, \hat{\mathbf{U}} \in \mathbb{R}^{m \times r}$, $\mathbf{V}, \hat{\mathbf{V}} \in \mathbb{R}^{n \times r}$ *are the matrices consisting of the top* $r$ *left and right singular vectors of* $\mathbf{A}$ *and* $\hat{\mathbf{A}}$, *respectively. Consequently, if* $\sigma_r(\mathbf{A}) > \sigma_{r+1}(\mathbf{A}) + \|\mathbf{E}\|_2$, *then by Weyl's inequality, we have:*

$$\max\{\mathrm{dist}(\mathbf{U}, \hat{\mathbf{U}}), \mathrm{dist}(\mathbf{V}, \hat{\mathbf{V}})\} \leq \frac{\|\mathbf{E}\|_2}{\sigma_r(\mathbf{A}) - \sigma_{r+1}(\mathbf{A}) - \|\mathbf{E}\|_2} .$$

**Lemma C.9.** *Let* $\mathbf{A} \in \mathbb{R}^{m \times n}$ *be a matrix. Then for any* $D > 0$, *it holds that:*

$$\underset{\|\mathbf{X}\|_2 \leq D}{\arg\min}\{\|\mathbf{X} - \mathbf{A}\|_F^2\} = \mathrm{Proj}_{\|\mathbf{X}\|_2 \leq D}(\mathbf{A}) = \mathrm{clip}_D^{\mathrm{sp}}(\mathbf{A}) .$$

*Proof.* Since the objective function is continous and the set is compact, the optimal solution exists. Let $\mathbf{A} = \mathbf{USV}^T$ be the full singular value decomposition of $\mathbf{A}$, where $\mathbf{S} \in \mathbb{R}^{m \times n}$ is a rectangular diagonal matrix with $\mathbf{S}_{ii} = \sigma_i$ for $i = 1, \ldots, q$ and $q := \min(m, n)$, and $\mathbf{U} \in \mathbb{R}^{m \times m}$, $\mathbf{V} \in \mathbb{R}^{n \times n}$ are orthogonal matrices. Then, by the unitary invariance of the Frobenius norm, we have: $\arg\min_{\|\mathbf{X}\|_2 \leq D}\{\|\mathbf{X} - \mathbf{A}\|_F^2\} = \arg\min_{\|\mathbf{X}\|_2 \leq D}\{\|\mathbf{U}^T\mathbf{XV} - \mathbf{S}\|_F^2\}$. Let $\mathbf{Y} = \mathbf{U}^T\mathbf{XV}$. Since $\|\mathbf{Y}\|_2 = \|\mathbf{X}\|_2$ and $\mathbf{X} = \mathbf{UYV}^T$, the transformation is bijective. Thus, the previous optimization problem is equivalent to $\arg\min_{\|\mathbf{Y}\|_2 \leq D}\{\|\mathbf{Y} - \mathbf{S}\|_F^2\}$. It is clear that the optimal solution is $\mathbf{Y}^\star$ is a rectangular diagonal matrix with $\mathbf{Y}_{ii}^\star = \min(\sigma_i, D)$ for $i = 1, \ldots, q$. It follows that $\mathbf{X}^\star = \mathbf{UY}^\star\mathbf{V}^T = \mathrm{clip}_D^{\mathrm{sp}}(\mathbf{A})$. Finally, since the objective is strictly convex and the feasible set is convex, the optimal solution is unique. $\qquad \square$

**Lemma C.10.** *Let* $\mathbf{G} \in \mathbb{R}^{m \times n}$ *be a fixed matrix. Let* $\mathbf{U} \in \mathbb{R}^{m \times r}$, $\mathbf{V} \in \mathbb{R}^{n \times r}$ *be two independent random matrices that satisfy Definition 4.1 with parameter* $\kappa$. *Then, it holds that:*

$$\mathbb{E}[\|\mathbf{U}^T\mathbf{GV}\|_F] \leq \frac{\kappa r}{\sqrt{mn}}\|\mathbf{G}\|_F, \quad \mathbb{E}[\|\mathbf{G}^T\mathbf{U}\|_F] \leq \sqrt{\frac{\kappa r}{m}}\|\mathbf{G}\|_F, \quad and \quad \mathbb{E}[\|\mathbf{GV}\|_F] \leq \sqrt{\frac{\kappa r}{n}}\|\mathbf{G}\|_F .$$

*Proof.* By Jensen's inequality, we have:

$$\mathbb{E}[\|\mathbf{U}^T\mathbf{GV}\|_F] \leq \sqrt{\mathbb{E}[\|\mathbf{U}^T\mathbf{GV}\|_F^2]} = \sqrt{\mathbb{E}[\mathrm{trace}(\mathbf{V}^T\mathbf{G}^T\mathbf{UU}^T\mathbf{GV})]} = \sqrt{\mathbb{E}[\mathrm{trace}(\mathbf{GVV}^T\mathbf{G}^T\mathbf{UU}^T)]}$$

$$= \sqrt{\mathrm{trace}(\mathbf{G}\,\mathbb{E}[\mathbf{VV}^T]\mathbf{G}^T\,\mathbb{E}[\mathbf{UU}^T])} \leq \sqrt{\frac{\kappa r}{n}\frac{\kappa r}{m}\,\mathrm{trace}(\mathbf{GG}^T))} = \frac{\kappa r}{\sqrt{mn}}\|\mathbf{G}\|_F ,$$

where in the last inequality, we used Lemma C.7 twice. We next show the second inequality:

$$\mathbb{E}[\|\mathbf{G}^T\mathbf{U}\|_F] \leq \sqrt{\mathbb{E}[\|\mathbf{G}^T\mathbf{U}\|_F^2]} = \sqrt{\mathbb{E}[\text{trace}(\mathbf{U}^T\mathbf{G}\mathbf{G}^T\mathbf{U})]} = \sqrt{\text{trace}(\mathbf{G}\mathbf{G}^T\,\mathbb{E}[\mathbf{U}\mathbf{U}^T])}$$

$$\leq \sqrt{\frac{\kappa r}{m}\,\text{trace}(\mathbf{G}\mathbf{G}^T)} = \sqrt{\frac{\kappa r}{m}}\|\mathbf{G}\|_F\;.$$

The third inequality can be proved similarly. $\square$

**Lemma C.11.** *Let* $\mathbf{U}, \mathbf{V} \in \mathbb{R}^{m \times r}$ *be two matrices with orthonormal columns. Consider any decomposition of the form:* $\mathbf{V} = \mathbf{U}\mathbf{C} + \mathbf{U}^\perp\mathbf{S}$ *where* $\mathbf{C} = \mathbf{U}^T\mathbf{V}$, $\mathbf{S} \in \mathbb{R}^{r \times r}$ *and* $\mathbf{U}^\perp \in \mathbb{R}^{m \times r}$ *has orthonormal columns and satisfies* $(\mathbf{U}^\perp)^T\mathbf{U} = 0$. *It holds that:* $\|\mathbf{S}\|_2 = \text{dist}(\mathbf{U}, \mathbf{V})$ *and* $\mathbf{C}^T\mathbf{C} + \mathbf{S}^T\mathbf{S} = \mathbf{I}_r$.

*Proof.* The decomposition always exists since $\mathbf{V} = \mathbf{I}\mathbf{V} = \mathbf{U}\mathbf{U}^T\mathbf{V} + (\mathbf{I} - \mathbf{U}\mathbf{U}^T)\mathbf{V}$. Let $\mathbf{C} = \mathbf{P}\cos(\Theta)\mathbf{Q}^T$ be the SVD of $\mathbf{C}$ where $\cos(\Theta) := \text{diag}(\cos(\theta_1), \ldots, \cos(\theta_r))$. By Definition B.3, $\{\theta_i\}_{i=1}^r$ are the principal angles between the subspaces spanned by $\mathbf{U}$ and $\mathbf{V}$. Let $\hat{\mathbf{U}} := \mathbf{U}\mathbf{P}$ and $\hat{\mathbf{V}} := \mathbf{V}\mathbf{Q}$. It holds that $\hat{\mathbf{U}}^T\hat{\mathbf{V}} = \cos(\Theta)$. Let $\mathbf{R} = \hat{\mathbf{V}} - \hat{\mathbf{U}}\cos(\Theta)$. Then $\mathbf{R}^T\mathbf{R} = \sin^2(\Theta)$ and thus there exists $\hat{\mathbf{U}}^\perp \in \mathbb{R}^{m \times r}$ such that:

$$\hat{\mathbf{V}} = \hat{\mathbf{U}}\cos(\Theta) + \hat{\mathbf{U}}^\perp\sin(\Theta)\;,$$

where the $i$-th column $\hat{\mathbf{U}}_i^\perp = \frac{\mathbf{R}_i}{\sin(\theta_i)}$ and $\sin(\Theta) := \text{diag}(\sin(\theta_1), \ldots, \sin(\theta_r))$. It follows that:

$$\|\mathbf{S}\|_2 = \|\mathbf{U}^\perp\mathbf{S}\|_2 = \|\mathbf{V} - \mathbf{U}\mathbf{C}\|_2 = \|\mathbf{V} - \mathbf{U}\mathbf{P}\cos(\Theta)\mathbf{Q}^T\|_2 = \|\mathbf{V}\mathbf{Q} - \mathbf{U}\mathbf{P}\cos(\Theta)\|_2 = \|\hat{\mathbf{V}} - \hat{\mathbf{U}}\cos(\Theta)\|_2$$

$$= \|\hat{\mathbf{U}}^\perp\sin(\Theta)\|_2 = \|\sin(\Theta)\|_2 = \max_i\{\sin(\theta_i)\} = \text{dist}(\mathbf{U}, \mathbf{V})\;,$$

where the last equality follows from Lemma C.5. For the second claim, we have:

$$\mathbf{I}_r = \mathbf{V}^T\mathbf{V} = (\mathbf{U}\mathbf{C} + \mathbf{U}^\perp\mathbf{S})^T(\mathbf{U}\mathbf{C} + \mathbf{U}^\perp\mathbf{S}) = \mathbf{C}^T\mathbf{C} + \mathbf{S}^T\mathbf{S}\;. \qquad \square$$

**Lemma C.12** (Chebyshev's sum inequality)**.** *Let* $f : \mathbb{R} \to \mathbb{R}$ *and* $h : \mathbb{R} \to \mathbb{R}$ *be two functions and let* $X \in \mathbb{R}$ *be a random variable. If* $f$ *is non-increasing and* $h$ *is non-decreasing, then it holds that:*

$$\mathbb{E}[f(X)h(X)] \leq \mathbb{E}[f(X)]\,\mathbb{E}[h(X)]\;.$$

*Proof.* Let $X$ and $Y$ be two independent random variables with the same distribution. Then we always have $Q(X, Y) = (f(X) - f(Y))(h(X) - h(Y)) \leq 0$ by the assumptions on $f$ and $h$. It follows that: $\mathbb{E}[Q(X, Y)] \leq 0$. Substituting the definition of $Q(X, Y)$ and using

$$\mathbb{E}[f(X)h(Y)] = \mathbb{E}[f(Y)h(X)] = \mathbb{E}[f(X)]\,\mathbb{E}[h(X)] \quad \text{and} \quad \mathbb{E}[f(Y)h(Y)] = \mathbb{E}[f(X)h(X)]\;,$$

we get the claim. $\square$

**Lemma C.13.** *Let* $f(x) = \frac{a^2 + x}{\sqrt{2x + a^2 + b^2}}$ *where* $0 < a \leq b$. *Then* $f$ *is concave over* $x \in [-ab, ab]$.

*Proof.* Indeed, the first and second derivatives of $f$ are:

$$f'(x) = \frac{x + b^2}{(2x + a^2 + b^2)^{3/2}}\;, \quad \text{and} \quad f''(x) = \frac{a^2 - 2b^2 - x}{(2x + a^2 + b^2)^{5/2}}\;.$$

For any $x \in [-ab, ab]$, we have $a^2 - 2b^2 - x \leq a^2 - 2b^2 + ab \leq 0$ and thus $f''(x) \leq 0$. $\square$

**Lemma C.14.** *Let* $c(x) := \text{sign}(x)\min(|x|, c)$ *and let* $h_c(x) := \frac{x}{\sqrt{1 + x^2/c^2}}$. *For any* $c > 0$ *and* $x \in \mathbb{R}$, *it holds that:*

$$|h_c(x) - c(x)| \leq \begin{cases} \frac{|x|^3}{2c^2}, & |x| \leq c \\ \frac{c^3}{2x^2} & |x| \geq c \end{cases}\;.$$

*Proof.* W.l.o.g., let $x \geq 0$. If $x \geq c$, then $|h_c(x) - c(x)| = \left|\frac{x}{\sqrt{1 + x^2/c^2}} - c\right| = \left|\frac{1}{\sqrt{c^2/x^2 + 1}} - 1\right|c \leq \frac{c^3}{2x^2}$ where we used $1 - \frac{1}{\sqrt{1+y}} \leq \frac{y}{2}$ for any $y \in [0, 1]$. Similarly, if $x \leq c$, then we have $|h_c(x) - c(x)| = \left|\frac{1}{\sqrt{1 + x^2/c^2}} - 1\right|x \leq \frac{x^3}{2c^2}$ The maximum of the error is attained at $x = c$. $\square$

# D. Analysis of SGD under Sparse Noise with Spectral Spikes

**Lemma D.1** (Rotation alignment). *Let $\mathbf{G} \in \mathbb{R}^{m \times n}$ be a fixed matrix and let $\mathbf{g} = \mathbf{G} + \mathbf{N}$ where $\mathbf{N} = \ell \mathbf{U}_N \mathbf{V}_N^T$, $\ell > 0$, and $\mathbf{U}_N \in \mathbb{R}^{m \times r}$ and $\mathbf{V}_N \in \mathbb{R}^{n \times r}$ are matrices with orthonormal columns. Let $\mathbf{U}_g$ and $\mathbf{V}_g$ be the top $r$ left and right singular vectors of $\mathbf{g}$. Suppose $\ell > \|\mathbf{G}\|_2$, Then, it holds that:*

$$\|\mathbf{C}_U - \mathbf{C}_V\|_F \leq \frac{\|\mathbf{U}_N^T \mathbf{G}\|_F + \|\mathbf{V}_N^T \mathbf{G}^T\|_F}{2\ell - \|\mathbf{G}\|_2} ,$$

*where $\mathbf{C}_U := \mathbf{U}_N^T \mathbf{U}_g$ and $\mathbf{C}_V := \mathbf{V}_N^T \mathbf{V}_g$.*

*Proof.* By definition, we have $\mathbf{g}\mathbf{V}_g = \mathbf{U}_g \Sigma_g$ where $\Sigma_g \in \mathbb{R}^{r \times r}$ is the diagonal matrix of the top $r$ singular values of $\mathbf{g}$. Substituting $\mathbf{g} = \mathbf{G} + \ell \mathbf{U}_N \mathbf{V}_N^T$ gives:

$$(\mathbf{G} + \ell \mathbf{U}_N \mathbf{V}_N^T)\mathbf{V}_g = \mathbf{U}_g \Sigma_g .$$

Multiplying $\mathbf{U}_N^T$ on both sides gives:

$$\mathbf{U}_N^T (\mathbf{G} + \ell \mathbf{U}_N \mathbf{V}_N^T)\mathbf{V}_g = \mathbf{U}_N^T \mathbf{U}_g \Sigma_g, \quad \Rightarrow \quad \ell \mathbf{C}_V = \mathbf{C}_U \Sigma_g - \mathbf{U}_N^T \mathbf{G} \mathbf{V}_g .$$

Similarly using $\mathbf{g}^T \mathbf{U}_g = \mathbf{V}_g \Sigma_g$, we have: $\ell \mathbf{C}_U = \mathbf{C}_V \Sigma_g - \mathbf{V}_N^T \mathbf{G}^T \mathbf{U}_g$. Subtracting these two equations gives:

$$\mathbf{C}_U - \mathbf{C}_V = (\Sigma_g + \ell \mathbf{I})^{-1}(\mathbf{U}_N^T \mathbf{G} \mathbf{V}_g + \mathbf{V}_N^T \mathbf{G}^T \mathbf{U}_g) .$$

By Lemma C.6, we have $\sigma_r(\mathbf{g}) \geq \ell - \|\mathbf{G}\|_2$. It follows that:

$$\|\mathbf{C}_U - \mathbf{C}_V\|_F \leq \|(\Sigma_g + \ell \mathbf{I})^{-1}\|_2(\|\mathbf{U}_N^T \mathbf{G} \mathbf{V}_g\|_F + \|\mathbf{V}_N^T \mathbf{G}^T \mathbf{U}_g\|_F) \leq \frac{\|\mathbf{U}_N^T \mathbf{G}\|_F + \|\mathbf{V}_N^T \mathbf{G}^T\|_F}{2\ell - \|\mathbf{G}\|_2} . \qquad \square$$

## D.1. Proof of Lemma 4.2

*Proof.* Let $\mathbf{R} = \mathbf{g} - \tilde{\mathbf{g}}$ be the residual matrix after spectral clipping. By Lemma C.6, we have: $\sigma_{r+1}(\mathbf{g}) \leq \sigma_1(\mathbf{G}) + \sigma_{r+1}(\mathbf{N}) = \|\mathbf{G}\|_2 \leq c$. Therefore, by the definition of spectral clipping, we have $\mathbf{R} = \mathbf{U}_g \mathbf{D}(\mathbf{V}_g)^T$ , where $\mathbf{D}$ is a diagonal matrix with element $\mathbf{D}_{ii} = \max(\sigma_i(\mathbf{g}), c) - c$ and $\mathbf{U}_g$ and $\mathbf{V}_g$ are the top left and right $r$ singular vectors of $\mathbf{g}$. It follows that:

$$\mathbb{E}_{\mathbf{N}}[\langle \mathbf{G}, \tilde{\mathbf{g}} \rangle] = \mathbb{E}_{\mathbf{N}}[\langle \mathbf{G}, \mathbf{g} - \mathbf{R} \rangle] = \|\mathbf{G}\|_F^2 - \mathbb{E}_{\mathbf{N}}\left[\langle \mathbf{G}, \mathbf{U}_g \mathbf{D}(\mathbf{V}_g)^T \rangle\right] .$$

We next upper bound the second error term. By Lemma C.6, we have for any $i \in [r]$, $\sigma_i(\mathbf{g}) \geq \sigma_i(\mathbf{N}) - \sigma_1(\mathbf{G}) = \ell - \|\mathbf{G}\|_2$ and $\sigma_i(\mathbf{g}) \leq \ell + \|\mathbf{G}\|_2$. If $c \geq \ell + \|\mathbf{G}\|_2$, then $\max(\sigma_i(\mathbf{g}), c) - c = 0$ and thus $\mathbf{D}$ is zero. In what follows, we consider the case when $c \leq \ell + \|\mathbf{G}\|_2$. Using $\max(\sigma_i(\mathbf{g}), c) - c = \ell - c + \max(\sigma_i(\mathbf{g}), c) - \ell$, we have:

$$\begin{aligned}
&\mathbb{E}_{\mathbf{N}}\left[\langle \mathbf{G}, \mathbf{U}_g \mathbf{D}(\mathbf{V}_g)^T \rangle\right] \\
&= (\ell - c)\, \mathbb{E}_{\mathbf{N}}\left[\langle \mathbf{G}, \mathbf{U}_g(\mathbf{V}_g)^T \rangle\right] + \mathbb{E}_{\mathbf{N}}\left[\langle \mathbf{U}_g^T \mathbf{G} \mathbf{V}_g, \tilde{\mathbf{D}} \rangle\right] \\
&\leq (\ell - c)\, \mathbb{E}_{\mathbf{N}}\left[\langle \mathbf{G}, \mathbf{U}_g(\mathbf{V}_g)^T \rangle\right] + \mathbb{E}_{\mathbf{N}}\left[\|\mathbf{U}_g^T \mathbf{G} \mathbf{V}_g\|_F \|\tilde{\mathbf{D}}\|_F\right] .
\end{aligned}$$

where $\tilde{\mathbf{D}}$ is a diagonal matrix with $\tilde{\mathbf{D}}_{ii} = \max(\sigma_i(\mathbf{g}), c) - \ell$. We first bound $|\max(\sigma_i(\mathbf{g}), c) - \ell|$ for any $i \in [r]$. If $\ell - \|\mathbf{G}\|_2 \geq c$, then $\max(\sigma_i(\mathbf{g}), c) = \sigma_i(\mathbf{g})$ since $\sigma_i(\mathbf{g}) \geq \ell - \|\mathbf{G}\|_2 \geq c$. We then have $|\sigma_i(\mathbf{g}) - \ell| \leq \|\mathbf{G}\|_2$. If $\ell - \|\mathbf{G}\|_2 \leq c \leq \ell + \|\mathbf{G}\|_2$ and $\sigma_i(\mathbf{g}) \geq c$, then it holds that $|\sigma_i(\mathbf{g}) - \ell| \leq \|\mathbf{G}\|_2$. It remains to consider the case when $\ell - \|\mathbf{G}\|_2 \leq c \leq \ell + \|\mathbf{G}\|_2$ and $\sigma_i(\mathbf{g}) < c$. Note that $|\max(\sigma_i(\mathbf{g}), c) - \ell| = |c - \ell| \leq \|\mathbf{G}\|_2$. Therefore, in all cases, we have for any $i \in [r]$, $|\max(\sigma_i(\mathbf{g}), c) - \ell| \leq \|\mathbf{G}\|_2$ and thus $\|\tilde{\mathbf{D}}\|_F \leq \sqrt{r}\|\mathbf{G}\|_2$. We next study the terms involving $\mathbf{U}_g$ and $\mathbf{V}_g$. Since $\mathbf{U}_g, \mathbf{U}_N, \mathbf{V}_g$ and $\mathbf{V}_N$ are matrices with orthonormal columns, according to Lemma C.11, we can decompose them as:

$$\begin{cases}
\mathbf{U}_g = \mathbf{U}_N \mathbf{C}_U + \mathbf{U}_N^\perp \mathbf{S}_U, & \mathbf{C}_U = \mathbf{U}_N^T \mathbf{U}_g, \ \mathbf{U}_N^\perp \in \mathbb{R}^{m \times r}, \ (\mathbf{U}_N^\perp)^T \mathbf{U}_N = \mathbf{0}, \\
\mathbf{V}_g = \mathbf{V}_N \mathbf{C}_V + \mathbf{V}_N^\perp \mathbf{S}_V, & \mathbf{C}_V = \mathbf{V}_N^T \mathbf{V}_g, \ \mathbf{V}_N^\perp \in \mathbb{R}^{n \times r}, \ (\mathbf{V}_N^\perp)^T \mathbf{V}_N = \mathbf{0},
\end{cases} \tag{D.1}$$

where $\|\mathbf{S}_U\|_2 = \text{dist}(\mathbf{U}_N, \mathbf{U}_g)$ and $\|\mathbf{S}_V\|_2 = \text{dist}(\mathbf{V}_N, \mathbf{V}_g)$. Applying Lemma C.5 and Lemma C.8 (with $\mathbf{A} = \mathbf{N}$, $\mathbf{E} = \mathbf{G}$, $\hat{\mathbf{A}} = \mathbf{g}$), we have:

$$\max\{\|\mathbf{S}_U\|_2, \|\mathbf{S}_V\|_2\} = \max\{\text{dist}(\mathbf{U}_N, \mathbf{U}_{\mathbf{g}}), \text{dist}(\mathbf{V}_N, \mathbf{V}_{\mathbf{g}})\} \leq \frac{\|\mathbf{G}\|_2}{\sigma_1(\mathbf{N}) - \sigma_{r+1}(\mathbf{N}) - \|\mathbf{G}\|_2} = \frac{\|\mathbf{G}\|_2}{\ell - \|\mathbf{G}\|_2} . \quad \text{(D.2)}$$

Substituting the above decompositions into $\langle \mathbf{G}, \mathbf{U}_g(\mathbf{V}_g)^T\rangle$ and using $\mathbb{E}_{\mathbf{N}}[\langle \mathbf{G}, \mathbf{U}_N\mathbf{V}_N^T\rangle] = 0$, we get:

$$\mathbb{E}_{\mathbf{N}}[\langle \mathbf{G}, \mathbf{U}_g(\mathbf{V}_g)^T\rangle]$$
$$= \mathbb{E}_{\mathbf{N}}[\langle \mathbf{G}, (\mathbf{U}_N\mathbf{C}_U + \mathbf{U}_N^\perp\mathbf{S}_U)(\mathbf{V}_N\mathbf{C}_V + \mathbf{V}_N^\perp\mathbf{S}_V)^T\rangle]$$
$$= \mathbb{E}_{\mathbf{N}}[\langle \mathbf{G}, \mathbf{U}_N(\mathbf{I} - \mathbf{C}_U\mathbf{C}_V^T)\mathbf{V}_N^T\rangle] + \mathbb{E}_{\mathbf{N}}[\langle \mathbf{G}, \mathbf{U}_N\mathbf{C}_U\mathbf{S}_V^T(\mathbf{V}_N^\perp)^T\rangle] + \mathbb{E}_{\mathbf{N}}[\langle \mathbf{G}, \mathbf{U}_N^\perp\mathbf{S}_U\mathbf{C}_V^T\mathbf{V}_N^T\rangle]$$
$$\quad + \mathbb{E}_{\mathbf{N}}[\langle \mathbf{G}, \mathbf{U}_N^\perp\mathbf{S}_U\mathbf{S}_V^T(\mathbf{V}_N^\perp)^T\rangle]$$
$$\leq \mathbb{E}_{\mathbf{N}}[\|\mathbf{U}_N^T\mathbf{G}\mathbf{V}_N\|_F\|\mathbf{I} - \mathbf{C}_U\mathbf{C}_V^T\|_F] + \mathbb{E}_{\mathbf{N}}[\sqrt{r}\|\mathbf{U}_N^T\mathbf{G}\|_F\|\mathbf{S}_V\|_2\|\mathbf{V}_N^\perp\|_2\|\mathbf{C}_U\|_2]$$
$$\quad + \mathbb{E}_{\mathbf{N}}[\sqrt{r}\|\mathbf{G}\mathbf{V}_N\|_F\|\mathbf{S}_U\|_2\|\mathbf{U}_N^\perp\|_2\|\mathbf{C}_V\|_2] + \mathbb{E}_{\mathbf{N}}[\sqrt{r}\|\mathbf{G}\|_F\|\mathbf{S}_U\|_2\|\mathbf{S}_V\|_2\|\mathbf{U}_N^\perp\|_2\|\mathbf{V}_N^\perp\|_2]$$
$$\leq \mathbb{E}_{\mathbf{N}}[\|\mathbf{U}_N^T\mathbf{G}\mathbf{V}_N\|_F\|\mathbf{I} - \mathbf{C}_U\mathbf{C}_V^T\|_F] + \frac{\sqrt{r}\|\mathbf{G}\|_2}{\ell - \|\mathbf{G}\|_2}\Big(\mathbb{E}_{\mathbf{N}}[\|\mathbf{U}_N^T\mathbf{G}\|_F] + \mathbb{E}_{\mathbf{N}}[\|\mathbf{G}\mathbf{V}_N\|_F]\Big) + \frac{\sqrt{r}\|\mathbf{G}\|_2^2}{(\ell - \|\mathbf{G}\|_2)^2}\|\mathbf{G}\|_F .$$

Let $\mathbf{C}_U = \mathbf{P}_U\cos(\Theta_U)\mathbf{Q}_U^T$ be the SVD of $\mathbf{C}_U$ where $\cos(\Theta_U) := \text{diag}(\cos(\theta_{U,1}), \ldots, \cos(\theta_{U,r}))$ and $\{\theta_{U,i}\}$ are the principal angles between the subspaces spanned by $\mathbf{U}_N$ and $\mathbf{U}_g$ (according to Definition B.3). Applying Lemma D.1, we have:

$$\|\mathbf{I} - \mathbf{C}_U\mathbf{C}_V^T\|_F = \|\mathbf{I} - \mathbf{C}_U\mathbf{C}_U^T + \mathbf{C}_U(\mathbf{C}_U^T - \mathbf{C}_V^T)\|_F$$
$$\leq \|\mathbf{I} - \cos^2(\Theta_U)\|_F + \|\mathbf{C}_U(\mathbf{C}_U^T - \mathbf{C}_V^T)\|_F$$
$$\leq \sqrt{r}\max_i\{\sin^2\theta_{U,i}\} + \|\mathbf{C}_U\|_2\|\mathbf{C}_U - \mathbf{C}_V\|_F$$
$$\leq \sqrt{r}\frac{\|\mathbf{G}\|_2^2}{(\ell - \|\mathbf{G}\|_2)^2} + \frac{\|\mathbf{U}_N^T\mathbf{G}\|_F + \|\mathbf{V}_N^T\mathbf{G}^T\|_F}{2\ell - \|\mathbf{G}\|_2} \leq \frac{\sqrt{r}\|\mathbf{G}\|_2^2}{(\ell - \|\mathbf{G}\|_2)^2} + \frac{2\|\mathbf{G}\|_F}{2\ell - \|\mathbf{G}\|_2} .$$

It follows that:

$$\mathbb{E}_{\mathbf{N}}[\langle \mathbf{G}, \mathbf{U}_g(\mathbf{V}_g)^T\rangle]$$
$$\leq \Big(\frac{\sqrt{r}\|\mathbf{G}\|_2^2}{(\ell - \|\mathbf{G}\|_2)^2} + \frac{2\|\mathbf{G}\|_F}{2\ell - \|\mathbf{G}\|_2}\Big)\mathbb{E}_{\mathbf{N}}[\|\mathbf{U}_N^T\mathbf{G}\mathbf{V}_N\|_F] + \frac{\sqrt{r}\|\mathbf{G}\|_2}{\ell - \|\mathbf{G}\|_2}\Big(\mathbb{E}_{\mathbf{N}}[\|\mathbf{U}_N^T\mathbf{G}\|_F] + \mathbb{E}_{\mathbf{N}}[\|\mathbf{G}\mathbf{V}_N\|_F]\Big)$$
$$\quad + \frac{\sqrt{r}\|\mathbf{G}\|_2^2}{(\ell - \|\mathbf{G}\|_2)^2}\|\mathbf{G}\|_F .$$

Applying Lemma C.10, we have:

$$\mathbb{E}_{\mathbf{N}}[\langle \mathbf{G}, \mathbf{U}_g(\mathbf{V}_g)^T\rangle]$$
$$\leq \Big(\frac{\sqrt{r}\|\mathbf{G}\|_2^2}{(\ell - \|\mathbf{G}\|_2)^2} + \frac{2\|\mathbf{G}\|_F}{2\ell - \|\mathbf{G}\|_2}\Big)\frac{\kappa r}{\sqrt{mn}}\|\mathbf{G}\|_F + \frac{2\sqrt{r}\|\mathbf{G}\|_2}{\ell - \|\mathbf{G}\|_2}\sqrt{\frac{\kappa r}{q}}\|\mathbf{G}\|_F + \frac{\sqrt{r}\|\mathbf{G}\|_2^2}{(\ell - \|\mathbf{G}\|_2)^2}\|\mathbf{G}\|_F .$$

We next bound $\mathbb{E}_{\mathbf{N}}[\|\mathbf{U}_g^T\mathbf{G}\mathbf{V}_g\|_F]$ in a similar way:

$$\mathbb{E}_{\mathbf{N}}[\|\mathbf{U}_g^T\mathbf{G}\mathbf{V}_g\|_F] \leq \mathbb{E}_{\mathbf{N}}[\|\mathbf{C}_U^T\mathbf{U}_N^T\mathbf{G}\mathbf{V}_N\mathbf{C}_V\|_F] + \mathbb{E}_{\mathbf{N}}[\|\mathbf{C}_U^T\mathbf{U}_N^T\mathbf{G}\mathbf{V}_N^\perp\mathbf{S}_V^T\|_F]$$
$$\quad + \mathbb{E}_{\mathbf{N}}[\|\mathbf{S}_U^T(\mathbf{U}_N^\perp)^T\mathbf{G}\mathbf{V}_N\mathbf{C}_V\|_F] + \mathbb{E}_{\mathbf{N}}[\|\mathbf{S}_U^T(\mathbf{U}_N^\perp)^T\mathbf{G}\mathbf{V}_N^\perp\mathbf{S}_V^T\|_F]$$
$$\leq \mathbb{E}_{\mathbf{N}}[\|\mathbf{U}_N^T\mathbf{G}\mathbf{V}_N\|_F] + \frac{\|\mathbf{G}\|_2}{\ell - \|\mathbf{G}\|_2}\Big(\mathbb{E}_{\mathbf{N}}[\|\mathbf{U}_N^T\mathbf{G}\|_F] + \mathbb{E}_{\mathbf{N}}[\|\mathbf{G}\mathbf{V}_N\|_F]\Big) + \frac{\|\mathbf{G}\|_2^2}{(\ell - \|\mathbf{G}\|_2)^2}\|\mathbf{G}\|_F$$
$$\leq \frac{\kappa r}{\sqrt{mn}}\|\mathbf{G}\|_F + \frac{2\sqrt{\kappa r/q}\|\mathbf{G}\|_2}{\ell - \|\mathbf{G}\|_2}\|\mathbf{G}\|_F + \frac{\|\mathbf{G}\|_2^2}{(\ell - \|\mathbf{G}\|_2)^2}\|\mathbf{G}\|_F .$$

Substituting the above two bounds back gives:

$$\mathbb{E}_{\mathbf{N}}\big[\langle\mathbf{G},\mathbf{U}_g\mathbf{D}(\mathbf{V}_g)^T\rangle\big]$$

$$\leq (\ell-c)\Bigg\{\Bigg(\frac{\sqrt{r}\|\mathbf{G}\|_2^2}{(\ell-\|\mathbf{G}\|_2)^2}+\frac{2\|\mathbf{G}\|_F}{2\ell-\|\mathbf{G}\|_2}\Bigg)\frac{\kappa r}{q}\|\mathbf{G}\|_F+\frac{2\sqrt{r}\|\mathbf{G}\|_2}{\ell-\|\mathbf{G}\|_2}\sqrt{\frac{\kappa r}{q}}\|\mathbf{G}\|_F+\frac{\sqrt{r}\|\mathbf{G}\|_2^2}{(\ell-\|\mathbf{G}\|_2)^2}\|\mathbf{G}\|_F\Bigg\}$$

$$+\sqrt{r}\|\mathbf{G}\|_2\Bigg\{\frac{\kappa r}{q}\|\mathbf{G}\|_F+\frac{2\sqrt{\kappa r/q}\|\mathbf{G}\|_2}{\ell-\|\mathbf{G}\|_2}\|\mathbf{G}\|_F+\frac{\|\mathbf{G}\|_2^2}{(\ell-\|\mathbf{G}\|_2)^2}\|\mathbf{G}\|_F\Bigg\}$$

$$\leq\Bigg(\frac{r^{3/2}\kappa\|\mathbf{G}\|_2}{(\ell-\|\mathbf{G}\|_2)q}+\frac{\kappa r}{q}+\frac{2r\sqrt{\kappa}}{\sqrt{q}}+\frac{\sqrt{r}\|\mathbf{G}\|_2}{\ell-\|\mathbf{G}\|_2}+\frac{\kappa r^{3/2}}{q}+\frac{2\sqrt{\kappa}r\|\mathbf{G}\|_2}{(\ell-\|\mathbf{G}\|_2)\sqrt{q}}+\frac{\sqrt{r}\|\mathbf{G}\|_2^2}{(\ell-\|\mathbf{G}\|_2)^2}\Bigg)\|\mathbf{G}\|_F^2\;.$$

Using the assumption that $\ell\geq 9\sqrt{r}\|\mathbf{G}\|_2$ and $25r^2\kappa\leq q$, we get:

$$\frac{r^{3/2}\kappa\|\mathbf{G}\|_2}{(\ell-\|\mathbf{G}\|_2)q}\leq\frac{r\kappa}{8q}\leq\frac{1}{200},\;\frac{\kappa r}{q}+\frac{2r\sqrt{k}}{\sqrt{q}}\leq\frac{1}{25}+\frac{2}{5}=\frac{11}{25},\;\frac{\sqrt{r}\|\mathbf{G}\|_2}{\ell-\|\mathbf{G}\|_2}\leq\frac{1}{8},$$

$$\frac{2\sqrt{\kappa}r\|\mathbf{G}\|_2}{(\ell-\|\mathbf{G}\|_2)\sqrt{q}}\leq\frac{\sqrt{\kappa r}}{4\sqrt{q}}\leq\frac{1}{20},\;\frac{\sqrt{r}\|\mathbf{G}\|_2^2}{(\ell-\|\mathbf{G}\|_2)^2}\leq\frac{1}{64}\;,\text{and }\mathbb{E}_{\mathbf{N}}\big[\langle\mathbf{G},\mathbf{U}_g\mathbf{D}(\mathbf{V}_g)^T\rangle\big]\leq\frac{2}{3}\|\mathbf{G}\|_F^2\;.$$

For the second claim, recall that $\sigma_{r+1}(\mathbf{g})\leq c$ and $\sigma_i(\mathbf{g})\leq\ell+\|\mathbf{G}\|_2$ for $i\in[r]$. It follows that:

$$\|\tilde{\mathbf{g}}\|_F^2=\sum_{i=1}^r\min(c,\sigma_i(\mathbf{g}))^2+\sum_{i=r+1}^q\sigma_i^2(\mathbf{g})$$

$$\leq r\min(c,\ell+\|\mathbf{G}\|_2)^2+\|\mathbf{g}\|_F^2-\sum_{i=1}^r\sigma_i^2(\mathbf{g})$$

$$= r\min(c,\ell+\|\mathbf{G}\|_2)^2+\|\mathbf{G}\|_F^2+\|\mathbf{N}\|_F^2+2\langle\mathbf{G},\mathbf{N}\rangle-\sum_{i=1}^r\sigma_i^2(\mathbf{g})\;.$$

Taking expectation with respect to $\mathbf{N}$ gives:

$$\mathbb{E}_{\mathbf{N}}[\|\tilde{\mathbf{g}}\|_F^2]\leq r\min(c,\ell+\|\mathbf{G}\|_2)^2+\|\mathbf{G}\|_F^2+r\ell^2-\mathbb{E}_{\mathbf{N}}\Big[\sum_{i=1}^r\sigma_i^2(\mathbf{g})\Big]\;.$$

Meanwhile, note that

$$\sum_{i=1}^r\sigma_i^2(\mathbf{g})=\max_{\mathbf{V}^T\mathbf{V}=\mathbf{I}_r}\|\mathbf{g}\mathbf{V}\|_F^2\geq\|\mathbf{g}\mathbf{V}_N\|_F^2=\|\mathbf{G}\mathbf{V}_N+\ell\mathbf{U}_N\mathbf{V}_N^T\mathbf{V}_N\|_F^2=\|\mathbf{G}\mathbf{V}_N\|_F^2+2\ell\langle\mathbf{G}\mathbf{V}_N,\mathbf{U}_N\rangle+r\ell^2\;.$$

We thus have $\mathbb{E}_{\mathbf{N}}\big[\sum_{i=1}^r\sigma_i^2(\mathbf{g})\big]\geq\mathbb{E}_{\mathbf{N}}[\|\mathbf{G}\mathbf{V}_N\|_F^2]+r\ell^2\geq r\ell^2$. Substituting it into the previous display gives the second claim.

$$\square$$

### D.2. Proof of Theorem 4.6

*Proof.* According to Assumption 3.3, we have for any $k\geq 0$,

$$f(\mathbf{X}_{k+1})\overset{\text{(B.1)}}{\leq} f(\mathbf{X}_k)-\eta\langle\nabla f(\mathbf{X}_k),\mathrm{clip}_c^{\mathrm{sp}}(\mathbf{g}(\mathbf{X}_k))\rangle+\frac{L_F\eta^2}{2}\|\mathrm{clip}_c^{\mathrm{sp}}(\mathbf{g}(\mathbf{X}_k))\|_F^2\;.$$

We first consider the case when $\ell\geq 9\sqrt{r}M$. Since $c\geq M\geq\|\nabla f(\mathbf{X}_k)\|_2$, we can apply Lemma 4.2 with $\mathbf{G}=\nabla f(\mathbf{X}_k)$ and obtain:

$$\mathbb{E}_{\mathbf{N}_k}[\langle\nabla f(\mathbf{X}_k),\mathrm{clip}_c^{\mathrm{sp}}(\mathbf{g}(\mathbf{X}_k))\rangle]\geq\frac{1}{3}\|\nabla f(\mathbf{X}_k)\|_F^2\quad\text{and}\quad\mathbb{E}_{\mathbf{N}_k}[\|\mathrm{clip}_c^{\mathrm{sp}}(\mathbf{g}(\mathbf{X}_k))\|_F^2]\leq rc^2+\|\nabla f(\mathbf{X}_k)\|_F^2\;,$$

where $\mathbf{N}_k$ denotes the noise at iteration $k$. We next consider the case when $\ell \leq 9\sqrt{r}M$. By lemma C.6, we have $\sigma_1(\mathbf{g}(\mathbf{X}_k)) \leq \ell + \|\nabla f(\mathbf{X}_k)\|_2 \leq 10\sqrt{r}M$. We thus have $\text{clip}_c^{\text{sp}}(\mathbf{g}(\mathbf{X}_k)) = \mathbf{g}(\mathbf{X}_k)$ since $c \geq 10\sqrt{r}M$. It follows that:

$$\mathbb{E}_{\mathbf{N}_k}[\langle \nabla f(\mathbf{X}_k), \text{clip}_c^{\text{sp}}(\mathbf{g}(\mathbf{X}_k))\rangle] = \|\nabla f(\mathbf{X}_k)\|_F^2 \quad \text{and} \quad \mathbb{E}_{\mathbf{N}_k}[\|\text{clip}_c^{\text{sp}}(\mathbf{g}(\mathbf{X}_k))\|_F^2] \leq r\ell^2 + \|\nabla f(\mathbf{X}_k)\|_F^2 \ ,$$

since $\text{clip}_c^{\text{sp}}(\mathbf{g}(\mathbf{X}_k)) = \mathbf{g}(\mathbf{X}_k)$. Combining both cases and taking expectation with respect to $\mathbf{N}_k$ on both sides of the first display, we have for any $k \geq 0$:

$$\mathbb{E}_{\mathbf{N}_k}[f(\mathbf{X}_{k+1})] \leq f(\mathbf{X}_k) - \frac{\eta}{3}\|\nabla f(\mathbf{X}_k)\|_F^2 + \frac{L_F\eta^2}{2}(r\min(c, 10\ell/9)^2 + \|\nabla f(\mathbf{X}_k)\|_F^2) \ .$$

Using $\frac{L_F\eta^2}{2} \leq \frac{\eta}{6}$, taking the full expectation and rearranging, we obtain:

$$\frac{\eta}{6}\mathbb{E}[\|\nabla f(\mathbf{X}_k)\|_F^2] \leq \mathbb{E}[f(\mathbf{X}_k)] - \mathbb{E}[f(\mathbf{X}_{k+1})] + \frac{L_F r\eta^2 \min(c, 10\ell/9)^2}{2} \ .$$

Summing up from $k = 0$ to $K - 1$ and dividing both sides by $K$, we have:

$$\frac{1}{K}\sum_{k=0}^{K-1}\mathbb{E}[\|\nabla f(\mathbf{X}_k)\|_F^2] \leq \frac{6F^0}{\eta K} + 3L_F r\eta \min(10\sqrt{r}M, 10\ell/9)^2 \ .$$

To get the final iteration complexity, it is sufficient to let each term of the right-hand side be at most $\frac{\epsilon^2}{2}$ and substituting $\frac{1}{\eta} = \frac{6rL_F \min(10\sqrt{r}M, 10\ell/9)^2}{\epsilon^2} + 3L_F$.

$\square$

## D.3. Proof of Lemma 4.3

*Proof.* When $\|\mathbf{G}\|_F = 0$, the claim holds clearly. We assume $\|\mathbf{G}\|_F > 0$ in what follows. Let us first consider the case when $c \leq \sqrt{r}(\ell - \|\mathbf{G}\|_2)$. By Lemma C.6, we have

$$\|\mathbf{g}\|_F \geq \sqrt{\sum_{i=1}^{r}\sigma_i(\mathbf{g})^2} \geq \sqrt{\sum_{i=1}^{r}(\sigma_i(\mathbf{N}) - \sigma_i(\mathbf{G}))^2} \geq \sqrt{r}(\ell - \|\mathbf{G}\|_2) \geq c \ .$$

Therefore, the global clipping is always active and we have $\tilde{\mathbf{g}} = \frac{c}{\|\mathbf{g}\|_F}\mathbf{g}$. It follows that:

$$\mathbb{E}_{\mathbf{N}}[\langle \mathbf{G}, \tilde{\mathbf{g}}\rangle] = \mathbb{E}_{\mathbf{N}}\left[\frac{c}{\|\mathbf{G} + \mathbf{N}\|_F}\langle \mathbf{G}, \mathbf{G} + \mathbf{N}\rangle\right] = \mathbb{E}_{\mathbf{N}}\left[\frac{c(G^2 + Z_{\mathbf{N}})}{\sqrt{G^2 + 2Z_{\mathbf{N}} + r\ell^2}}\right] := \mathbb{E}_{\mathbf{N}}[h(Z_{\mathbf{N}})] \ ,$$

where $G := \|\mathbf{G}\|_F$ and $Z_{\mathbf{N}} := \langle \mathbf{G}, \mathbf{N}\rangle$. Applying Lemma C.13 (with $a = G$ and $b = \sqrt{r}\ell$), $h$ is concave in $Z_{\mathbf{N}}$ since $\sqrt{r}\ell \geq G$. We thus have:

$$\mathbb{E}_{\mathbf{N}}[h(Z_{\mathbf{N}})] \leq h(\mathbb{E}_{\mathbf{N}}[Z_{\mathbf{N}}]) = h(0) = \frac{cG^2}{\sqrt{G^2 + r\ell^2}} \leq \frac{c}{\sqrt{r}\ell}\|\mathbf{G}\|_F^2 \ .$$

We next prove the lower bound. Using taylor expansion, we have:

$$h(Z_{\mathbf{N}}) = h(0) + h'(0)Z_{\mathbf{N}} + \frac{1}{2}h''(\xi)Z_{\mathbf{N}}^2 \ ,$$

where $\xi \in [-\sqrt{r}G\ell, \sqrt{r}G\ell]$. It follows that:

$$\mathbb{E}_{\mathbf{N}}[h(Z_{\mathbf{N}})] = h(0) + h'(0)\mathbb{E}_{\mathbf{N}}[Z_{\mathbf{N}}] + \frac{1}{2}\mathbb{E}_{\mathbf{N}}[h''(\xi)Z_{\mathbf{N}}^2] \geq h(0) - \max_{\xi \in [-\sqrt{r}G\ell, \sqrt{r}G\ell]}[h''(\xi)]\frac{1}{2}\mathbb{E}_{\mathbf{N}}[Z_{\mathbf{N}}^2] \ .$$

Applying Lemma C.13, we have:

$$\max_{\xi \in [-\sqrt{r}G\ell, \sqrt{r}G\ell]}|h''(\xi)| \leq \max_{\xi \in [-\sqrt{r}G\ell, \sqrt{r}G\ell]}c\frac{G^2 + 2r\ell^2 + |\xi|}{(G^2 + r\ell^2 - 2|\xi|)^{5/2}} \leq c\frac{\|\mathbf{G}\|_F^2 + 2r\ell^2 + \sqrt{r}\|\mathbf{G}\|_F\ell}{(r\ell^2 + \|\mathbf{G}\|_F^2 - 2\sqrt{r}\|\mathbf{G}\|_F\ell)^{5/2}} \ .$$

By Lemma C.10, we obtain:

$$\mathbb{E}_{\mathbf{N}}[Z_{\mathbf{N}}^2] = \ell^2 \, \mathbb{E}_{\mathbf{N}}[\langle \mathbf{G}, \mathbf{U}_N \mathbf{V}_N^T \rangle^2] \leq \ell^2 \, \mathbb{E}_{\mathbf{N}}[\|\mathbf{U}_N^T \mathbf{G} \mathbf{V}_N\|_F^2 \|I_r\|_F^2] \leq r^2 \ell^2 \frac{\kappa^2 r^2 \|\mathbf{G}\|_F^2}{mn} \leq \frac{\ell^2}{25^2} \|\mathbf{G}\|_F^2 \, .$$

Combining the above inequalities yields:

$$
\begin{aligned}
\mathbb{E}_{\mathbf{N}}[h(Z_{\mathbf{N}})] &\geq \frac{c\|\mathbf{G}\|_F^2}{\sqrt{r\ell^2 + \|\mathbf{G}\|_F^2}} - c\frac{\|\mathbf{G}\|_F^2 + 2r\ell^2 + \sqrt{r}\|\mathbf{G}\|_F \ell}{(r\ell^2 + \|\mathbf{G}\|_F^2 - 2\sqrt{r\ell}\|\mathbf{G}\|_F)^{5/2}} \frac{\ell^2}{2 \cdot 25^2} \|\mathbf{G}\|_F^2 \\
&= \frac{c\|\mathbf{G}\|_F^2}{\sqrt{r\ell^2 + \|\mathbf{G}\|_F^2}} \left( 1 - \frac{(\|\mathbf{G}\|_F^2 + 2r\ell^2 + \sqrt{r}\|\mathbf{G}\|_F \ell)\ell^2 \sqrt{r\ell^2 + \|\mathbf{G}\|_F^2}}{2 \cdot 25^2(r\ell^2 + \|\mathbf{G}\|_F^2 - 2\sqrt{r\ell}\|\mathbf{G}\|_F)^{5/2}} \right) \\
&\geq \frac{c\|\mathbf{G}\|_F^2}{\sqrt{r\ell^2 + \|\mathbf{G}\|_F^2}} \left( 1 - \frac{(1/9 + 2 + 1/3)r\ell^4 \sqrt{10/9 r\ell^2}}{2 \cdot 25^2 (1/3)^{5/2} r^{2.5}\ell^5} \right) \\
&\geq \frac{9}{10} \frac{c\|\mathbf{G}\|_F^2}{\sqrt{r\ell^2 + \|\mathbf{G}\|_F^2}} \geq \frac{4c}{5\sqrt{r\ell}} \|\mathbf{G}\|_F^2 \, .
\end{aligned}
$$

Finally, since clipping is always active, we have: $\mathbb{E}_{\mathbf{N}}[\|\tilde{\mathbf{g}}\|_F^2] = \mathbb{E}_{\mathbf{N}}\left[\frac{c^2}{\|\mathbf{g}\|_F^2}\|\mathbf{g}\|_F^2\right] = c^2$. We next consider the case when $\sqrt{r}(\ell - \|\mathbf{G}\|_2) \leq c \leq \ell + \|\mathbf{G}\|_2$. Let the clipping factor $\alpha_{\mathbf{N}} := \min(1, \frac{c}{\|\mathbf{g}\|_F})$. The upper bound of the inner product can be derived as follows:

$$\mathbb{E}_{\mathbf{N}}[\langle \mathbf{G}, \tilde{\mathbf{g}} \rangle] = \mathbb{E}_{\mathbf{N}}[\alpha_{\mathbf{N}}\langle \mathbf{G}, \mathbf{g} \rangle] = \mathbb{E}_{\mathbf{N}}[\alpha_{\mathbf{N}}]\|\mathbf{G}\|_F^2 + \mathbb{E}_{\mathbf{N}}[\alpha_{\mathbf{N}}\langle \mathbf{G}, \mathbf{N} \rangle] \leq \|\mathbf{G}\|_F^2 + \mathbb{E}_{\mathbf{N}}[\alpha_{\mathbf{N}}\langle \mathbf{G}, \mathbf{N} \rangle] \, .$$

Since $\|\mathbf{g}\|_F = \sqrt{\|\mathbf{G}\|_F^2 + r\ell^2 + 2\langle \mathbf{G}, \mathbf{N} \rangle}$ is increasing in $\langle \mathbf{G}, \mathbf{N} \rangle$, $\alpha_{\mathbf{N}}$ is non-increasing in $\langle \mathbf{G}, \mathbf{N} \rangle$. Using Lemma C.12, we obtain:

$$\mathbb{E}_{\mathbf{N}}[\langle \mathbf{G}, \tilde{\mathbf{g}} \rangle] \leq \|\mathbf{G}\|_F^2 + \mathbb{E}_{\mathbf{N}}[\alpha_{\mathbf{N}}]\, \mathbb{E}_{\mathbf{N}}[\langle \mathbf{G}, \mathbf{N} \rangle] = \|\mathbf{G}\|_F^2 \, .$$

Next recall that $\|\mathbf{g}\|_F \geq \sqrt{r}(\ell - \|\mathbf{G}\|_2)$ and $c \geq \sqrt{r}(\ell - \|\mathbf{G}\|_2)$, we get:

$$c^2 \geq \mathbb{E}_{\mathbf{N}}[\|\tilde{\mathbf{g}}\|_F^2] = \mathbb{E}_{\mathbf{N}}[\alpha_{\mathbf{N}}^2 \|\mathbf{g}\|_F^2] = \mathbb{E}_{\mathbf{N}}[\min(c^2, \|\mathbf{g}\|_F^2)] \geq r(\ell - \|\mathbf{G}\|_2)^2 \geq \frac{4r\ell^2}{9} \, .$$

Finally, if $c \geq \ell + \|\mathbf{G}\|_2$, then clipping is never active and we have $\tilde{\mathbf{g}} = \mathbf{g}$ and the claims follows. $\qquad\square$

### D.4. Proof of Theorem 4.7

*Proof.* According to Assumption 3.3, we have for any $k \geq 0$,

$$f(\mathbf{X}_{k+1}) \overset{(B.1)}{\leq} f(\mathbf{X}_k) - \eta\langle \nabla f(\mathbf{X}_k), \mathrm{clip}_c^{\mathrm{g}}(\mathbf{g}(\mathbf{X}_k)) \rangle + \frac{L_F \eta^2}{2}\|\mathrm{clip}_c^{\mathrm{g}}(\mathbf{g}(\mathbf{X}_k))\|_F^2 \, .$$

According to Lemma 4.3 (with $\mathbf{G} = \nabla f(\mathbf{X}_k)$), if $c \geq \sqrt{r}(\ell - \|\nabla f(\mathbf{X}_k)\|_2)$, then $\mathbb{E}_{\mathbf{N}_k}[\langle \nabla f(\mathbf{X}_k), \mathrm{clip}_c^{\mathrm{g}}(\mathbf{g}(\mathbf{X}_k)) \rangle] \lesssim \|\nabla f(\mathbf{X}_k)\|_F^2$ and $\mathbb{E}_{\mathbf{N}_k}[\|\mathrm{clip}_c^{\mathrm{g}}(\mathbf{g}(\mathbf{X}_k))\|_F^2] \gtrsim r\ell^2$, where $\mathbf{N}_k$ denotes the noise at iteration $k$. The bounds are not better than the ones for vanilla SGD without clipping. Let us next consider the case when $c \leq \frac{2}{3}\sqrt{r}\ell \leq \sqrt{r}(\ell - \|\nabla f(\mathbf{X}_k)\|_2)$. We obtain:

$$\mathbb{E}_{\mathbf{N}_k}[\langle \nabla f(\mathbf{X}_k), \mathrm{clip}_c^{\mathrm{g}}(\mathbf{g}(\mathbf{X}_k)) \rangle] \geq \frac{4c}{5\sqrt{r\ell}}\|\nabla f(\mathbf{X}_k)\|_F^2 \quad \text{and} \quad \mathbb{E}_{\mathbf{N}_k}[\|\mathrm{clip}_c^{\mathrm{g}}(\mathbf{g}(\mathbf{X}_k))\|_F^2] = c^2 \, .$$

Taking the expectation with respect to $\mathbf{N}_k$ on both sides of the first display and substituting the above two bounds, we have for any $k \geq 0$:

$$\mathbb{E}_{\mathbf{N}_k}[f(\mathbf{X}_{k+1})] \leq f(\mathbf{X}_k) - \eta\frac{4c}{5\sqrt{r\ell}}\|\nabla f(\mathbf{X}_k)\|_F^2 + \frac{L_F \eta^2}{2}c^2 \, .$$

Rearranging and taking the full expectation gives:

$$\eta\frac{4c}{5\sqrt{r\ell}}\mathbb{E}[\|\nabla f(\mathbf{X}_k)\|_F^2] \leq \mathbb{E}[f(\mathbf{X}_k)] - \mathbb{E}[f(\mathbf{X}_{k+1})] + \frac{L_F \eta^2}{2}c^2 \, .$$

Summing up from $k = 0$ to $K - 1$ and dividing both sides by $K$, we have:

$$\frac{1}{K} \sum_{k=0}^{K-1} \mathbb{E}[\|\nabla f(\mathbf{X}_k)\|_F^2] \leq \frac{5\sqrt{r}\ell F^0}{4c\eta K} + \frac{5\sqrt{r}\ell L_F \eta c}{8} = 5\sqrt{r}\ell \Big(\frac{F^0}{4c\eta K} + \frac{L_F \eta c}{8}\Big) .$$

Let $\frac{F^0}{4c\eta K} = \frac{L_F \eta c}{8}$. We have $\eta = \sqrt{\frac{2F^0}{L_F K c^2}}$ and

$$\frac{1}{K} \sum_{k=0}^{K-1} \mathbb{E}[\|\nabla f(\mathbf{X}_k)\|_F^2] \leq \frac{5\sqrt{r}\ell}{2} \sqrt{\frac{L_F F^0}{2K}} \leq \epsilon^2 .$$

Rearranging concludes the proof. □

## E. Analysis of Reformulation as Composite Frank-Wolfe

### E.1. Proof of Proposition 3.1

*Proof.* According to Lemma C.9, we have:

$$\mathbf{V}_{k+1} = \underset{\mathbf{X} \in Q_2}{\arg\min} \Big\{ \langle \mathbf{M}_k, \mathbf{X} \rangle + \frac{\lambda}{2\alpha} \|\mathbf{X}\|_F^2 \Big\}$$

$$= \underset{\mathbf{X} \in Q_2}{\arg\min} \Big\{ \frac{\lambda}{2\alpha} \|\mathbf{X} + \frac{\alpha}{\lambda} \mathbf{M}_k\|_F^2 \Big\} = \text{proj}_{Q_2} \Big\{ -\frac{\alpha}{\lambda} \mathbf{M}_k \Big\}$$

$$= \text{clip}_{D_2}^{\text{sp}} \Big( -\frac{\alpha}{\lambda} \mathbf{M}_k \Big) = -\frac{\alpha}{\lambda} \text{clip}_{\frac{\lambda D_2}{\alpha}}^{\text{sp}} (\mathbf{M}_k) .$$

Substituting $\mathbf{V}_{k+1}$ and the choices of $\gamma_k$ and $c_k$ into the equation of $\mathbf{X}_{k+1}$ gives the desired update. □

**Lemma E.1.** *Let CFW (5) be applied to problem (3) under Assumption 3.2 and 3.3. Then for any $k \geq 1$, it holds that:*

$$\mathbb{E}[\|\mathbf{E}_k\|_F] \leq \Pi_{i=0}^{k-1}(1 - \beta_i)\|\mathbf{E}_0\|_F + L D_F \sum_{i=1}^{k} \big(\Pi_{j=i}^{k-1}(1 - \beta_j)\big)\gamma_{i-1} + \sqrt{\sum_{i=1}^{k} \big(\Pi_{j=i}^{k-1}(1 - \beta_j)\big)^2 \beta_{i-1}^2} \sigma .$$

*where $\mathbf{E}_k := \mathbf{M}_k - \nabla f(\mathbf{X}_k)$ and $D_F := \sup_{\mathbf{X}, \mathbf{Y} \in Q}\{\|\mathbf{X} - \mathbf{Y}\|_F\}$.*

*Proof.* The proof is standard. By the definition of $\mathbf{M}_{i+1}$, we have for any $i \geq 0$:

$$\mathbf{E}_{i+1} = (1 - \beta_i)\mathbf{E}_i + \beta_i \big(\mathbf{g}(\mathbf{X}_i) - \nabla f(\mathbf{X}_i)\big) + \nabla f(\mathbf{X}_i) - \nabla f(\mathbf{X}_{i+1}) .$$

Applying Lemma C.2, we have for any $k \geq 1$:

$$\mathbf{E}_k = \Pi_{i=0}^{k-1}(1-\beta_i)\mathbf{E}_0 + \sum_{i=1}^{k} \big(\Pi_{j=i}^{k-1}(1-\beta_j)\big)\big(\nabla f(\mathbf{X}_{i-1}) - \nabla f(\mathbf{X}_i)\big) + \sum_{i=1}^{k} \big(\Pi_{j=i}^{k-1}(1-\beta_j)\big)\beta_{i-1}\big(\mathbf{g}(\mathbf{X}_{i-1}) - \nabla f(\mathbf{X}_{i-1})\big) .$$

Taking the norm on both sides, we get:

$$\|\mathbf{E}_k\|_F \leq \Pi_{i=0}^{k-1}(1 - \beta_i)\|\mathbf{E}_0\|_F + \sum_{i=1}^{k} \big(\Pi_{j=i}^{k-1}(1 - \beta_j)\big)\|\nabla f(\mathbf{X}_{i-1}) - \nabla f(\mathbf{X}_i)\|_F$$

$$+ \Big\|\sum_{i=1}^{k} \big(\Pi_{j=i}^{k-1}(1 - \beta_j)\big)\beta_{i-1}\big(\mathbf{g}(\mathbf{X}_{i-1}) - \nabla f(\mathbf{X}_{i-1})\big)\Big\|_F .$$

Using Assumption 3.3, $\|\mathbf{X}_i - \mathbf{X}_{i-1}\|_F = \gamma_{i-1}\|\mathbf{V}_i - \mathbf{X}_{i-1}\|_F \leq \gamma_{i-1}D_F$, taking expectation on both sides and applying the Jensen inequality, we have for any $k \geq 1$:

$$\mathbb{E}[\|\mathbf{E}_k\|_F] \leq \Pi_{i=0}^{k-1}(1-\beta_i)\|\mathbf{E}_0\|_F + L_F D_F \sum_{i=1}^{k}\big(\Pi_{j=i}^{k-1}(1-\beta_j)\big)\gamma_{i-1}$$

$$+ \sqrt{\mathbb{E}\Big[\Big\|\sum_{i=1}^{k}\big(\Pi_{j=i}^{k-1}(1-\beta_j)\big)\beta_{i-1}\big(\mathbf{g}(\mathbf{X}_{i-1}) - \nabla f(\mathbf{X}_{i-1})\big)\Big\|_F^2\Big]}$$

$$\leq \Pi_{i=0}^{k-1}(1-\beta_i)\|\mathbf{E}_0\|_F + L_F D_F \sum_{i=1}^{k}\big(\Pi_{j=i}^{k-1}(1-\beta_j)\big)\gamma_{i-1} + \sqrt{\sum_{i=1}^{k}\big(\Pi_{j=i}^{k-1}(1-\beta_j)\big)^2\beta_{i-1}^2\sigma^2} \ . \qquad \square$$

**Theorem E.2.** *Let CFW* (5) *be applied to problem* (3) *under convexity of $f$ and Assumption 3.2. Suppose that for any $k \geq 0$, $\mathbf{V}_{k+1}$ satisfies:*

$$F_k(\mathbf{V}_{k+1}) \leq F_k(\mathbf{V}_{k+1}^\star) + e_{k+1} \ , \tag{E.1}$$

*where $\mathbf{V}_{k+1}^\star = \arg\min_{\mathbf{X} \in Q}\{F_k(\mathbf{X})\}$ and $e_k \leq \frac{q_1}{k} + q_2$ for some $q_1, q_2 \geq 0$. Let $\gamma_k = \frac{2}{k+2}$. If we choose $\beta_k \equiv 1$ and use a batchsize of $B$ for computing $\mathbf{g}(\mathbf{X}_k)$, then we have for any $K \geq 1$:*

$$F(\mathbf{X}_K) - F^\star \leq \frac{2(C_f + q_1)}{K+1} + q_2 + + \frac{D_F\sigma}{\sqrt{B}} \ ,$$

*where*

$$C_f := \sup_{\mathbf{X},\mathbf{S} \in Q, \gamma \in [0,1], \mathbf{Y} = \mathbf{X} + \gamma(\mathbf{S}-\mathbf{X})} \frac{2}{\gamma^2}(f(\mathbf{Y}) - f(\mathbf{X}) - \langle\nabla f(\mathbf{X}), \mathbf{Y} - \mathbf{X}\rangle) \ .$$

*If Assumption 3.3 additionally holds, then by choosing $\beta_k = \frac{1}{(k+1)^{2/3}}$ and $\mathbf{M}_0 = \mathbf{g}(\mathbf{X}_0)$ with batch size $B$, we have for any $K \geq 1$:*

$$\mathbb{E}[F(\mathbf{X}_K) - F^\star] \leq \frac{2(C_f + q_1)}{K+1} + \frac{12.4 L_F D_F^2 + 6.2 D_F\sigma/\sqrt{B}}{K^{1/3}} + q_2 \ ,$$

*where $D_F := \sup_{\mathbf{X},\mathbf{Y} \in Q}\{\|\mathbf{X} - \mathbf{Y}\|_F\}$.*

*Proof.* Let $\gamma_k = \frac{a_{k+1}}{A_{k+1}}$ where $A_{k+1} = A_k + a_{k+1}$ for $k \geq 0$ and $A_0 = 0$. Using convexity of $f$, we have:

$$A_k F(\mathbf{X}_k) + a_{k+1}F(\mathbf{X}^\star)$$
$$\geq A_k F(\mathbf{X}_k) + a_{k+1}\big(f(\mathbf{X}_k) + \langle\nabla f(\mathbf{X}_k), \mathbf{X}^\star - \mathbf{X}_k\rangle + \psi(\mathbf{X}^\star)\big)$$
$$= A_k F(\mathbf{X}_k) + a_{k+1}\big(f(\mathbf{X}_k) + \langle\mathbf{M}_k, \mathbf{X}^\star\rangle + \psi(\mathbf{X}^\star) + \langle\nabla f(\mathbf{X}_k) - \mathbf{M}_k, \mathbf{X}^\star\rangle - \langle\nabla f(\mathbf{X}_k), \mathbf{X}_k\rangle\big) \ .$$

Let $\mathbf{E}_k := \mathbf{M}_k - \nabla f(\mathbf{X}_k)$. Using $F_k(\mathbf{X}^\star) \geq F_k(\mathbf{V}^\star) \geq F_k(\mathbf{V}_{k+1}) - e_{k+1}$, we get:

$$A_k F(\mathbf{X}_k) + a_{k+1}F(\mathbf{X}^\star)$$
$$\geq A_k F(\mathbf{X}_k) + a_{k+1}\big(f(\mathbf{X}_k) + \langle\mathbf{M}_k, \mathbf{V}_{k+1}\rangle + \psi(\mathbf{V}_{k+1}) - e_{k+1} + \langle\nabla f(\mathbf{X}_k) - \mathbf{M}_k, \mathbf{X}^\star\rangle - \langle\nabla f(\mathbf{X}_k), \mathbf{X}_k\rangle\big)$$
$$= A_k F(\mathbf{X}_k) + a_{k+1}\big(f(\mathbf{X}_k) + \langle\nabla f(\mathbf{X}_k), \mathbf{V}_{k+1} - \mathbf{X}_k\rangle + \psi(\mathbf{V}_{k+1}) - e_{k+1} - \langle\mathbf{E}_k, \mathbf{X}^\star - \mathbf{V}_{k+1}\rangle\big)$$
$$= A_{k+1}f(\mathbf{X}_k) + a_{k+1}\langle\nabla f(\mathbf{X}_k), \mathbf{V}_{k+1} - \mathbf{X}_k\rangle + A_k\psi(\mathbf{X}_k) + a_{k+1}\psi(\mathbf{V}_{k+1}) - a_{k+1}e_{k+1} - a_{k+1}\langle\mathbf{E}_k, \mathbf{X}^\star - \mathbf{V}_{k+1}\rangle$$
$$\geq A_{k+1}f(\mathbf{X}_k) + A_{k+1}\langle\nabla f(\mathbf{X}_k), \mathbf{X}_{k+1} - \mathbf{X}_k\rangle + A_{k+1}\psi((1-\gamma_k)\mathbf{X}_k + \gamma_k\mathbf{V}_{k+1}) - a_{k+1}e_{k+1} - a_{k+1}D_F\|\mathbf{E}_k\|_F$$
$$= A_{k+1}F(\mathbf{X}_{k+1}) - A_{k+1}(f(\mathbf{X}_{k+1}) - f(\mathbf{X}_k) - \langle\nabla f(\mathbf{X}_k), \mathbf{X}_{k+1} - \mathbf{X}_k\rangle) - a_{k+1}e_{k+1} - a_{k+1}D_F\|\mathbf{E}_k\|_F$$
$$\geq A_{k+1}F(\mathbf{X}_{k+1}) - \frac{\gamma_k^2 A_{k+1}}{2}C_f - a_{k+1}e_{k+1} - a_{k+1}D_F\|\mathbf{E}_k\|_F \ ,$$

where the second inequality follows from the convexity of $\psi$ and the third inequality follows from the definition of $C_f$. Substracting $A_{k+1}F(\mathbf{X}^\star)$ from both sides gives:

$$A_{k+1}(F(\mathbf{X}_{k+1}) - F^\star) \leq A_k(F(\mathbf{X}_k) - F^\star) + \frac{a_{k+1}^2}{2A_{k+1}}C_f + a_{k+1}e_{k+1} + a_{k+1}D_F\|\mathbf{E}_k\|_F \ .$$

Summing up from $k = 0$ to $k = K - 1$, we have for any $K \geq 1$:

$$A_K(F(\mathbf{X}_K) - F^\star) \leq A_0(F(\mathbf{X}_0) - F^\star) + \frac{C_f}{2}\sum_{k=0}^{K-1}\frac{a_{k+1}^2}{A_{k+1}} + \sum_{k=0}^{K-1}a_{k+1}e_{k+1} + D_F\sum_{k=0}^{K-1}a_{k+1}\|\mathbf{E}_k\|_F \ .$$

Dividing both sides by $A_K$ and substituting $A_0 = 0$, we get:

$$F(\mathbf{X}_K) - F^\star \leq \frac{1}{A_K}\frac{C_f}{2}\sum_{k=0}^{K-1}\frac{a_{k+1}^2}{A_{k+1}} + \frac{1}{A_K}\sum_{k=0}^{K-1}a_{k+1}e_{k+1} + \frac{D_F}{A_K}\sum_{k=0}^{K-1}a_{k+1}\|\mathbf{E}_k\|_F \ .$$

Let $a_k = k$ for $k \geq 1$. Then we have $A_k = \frac{k(k+1)}{2}$ and $\gamma_k = \frac{a_{k+1}}{A_{k+1}} = \frac{2}{k+2}$. Consider mini-batch SGD with $\beta_k \equiv 1$ and $\mathbf{M}_k = \mathbf{g}(\mathbf{X}_k)$ using a batch size of $B$. Then $\mathbb{E}[\|\mathbf{E}_k\|_F] \leq \mathbb{E}[\|\mathbf{E}_k\|_F^2]^{1/2} \leq \frac{\sigma}{\sqrt{B}}$ for any $k \geq 0$. It follows that:

$$\mathbb{E}[F(\mathbf{X}_K) - F^\star] \leq \frac{2}{K(K+1)}\frac{C_f}{2}\sum_{k=0}^{K-1}\frac{2(k+1)}{k+2} + \frac{2}{K(K+1)}\sum_{k=0}^{K-1}(k+1)\Big(\frac{q_1}{k+1} + q_2\Big) + \frac{D_F\sigma}{\sqrt{B}}$$

$$\leq \frac{2C_f K}{K(K+1)} + \frac{2q_1 K}{K(K+1)} + \frac{2q_2}{K(K+1)}\frac{(K+1)K}{2} + \frac{D_F\sigma}{\sqrt{B}} = \frac{2(C_f + q_1)}{K+1} + q_2 + \frac{D_F\sigma}{\sqrt{B}} \ .$$

Suppose now Assumption 3.3 holds and $\beta_k = \frac{1}{(k+1)^{2/3}}$. Applying Lemma E.1 and substituting the bounds from Lemma C.3, we have for any $k \geq 1$:

$$\mathbb{E}[\|\mathbf{E}_k\|_F] \leq L_F D_F \frac{3}{k^{1/3}} + \frac{1}{k^{1/3}}\sigma = \frac{3L_F D_F + \sigma}{k^{1/3}} \ ,$$

where we used $\Pi_{i=0}^{k-1}(1 - \beta_i) = 0$ since $\beta_0 = 1$. If follows that for any $K \geq 2$,

$$\frac{D_F}{A_K}\sum_{k=0}^{K-1}a_{k+1}\,\mathbb{E}[\|\mathbf{E}_k\|_F] \leq \frac{D_F}{A_K}\sum_{k=1}^{K-1}(k+1)\frac{3L_F D_F + \sigma}{k^{1/3}} + \frac{D_F}{A_K}\mathbb{E}[\|\mathbf{E}_0\|_F] \leq \frac{3L_F D_F^2 + \sigma D_F}{A_K}2.1K^{5/3} + \frac{D_F}{A_K}\mathbb{E}[\|\mathbf{E}_0\|_F] \ ,$$

where we used Lemma C.4. (For $K = 1$, $\frac{D_F}{A_K}\sum_{k=0}^{K-1}a_{k+1}\,\mathbb{E}[\|\mathbf{E}_k\|_F] = \frac{D_F}{A_K}\mathbb{E}[\|\mathbf{E}_0\|_F]$.) Substituting the previous display and $A_K = \frac{K(K+1)}{2} \geq \frac{K^2}{2}$ into the main recurrence and using $\mathbb{E}[\|\mathbf{E}_0\|_F] = \mathbb{E}[\|\mathbf{M}_0 - \nabla f(\mathbf{X}_0)\|_F] = \mathbb{E}[\|\mathbf{g}(\mathbf{X}_0) - \nabla f(\mathbf{X}_0)\|_F] \leq \sigma/\sqrt{B}$, we conclude the proof. $\square$

### E.2. Discussion of CFW under sparse noise with spikes

Suppose the stochastic gradient is corrupted by sparse noise with spikes as described in Assumption 4.5. Recall the definition of the noise $N(\mathbf{X}) = \ell\mathbf{U}_X\mathbf{V}_X^T$ which satisfies $\mathbb{E}[\|N(\mathbf{X})\|_F^2] = r\ell^2$. According to Theorem E.2, the convergence rate of standard CFW (5) under this assumption is of order $K = \mathcal{O}(\frac{L^3 D_F^6 + D_F^3 r^{3/2}\ell^3}{\epsilon^3})$ (to reach $\mathbb{E}[F(\mathbf{X}_K) - F^\star] \leq \epsilon$). We next consider the variant of CFW (5) with pre-spectral clipping:

$$\mathbf{M}_{k+1} = (1 - \beta_k)\mathbf{M}_k + \beta_k\tilde{\mathbf{g}}(\mathbf{X}_{k+1}), \quad \tilde{\mathbf{g}}(\mathbf{X}_{k+1}) = \text{clip}_{c_k}^{\text{sp}}(\mathbf{g}(\mathbf{X}_{k+1})) \ . \tag{E.2}$$

Recall that at any point $\mathbf{X}$, $\mathbf{g}(\mathbf{X}) = \nabla f(\mathbf{X}) + \ell\mathbf{U}_X\mathbf{V}_X^T$. Intuitively, we have $\text{clip}_c^{\text{sp}}(\mathbf{g}(\mathbf{X})) \approx \nabla f(\mathbf{X}) + c\mathbf{U}_X\mathbf{V}_X^T$ for large $\ell$. The clipped stochastic gradient has noise level controlled by $c$ instead of $\ell$. On the other hand, the estimator becomes biased, i.e., $\mathbb{E}[\text{clip}_c^{\text{sp}}(\mathbf{g}(\mathbf{X}))] \neq \nabla f(\mathbf{X})$. In certain cases, however, the bias can be negligible with high probability. For instance, suppose $\nabla f(\mathbf{X})$ has rank much smaller compared to $q := \min(m, n)$. Then the injected low-rank noise component are likely orthogonal to the singular vectors of $\nabla f(\mathbf{X})$, which implies $\text{clip}_c^{\text{sp}}(\mathbf{g}(\mathbf{X}_k)) = \nabla f(\mathbf{X}) + c\mathbf{U}_X\mathbf{V}_X^T$ with high probability. In general, let $\mathbf{R} := \mathbf{g}(\mathbf{X}) - \text{clip}_c^{\text{sp}}(\mathbf{g}(\mathbf{X}))$. Suppose $\|\nabla f(\mathbf{X})\|_2 \leq c \leq \ell - \|\nabla f(\mathbf{X})\|_2$, then the bias can be written as:

$$\text{bias} = \mathbb{E}[\text{clip}_c^{\text{sp}}(\mathbf{g}(\mathbf{X}))] - \nabla f(\mathbf{X}) = -\mathbb{E}[\mathbf{R}] = -\mathbb{E}[\mathbf{U}_{\mathbf{g}}\mathbf{D}\mathbf{V}_{\mathbf{g}}^T]$$

where $\mathbf{D}$ is a diagonal matrix with element $\mathbf{D}_{ii} = \sigma_i(\mathbf{g}) - c$ and $\mathbf{U}_g$ and $\mathbf{V}_g$ are the top left and right $r$ singular vectors of $\mathbf{g}$. From the perturbation theory of singular vectors, $\mathbf{U}_g$ and $\mathbf{V}_g$ are close to $\mathbf{U}_N$ and $\mathbf{V}_N$ which have zero mean (when $\ell$ is large enough). We leave the rigorous theoretical analysis of the bias error to future work. We provide empirical performance of using momentum with and without pre-spectral clipping under spectral spike noise in Figure F.6.

---

**Algorithm 3** Matrix inverse square root via Newton-Schulz

---

1: **Input:** P.D. $\mathbf{X} \in \mathbb{R}^{m \times m}$, $\alpha \geq \sigma_{\max}(\mathbf{X}) > 0$, $K > 0$.
2: **Signature:** $\mathrm{MISR}(\mathbf{X}, \alpha, K)$.
3: Set $\hat{\mathbf{X}} = \mathbf{X}/\alpha$ and initialize $\mathbf{Y}_0 = \hat{\mathbf{X}}$ and $\mathbf{Z}_0 = \mathbf{I}$.
4: **for** $k = 0, 1, 2 \ldots, K - 1$ **do**
5:     $\mathbf{T}_k = \frac{1}{2}(3\mathbf{I} - \mathbf{Z}_k \mathbf{Y}_k)$.
6:     $\mathbf{Y}_{k+1} = \mathbf{Y}_k \mathbf{T}_k$.
7:     $\mathbf{Z}_{k+1} = \mathbf{T}_k \mathbf{Z}_k$.
8: **end for**
9: **Return:** $\mathbf{Z}_K / \alpha^{1/2}$.

---

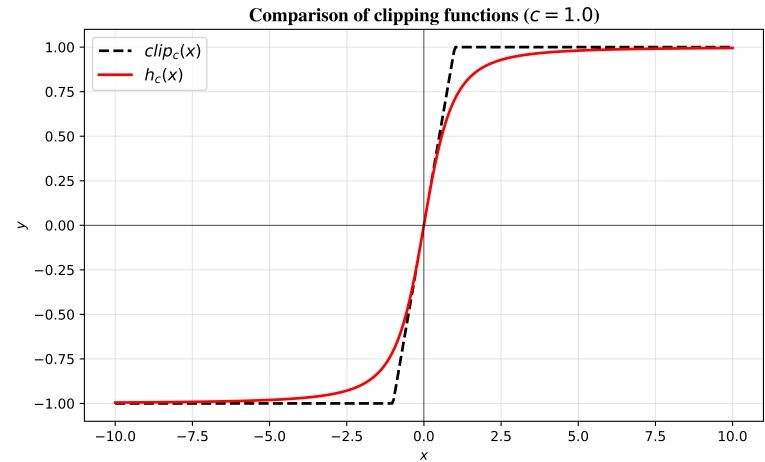

*Figure E.1.* The plot of $\mathrm{clip}_c(x) := \mathrm{sign}(x) \min(|x|, c)$ and $h_c(x) := \frac{x}{\sqrt{1+x^2/c^2}}$.

### E.3. Solution of the Subproblem with Other Regularizers

Consider subproblem (6). Suppose that $\psi$ acts only on the singular values of $\mathbf{X}$, i.e., $\psi(\mathbf{X}) = \tilde{\psi}(\sigma(\mathbf{X}))$ for some $\tilde{\psi} : \mathbb{R}^q \to \mathbb{R}$ where $\sigma(\mathbf{X})$ denotes the vector of singular values with $q := \min(m, n)$. Then $\psi$ is unitarily invariant, i.e., for any orthogonal matrices $\mathbf{U}$ and $\mathbf{V}$ of appropriate dimensions, we have $\psi(\mathbf{X}) = \psi(\mathbf{U}\mathbf{X}\mathbf{V}^T)$. Let $\mathbf{G} = \mathbf{U}_G \mathbf{S}_G \mathbf{V}_G^T$ and $\mathbf{X} = \mathbf{U}_X \mathbf{S}_X \mathbf{V}_X^T$ be the SVD of $\mathbf{G}$ and $\mathbf{X}$, respectively. Then we can first minimize $\langle \mathbf{G}, \mathbf{X} \rangle$ over the choices of $\mathbf{U}_X$ and $\mathbf{V}_X$. By Von Neumann's Trace Inequality, we have: $\langle \mathbf{G}, \mathbf{X} \rangle \geq -\sum_{i=1}^q \sigma_i(\mathbf{G})\sigma_i(\mathbf{X})$. Choosing $\mathbf{U}_X = -\mathbf{U}_G$ and $\mathbf{V}_X = \mathbf{V}_G$, we have: $\langle \mathbf{G}, \mathbf{X} \rangle = -\sum_{i=1}^q \sigma_i(\mathbf{G})\sigma_i(\mathbf{S}_X)$. Then the original problem is reduced to:

$$\min_{\mathbf{x} \in \mathbb{R}^q, \mathbf{x}_i \in [0, D_2]} \left\{ -\sum_{i=1}^q \sigma_i(\mathbf{G})\mathbf{x}_i + \tilde{\psi}(\mathbf{x}) \right\} .$$

If $\tilde{\psi}$ is separable, i.e., $\tilde{\psi}(\mathbf{x}) = \sum_{i=1}^q \psi_i(\mathbf{x}_i)$ for some $\psi_i : \mathbb{R}_{\geq 0} \to \mathbb{R}$, then the problem further decomposes into $q$ independent one-dimensional problems:

$$\sigma_i^\star = \arg\min_{x \in [0, D_2]} \{-\sigma_i(\mathbf{G}) \cdot x + \psi_i(x)\}, \quad i = 1, \ldots, q . \tag{E.3}$$

Then $\mathbf{X}^\star = -\mathbf{U}_G \mathbf{S}^\star \mathbf{V}_G^T$ where $\mathbf{S}^\star$ is the diagonal rectangular matrix with $\mathbf{S}_{ii}^\star = \sigma_i^\star$. For non-separable $\tilde{\psi}$, one can still use standard convex optimization methods to solve the reduced $q$-dimensional problem, e.g. when $\tilde{\psi}(\mathbf{x}) = \frac{b}{2}(\sum_{i=1}^q \mathbf{x}_i)^2$, which we leave as future explorations.

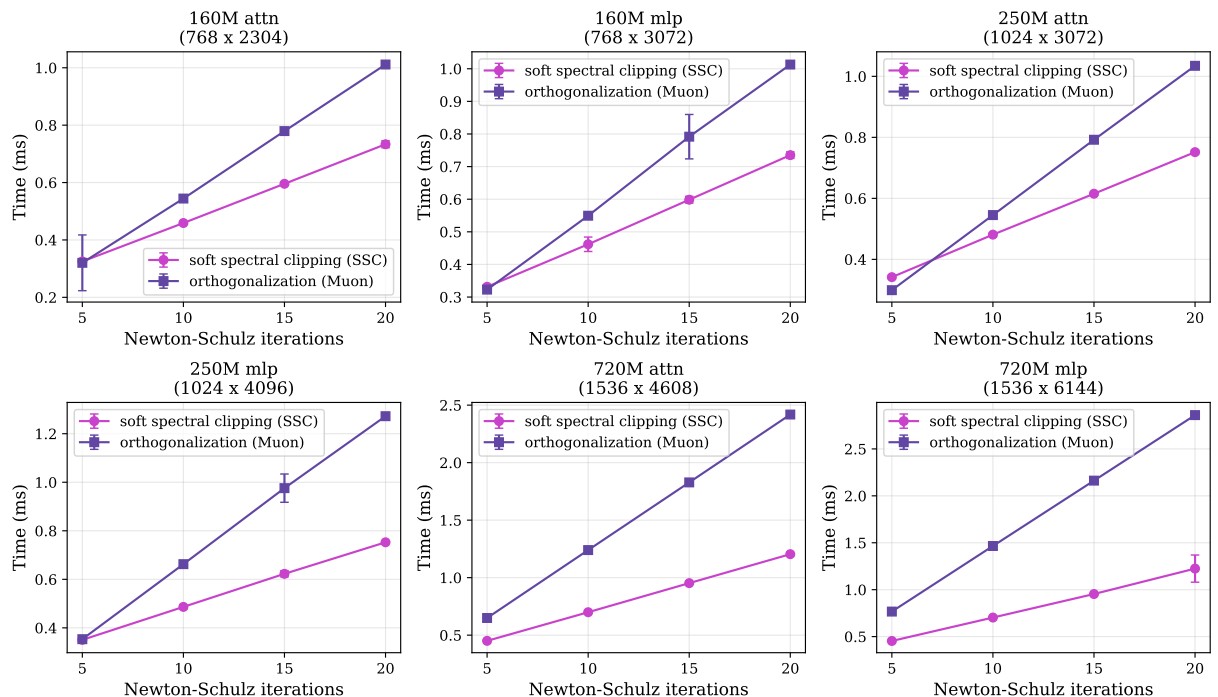

*Figure E.2.* Runtime comparison between Soft Spectral Clipping (SSC) and the orthogonalization operator on matrices with various sizes. By default, we use 10 Newton-Schulz iterations for SSC and 5 iterations for the latter for running the corresponding optimizers.

# F. Additional Experiment Results and Detailed Setup

## F.1. Synthetic experiments

We consider the binary classification task of minimizing the logistic loss: $f(\mathbf{X}) = \frac{1}{n} \sum_{i=1}^{n} \log(1 + \exp(-y_i \langle \mathbf{A}_i, \mathbf{X} \rangle))$, where $\mathbf{A}_i \in \mathbb{R}^{d \times d}$ are data matrices and $y_i \in \{-1, 1\}$ are labels from the training dataset. Let $x = \text{vec}(\mathbf{X})$ and $a_i = \text{vec}(\mathbf{A}_i)$. The gradients and the Hessians are:

$$\nabla f(\mathbf{X}) = \frac{1}{n} \sum_{i=1}^{n} \left( \frac{-y_i}{1 + \exp(y_i \langle \mathbf{A}_i, \mathbf{X} \rangle)} \right) \mathbf{A}_i, \quad \nabla^2 f(x) = \frac{1}{n} \sum_{i=1}^{n} \left( \frac{\exp(y_i a_i^T x)}{(1 + \exp(y_i a_i^T x))^2} \right) a_i a_i^T ,$$

Therefore, the loss $f$ satisfies Assumption 4.4 and 3.3 with $M \leq \frac{1}{n} \sum_{i=1}^{n} \|\mathbf{A}_i\|_2$ and $L_F \leq \frac{1}{4n} \| \sum_{i=1}^{n} a_i a_i^T \|_2 = \frac{1}{4n} \sigma_{\max}(\tilde{\mathbf{A}})^2$ where $\tilde{\mathbf{A}} \in \mathbb{R}^{n \times d^2}$ is the matrix whose rows are $a_i^T$.

### F.1.1. VALIDATION OF CONSTRAINED OPTIMIZATION WITH WEIGHT REGULARIZATION

**Data generation.** We examine a standard over-parameterized regime with label noise, a setting where unregularized optimization is typically prone to overfitting. We use $d = 10$ and generate $n = 80 < 100$ training samples and $n_{\text{test}} = 200$ test samples ($D_{\text{test}}$) as follows. First, a weight matrix $\tilde{\mathbf{X}}$ is sampled entry-wise from a standard normal distribution $\mathcal{N}(0, 1)$. The feature matrices $\mathbf{A}_i$ are similarly sampled with entries from $\mathcal{N}(0, 1)$. The labels are then generated via: $y_i = \text{sign}(\langle \mathbf{A}_i, \mathbf{X}^\star \rangle + \xi_i)$, where $\xi_i \sim \mathcal{N}(0, \sigma_{\text{noise}}^2)$. We set the noise level $\sigma_{\text{noise}} = 5.0$.

**Configuration.** Let $F(\mathbf{X}) = \min_{\mathbf{X} \in Q_2} \{ f(\mathbf{X}) + \frac{b}{2} \|\mathbf{X}\|_F^2 \}$. We evaluate the update rule (4) to verify its convergence properties on the function $F$. We vary the regularization coefficient $b \in \{0, 0.1, 1.0\}$ and $D_2 \in \{0.2, 1, 5\}$. We note that the unconstrained solution of $f(\mathbf{X})$ has spectral norm close to 1; thus, these three choices of $D_2$ correspond to the situations where the solution is outside, close to and inside the spectral ball, respectively. We adopt the same choice of hyper-parameters suggested by Theorem 3.4:

$$\frac{1}{\lambda} = D_2, \quad \alpha = \frac{\lambda}{b}, \quad c_k \equiv b D_2, \quad \lambda \eta_k = \frac{2}{k+2} .$$

**Results.** For each pair of $b$ and $D_2$, we report four metrics: 1) Function residual $F(\mathbf{X}_k) - F^\star$ where $F^\star$ is the optimal value.

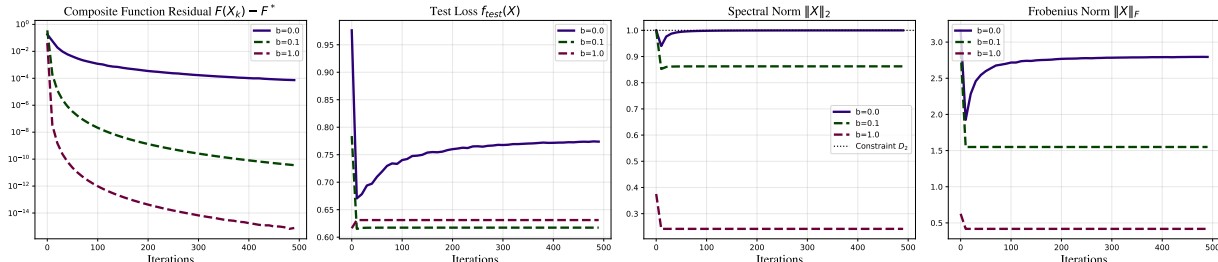

*Figure F.1.* Convergence dynamics of update rule (4) using full-batch gradient. We report the function residual of the constrained composite logistic loss $F(\mathbf{X})$ with $D_2 = 1.0$ under different $b$. The unconstrained solution of $f(\mathbf{X})$ has spectral norm close to 1.

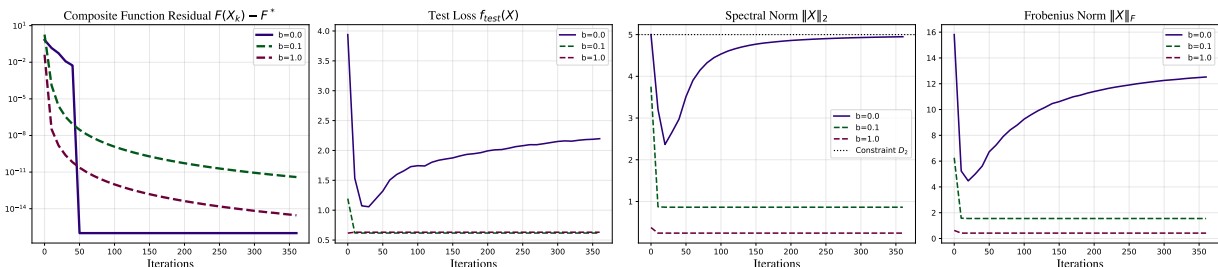

*Figure F.2.* Convergence dynamics of update rule (4) using full-batch gradient. We report the function residual of the constrained composite logistic loss $F(\mathbf{X})$ with $D_2 = 5.0$ under different $b$. The unconstrained solution of $f(\mathbf{X})$ has spectral norm close to 1.

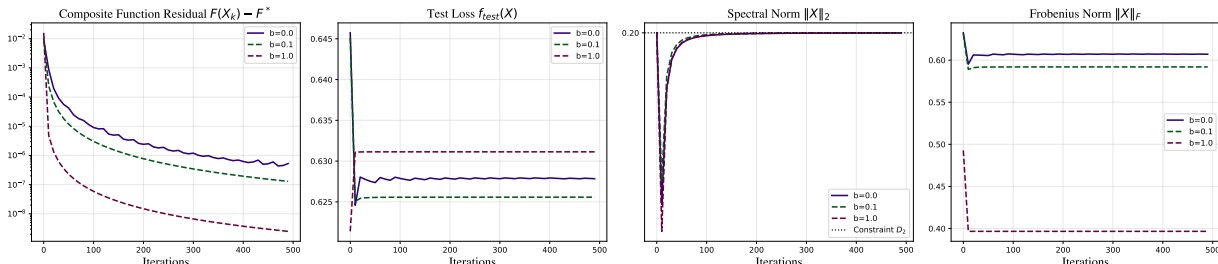

*Figure F.3.* Convergence dynamics of update rule (4) using full-batch gradient. We report the function residual of the constrained composite logistic loss $F(\mathbf{X})$ with $D_2 = 0.2$ under different $b$. The unconstrained solution of $f(\mathbf{X})$ has spectral norm close to 1.

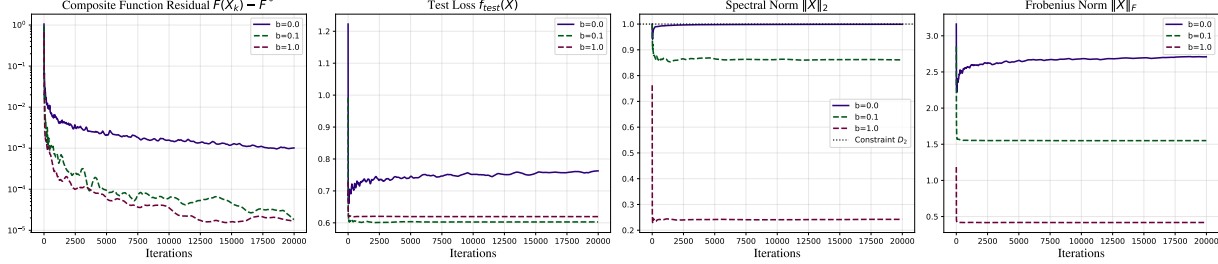

*Figure F.4.* Convergence dynamics of update rule (4) using stochastic gradient. We report the function residual of the constrained composite logistic loss $F(\mathbf{X})$ with $D_2 = 1.0$ under different $b$. The unconstrained solution of $f(\mathbf{X})$ has spectral norm close to 1.

2) Test loss: $f_{\text{test}}(\mathbf{X}_k) = \frac{1}{n_{\text{test}}} \sum_{i \in D_{\text{test}}} \log(1 + \exp(-y_i \langle \mathbf{A}_i, \mathbf{X}_k \rangle))$. 3) Spectral norm of $\mathbf{X}_k$. 4) Frobenius norm of $\mathbf{X}_k$. Figures F.1, F.2, and F.3 display results using full-batch gradient descent with $b_k \equiv 1$, while Figure F.4 utilizes a batch size of 20 and $\beta_k = \frac{1}{(k+1)^{2/3}}$. These figures demonstrate that update rule (4) exactly minimizes the composite function $F$ within the spectral ball of radius $D_2$ using the prescribed parameters. Furthermore, we observe that appropriate regularization ($b > 0$) improves generalization (lower test loss) compared to the unregularized case ($b = 0$), and that larger values of $b$ result in solutions with smaller Frobenius norms, consistent with theoretical expectations.

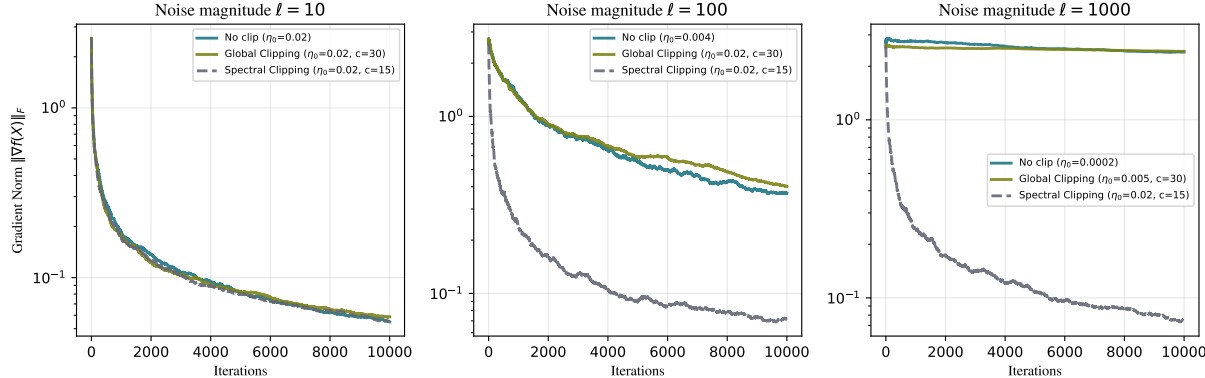

*Figure F.5.* Convergence comparison of Vanilla SGD, SGD with Global Clipping, and SGD with Spectral Clipping under low-rank spectral noise. We vary the noise spike magnitude $\ell \in \{10, 100, 1000\}$ relative to the gradient spectral norm bound $M \approx 15$.

### F.1.2. VALIDATION OF ROBUSTNESS TO SPARSE SPECTRAL SPIKES

**Data generation.** We now investigate the robustness of spectral clipping against sparse spectral noise with spikes. We consider the same matrix logistic regression as described previously with $d = 50, n = 100$. We simulate the stochastic gradient oracle of the form $\mathbf{g}(\mathbf{X}) = \nabla f(\mathbf{X}) + \mathbf{N}$ where $\mathbf{N} = \ell \mathbf{U} \mathbf{V}^T$. Here, $\ell$ represents the noise magnitude, and the matrices $\mathbf{U}, \mathbf{V} \in \mathbb{R}^{d \times r}$ are sampled independently from the uniform distribution on the Stiefel manifold. We construct these matrices using the standard polar decomposition method (Chikuse, 2003). Specifically, let $\mathbf{A} \in \mathbb{R}^{d \times r}$ be a random matrix with independent entries $\mathbf{A}_{ij} \sim \mathcal{N}(0, 1)$. We compute $\mathbf{U} = \mathbf{A}(\mathbf{A}^T \mathbf{A})^{-1/2}$ (and similarly for $\mathbf{V}$). By construction, $\mathbf{U}$ has orthonormal columns ($\mathbf{U}^T \mathbf{U} = \mathbf{I}_r$). Moreover, due to the symmetry of the standard normal distribution, we have $\mathbb{E}[\mathbf{U}] = \mathbf{0}$ (Theorem 2.2.1, (Chikuse, 2003)). We next show that $\mathbf{U}$ satisfies Definition 4.1 with $\kappa = 1$.

Let $\mathbf{P} := \mathbf{U}\mathbf{U}^T = \mathbf{A}(\mathbf{A}^T \mathbf{A})^{-1}\mathbf{A}^T$ be the projection matrix. Since $\mathbf{A}$ has rank $r$ with probability 1, we have $\mathrm{trace}(\mathbf{P}) = r$. Consider any fixed orthogonal matrix $\mathbf{Q} \in \mathbb{R}^{d \times d}$. By the rotational invariance of the Gaussian distribution, $\mathbf{Q}\mathbf{A}$ has the same distribution as $\mathbf{A}$ (Vershynin, 2018). Consequently, the projection matrix constructed from $\mathbf{Q}\mathbf{A}$, $\mathbf{Q}\mathbf{P}\mathbf{Q}^T = (\mathbf{Q}\mathbf{A})((\mathbf{Q}\mathbf{A})^T (\mathbf{Q}\mathbf{A}))^{-1}(\mathbf{Q}\mathbf{A})^T$, shares the distribution of $\mathbf{P}$. Taking the expectation yields $\mathbb{E}[\mathbf{P}] = \mathbf{Q}\mathbb{E}[\mathbf{P}]\mathbf{Q}^T$. Since this holds for any orthogonal $\mathbf{Q}$, Schur's Lemma implies that $\mathbb{E}[\mathbf{P}] = \lambda \mathbf{I}_d$ for some scalar $\lambda \geq 0$. Using the linearity of the trace, we have:

$$r = \mathbb{E}[\mathrm{trace}(\mathbf{P})] = \mathrm{trace}(\mathbb{E}[\mathbf{P}]) = \mathrm{trace}(\lambda \mathbf{I}_d) = \lambda d \implies \lambda = \frac{r}{d} \implies \mathbb{E}[\mathbf{U}\mathbf{U}^T] = \frac{r}{d}\mathbf{I}_d .$$

**Results.** For the problem instance, the gradient spectral norm is bounded by $M \approx 15$. We evaluate the algorithms under varying noise scales $\ell \in \{10, 100, 1000\}$. We compare three update rules: 1) Vanilla SGD, 2) SGD with global clipping, and 3) SGD with spectral clipping. We employ a decay schedule $\eta_k = \frac{\eta_0}{\sqrt{k+1}}$ for all methods. We tune the initial learning rate $\eta_0$ individually for each method to maximize performance. For spectral clipping, we fix the threshold $c = 15$ across all noise levels, matching the theory.

Figure F.5 illustrates the convergence behavior (training loss). In the low-noise regime ($\ell = 10$), all three methods perform comparably. As the noise scale $\ell$ increases, the performance of spectral clipping remains robust, maintaining fast convergence. In contrast, Vanilla SGD and SGD with global clipping require significantly reduced learning rates $\eta_0$ to prevent divergence, resulting in slower convergence. We observe a similar phenomenon in the momentum setting. Figure F.6 compares Momentum SGD with and without spectral clipping (applied to the raw stochastic gradient). The results confirm that spectral clipping effectively filters sparse spectral spikes, enabling the momentum buffer to accumulate valid signal directions.

### F.2. Recording Singular Value Distribution of Stochastic Gradients and Updates & Noise Structure

We conduct our analysis on a LLaMA-style transformer model with 12 layers, 12 attention heads, and embedding dimension 768 (approximately 124M parameters). The model is pretrained on the SlimPajama dataset (Soboleva et al., 2023) using a sequence length of 512 tokens and the batch size of 256. We use AdamW with the standard configuration: $\eta = 10^{-3}, \beta_1 = 0.8, \beta_2 = 0.999, \lambda = 0.1$ and we run 16000 iterations with a cosine learning rate schedule and 2000 warmup steps.

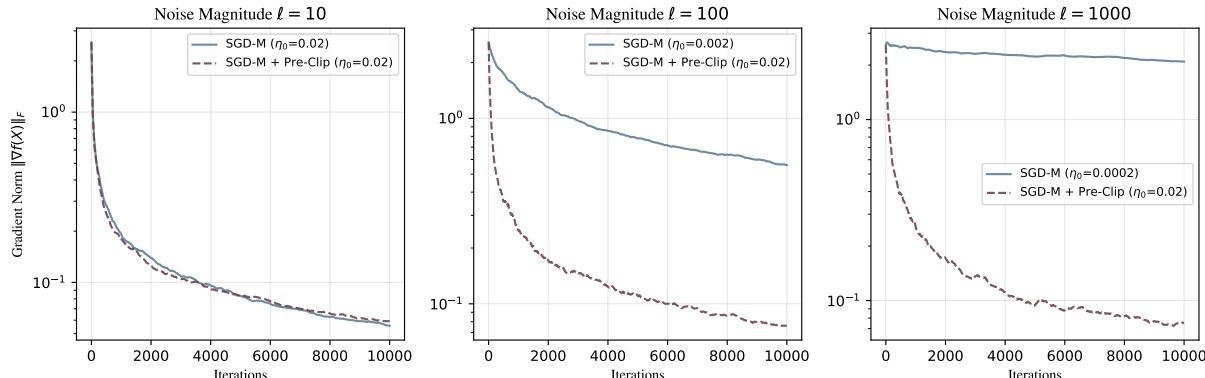

*Figure F.6.* Convergence comparison of Vanilla SGD-M and SGD-M with Spectral Clipping under low-rank spectral noise. We vary the noise spike magnitude $\ell \in \{10, 100, 1000\}$ relative to the gradient spectral norm bound $M \approx 15$.

### F.2.1. SINGULAR VALUE DISTRIBUTION OF STOCHASTIC GRADIENTS AND UPDATES.

We record the full singular value distributions of both the raw stochastic gradient $\mathbf{g}_k$ and the optimizer update direction $\mathbf{U}_k$ at selected checkpoints: $0\%$, $5\%$, $50\%$, and $99\%$ of training (corresponding to iterations 1, 800, 8000, and 15840). We record these quantities for representative weight matrices across the network depth: the token embedding matrix, the attention and MLP weights from early (layer 0), middle (layer 6), and late (layer 11) transformer blocks.

**Heavy-tailed stochastic gradient spectrum**. Figure F.9 illustrates the singular value distributions of the raw stochastic gradients and the ratio of $\sigma_{\max}/\sigma_{\min}$ for each weight matrix. We observe the a small number of singular values are orders of magnitude larger than the rest, leading to high condition numbers. This phenomenon persists throughout the training trajectory and is ubiquitous across the recorded layers.

**Comparisons of update norms.** As shown in Figure F.10, the AdamW update shows a more uniform spectral distribution compared to the raw gradient, suggesting that coordinate-wise normalization partially mitigates the spectral imbalance. However, they fail to impose a hard ceiling. In contrast, Figure F.12 demonstrates that SPECTRA-AdamW maintains a rigorously controlled bound on the spectral norm and exhibits a balanced spectral distribution. (The comparisons between Signum and Spectra-Signum can be found in Figure F.11 and F.13.)

### F.2.2. NOISE STRUCTURE ANALYSIS.

We next investigate the structure of stochastic gradient noise. Let $\mathbf{G} \in \mathbb{R}^{m \times n}$ denote the 'true' gradient signal, approximated by averaging gradients over a large batch of samples. Let $\mathbf{g}$ denote a single-sample stochastic gradient. The *stochastic noise* is defined as $\mathbf{N} := \mathbf{g} - \mathbf{G}$. Our goal is to determine the magnitude of the noise and how these directions align with the signal. To measure the alignment, we use the following two standard subspace distance metrics.

For two orthonormal matrices $\mathbf{U}_1, \mathbf{U}_2 \in \mathbb{R}^{m \times r}$, we consider both the *spectral distance* (worst-case misalignment) and the *chordal distance* (average-case misalignment):

$$d_{\mathrm{spec}}(\mathbf{U}_1, \mathbf{U}_2) := \mathrm{dist}(\mathbf{U}_1, \mathbf{U}_2) = \|\mathbf{U}_1\mathbf{U}_1^T - \mathbf{U}_2\mathbf{U}_2^T\|_2 = \max_{i \in [r]}\{\sin\theta_i\},$$

and

$$d_{\mathrm{chord}}(\mathbf{U}_1, \mathbf{U}_2) := \frac{1}{\sqrt{r}}\|\mathbf{U}_1\mathbf{U}_1^T - \mathbf{U}_2\mathbf{U}_2^T\|_F = \sqrt{\frac{1}{r}\sum_{i=1}^{r}\sin^2\theta_i},$$

where $\theta_1, \ldots, \theta_k$ are the principal angles between the subspaces (Definition B.3).

These two distances can be efficiently computed by:

$$d_{\mathrm{spec}}(\mathbf{U}_1, \mathbf{U}_2) = \sqrt{\lambda_{\max}(\mathbf{B})}, \quad d_{\mathrm{chord}}(\mathbf{U}_1, \mathbf{U}_2) = \sqrt{\frac{\mathrm{trace}(\mathbf{B})}{r}},$$

where $\mathbf{A} = \mathbf{U}_1^T\mathbf{U}_2 \in \mathbb{R}^{r \times r}$ and $\mathbf{B} = \mathbf{I}_r - \mathbf{A}\mathbf{A}^T$. Both metrics range from $0$ (identical subspaces) to $1$ (orthogonal subspaces). The total distance $d_r(\mathbf{N}, \mathbf{G})$ is defined as the maximum distance between the top-$r$ left subspaces and top-$r$ right subspaces:

$$d_r(\mathbf{N}, \mathbf{G}) = \max\{d(\mathbf{U}_N, \mathbf{U}_G), d(\mathbf{V}_N, \mathbf{V}_G)\}.$$

At each checkpoint, we perform the following steps:

1. Compute $\mathbf{G}$ using a large batch-size of $B = 4096$. Compute its top-$r$ left and right singular vectors $\mathbf{U}_G \in \mathbb{R}^{m \times r}$ and $\mathbf{V}_G \in \mathbb{R}^{n \times r}$ with $r = 5$.

2. For each of $b = 50$ independent noise samples:
    (a) Compute a single sample stochastic gradient $\mathbf{g}$.
    (b) Compute the noise matrix $\mathbf{N} = \mathbf{g} - \mathbf{G}$.
    (c) Compute and record the top-$r$ singular values and vectors ($\mathbf{U}_N$ and $\mathbf{V}_N$) of $\mathbf{N}$.
    (d) Compute and record the subspace distance $d_r(\mathbf{N}, \mathbf{G})$.

The results are shown in Figure F.14 and F.15. We observe two consistent phenomena: 1) The top singular values of the noise frequently exceed the signal norm $\|\mathbf{G}\|_2$ by a large margin. This confirms the presence of "spectral spikes" in the noise. Moreover, since the bulk of the singular values of stochastic gradient is small, the high-magnitude components in the noise should be low-rank (according to C.6). 2) The subspace distances between the noise and signal principal directions are consistently close to 1. This indicates that the dominant noise spikes are nearly orthogonal to the principal directions of the true signal.

### F.3. LLM Pretraining Tasks.

**Hardware.** All experiments were conducted on a computation node equipped with $4\times$ NVIDIA H100 (96GB) GPUs.

**Dataset.** We use a subset of the FineWeb-Edu dataset (Lozhkov et al., 2024), partitioned into training and validation sequences. After completion of training, we report the full validation loss and perplexity.

**Model architecture**. We adopt the standard decoder-only transformer following (Semenov et al., 2025). Key components include SwiGLU activation functions (Shazeer, 2020), Rotary Positional Embeddings (RoPE) (Su et al., 2024) RMSNorm (Zhang & Sennrich, 2019) with $\epsilon = 10^{-5}$. We utilize the GPT-2 tokenizer (Radford et al., 2019a) with a vocabulary size of 50,304. Unless otherwise specified, we employ untied embeddings (i.e., do not use weight tying (Press & Wolf, 2017)). Detailed configurations for all considered model architectures are presented in Table F.1.

*Table F.1.* Configuration of the considered model architecture.

| PARAMETERS | 124M | 160M | 780M | 820M |
|---|---|---|---|---|
| EMBED DIMENSION | 768 | 768 | 1280 | 2048 |
| # ATTENTION HEADS | 12 | 12 | 20 | 16 |
| # LAYERS | 12 | 12 | 32 | 12 |
| UNTIED EMBED | FALSE | TRUE | TRUE | TRUE |

**Optimizers.** As detailed in the main text, we benchmark SPECTRA against state-of-the-art optimizers, including AdEMAMix (Pagliardini et al., 2025), AdamW (Loshchilov & Hutter, 2019), Signum (Bernstein et al., 2018), D-Muon (Liu et al., 2025), Mars (AdamW) (Yuan et al., 2025), and SOAP (Vyas et al., 2024). We follow the evaluation protocol of (Semenov et al., 2025), which provides comprehensive descriptions of these baselines. The **memory footprint** of each optimizer is reported in Table F.3; notably, SOAP, Mars, and AdEMAMix incur the highest memory overhead. Applying SPECTRA does not increase memory requirements. The **per-step timing** comparison of each optimizer is reported in Table F.2. The overhead of SPECTRA (due to SSC) shrinks dramatically as scale increases, as the forward/backward cost (scaling with batch size, accumulation steps, sequence length, number of layers, hidden dimension) dominates at larger scale.

**Hyperparameters.** We report the specific hyperparameter settings used across all experiments in the following tables (the row 'Gradient Clipping' means the pre global Frobenius clipping threshold applied to all the parameters.). We adopt the majority of configurations directly from the benchmark protocol (Semenov et al., 2025), while additionally tuning the

*Table F.2.* Per-step timing at 160M, 780M, and 820M scale (4× H100 GPU, averaged over 180 steps after warm-up). SPECTRA overhead shrinks from ∼17–19% at 160M to ∼2% at 780M/820M as the forward/backward pass dominates at larger scale. D-Muon has comparable overhead to SPECTRA due to its NS-based orthogonalization (∼9% at 160M, ∼1–2% at 780M/820M). SOAP is the most expensive optimizer, adding +49% at 160M and +21% at 780M vs. AdamW, due to its larger memory footprint (requiring more accumulation steps), periodic eigendecomposition, and per-step gradient projection.

| Method | 160M | | | 780M | | | 820M | | |
|---|---|---|---|---|---|---|---|---|---|
| | Total (ms) | Opt Step (%) | Overhead | Total (ms) | Opt Step (%) | Overhead | Total (ms) | Opt Step (%) | Overhead |
| AdamW | 200.7 | 1.0 | — | 6550 | 0.2 | — | 4412 | 0.2 | — |
| + Pre-Clip | 232.6 | 13.7 | +15.9% | 6640 | 1.4 | +1.4% | 4480 | 1.7 | +1.5% |
| + SPECTRA | 238.3 | 15.6 | +18.7% | 6654 | 1.7 | +1.6% | 4504 | 2.4 | +2.1% |
| Signum | 203.2 | 1.9 | — | 6536 | 0.3 | — | 4408 | 0.5 | — |
| + SPECTRA | 241.2 | 17.1 | +18.7% | 6667 | 1.9 | +2.0% | 4514 | 2.6 | +2.4% |
| AdEMAMix | 207.9 | 4.3 | — | 6575 | 0.7 | — | 4456 | 1.3 | — |
| + SPECTRA | 242.5 | 18.2 | +16.6% | 6691 | 2.2 | +1.8% | 4568 | 3.3 | +2.5% |
| MARS | 210.8 | 5.5 | — | 6591 | 0.8 | — | 4464 | 1.4 | — |
| + SPECTRA | 248.3 | 18.1 | +17.8% | 6699 | 2.4 | +1.6% | 4553 | 3.4 | +2.0% |
| D-Muon | 220.5 | 8.8 | — | 6637 | 1.4 | — | 4496 | 2.1 | — |
| SOAP | 299.1 | 33.2 | — | 7917 | 7.9 | — | 4868 | 11.0 | — |

Opt Step = optimizer step fraction of total step time. Overhead = overhead vs. corresponding base optimizer.

*Table F.3.* Memory requirements of different optimizers. $M$ denotes total model parameters, $M_{2d}$ and $M_{1d}$ denote 2D and 1D parameters respectively, $m$ and $v$ denote the first and second momentum buffer, and $d_i$ represents the size of each dimension of a parameter tensor (e.g., for a matrix of shape $(d_1, d_2)$, both $d_1$ and $d_2$ are considered)

| Optimizer | State Buffers | Memory | Factor |
|---|---|---|---|
| AdamW | $m, v$ | $2M$ | $2\times$ |
| AdEMAMix | $m_{\text{fast}}, m_{\text{slow}}, v$ | $3M$ | $3\times$ |
| D-Muon[1] | $m$ (2D), $m, v$ (1D) | $M_{2d} + 2M_{1d}$ | $< 2\times$ |
| Signum | $m$ (momentum) | $M$ | $1\times$ |
| SOAP[2] | $m, v, \text{GG}, \text{Q}$ | $2M + \sum_{d_i \leq D} 2d_i^2$ | $\approx 4\times$ |
| MARS (AdamW)[3] | $m, v, g_{\text{prev}}$ (2D) | $\approx 3M$ | $\approx 3\times$ |

[1]D-Muon uses momentum-only updates for 2D matrices (with rows $< 10000$) and AdamW for others including embeddings/biases.
[2]SOAP stores preconditioner matrices for any parameters with $d_i \leq D$ (default $D = 10000$). For typical LLM architectures with $d_i \sim 4096$, preconditioner overhead is approximately $2M$.
[3]MARS stores gradient history for variance reduction where $g_{\text{prev}}$ represents the previous gradient.

learning rate, as it generally has the most significant influence on performance. For the WSD learning rate schedule, we consistently adopt the recommended square root decay to zero, initiating at 80% of the total training steps. For the spectral clipping schedule, as discussed in Section 5.2, we use the following default choice:

$$c_k = \begin{cases} \frac{c\eta}{\eta_k} & k \leq K_{\text{warmup}}, \\ c & \text{otherwise} \end{cases} . \qquad \text{(standard)}$$

**Comparison with Post-Orthogonalization.** A natural alternative to spectral clipping is to apply explicit post-orthogonalization to the update matrix. This update rule is given by:

$$\mathbf{X}_{k+1} = (1 - \lambda\eta_k)\mathbf{X}_k - \alpha\eta_k \, \text{orth}(\mathbf{U}_k); ,$$

where $\alpha = \max(\sqrt{\frac{m}{n}}, 1)$ scales the update as before, and the operator $\text{orth}(\cdot)$ normalizes all singular values to 1. We implement $\text{orth}(\cdot)$ using the same Newton-Schultz iterations employed by Muon (Jordan et al., 2024).

We compare Spectra-AdamW against this AdamW-Post-Orth method on the 160M-parameter task trained for 16.8B tokens. The results, summarized in Table F.8, show the final validation loss across various learning rates. We observe that Spectra-AdamW outperforms the best-performing post-orthogonalized AdamW baseline in both schedule configurations.

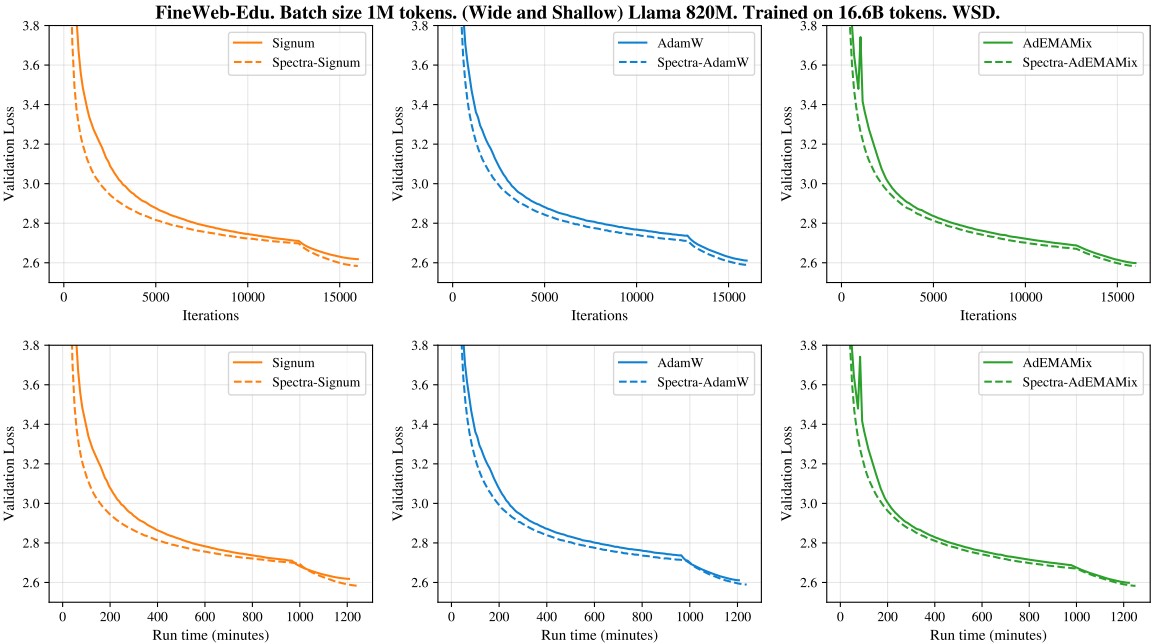

*Figure F.7.* Training dynamics of the 820M-parameter model. The plots display validation loss with respect to training iterations (up) and wall-clock time (down), conducted on 4 H100 GPUs. We use a micro batchsize of 31 with 8 accumulation steps on each machine. The hyperparameters such as learning rate used for each method are reported in the tables in Section F.3.

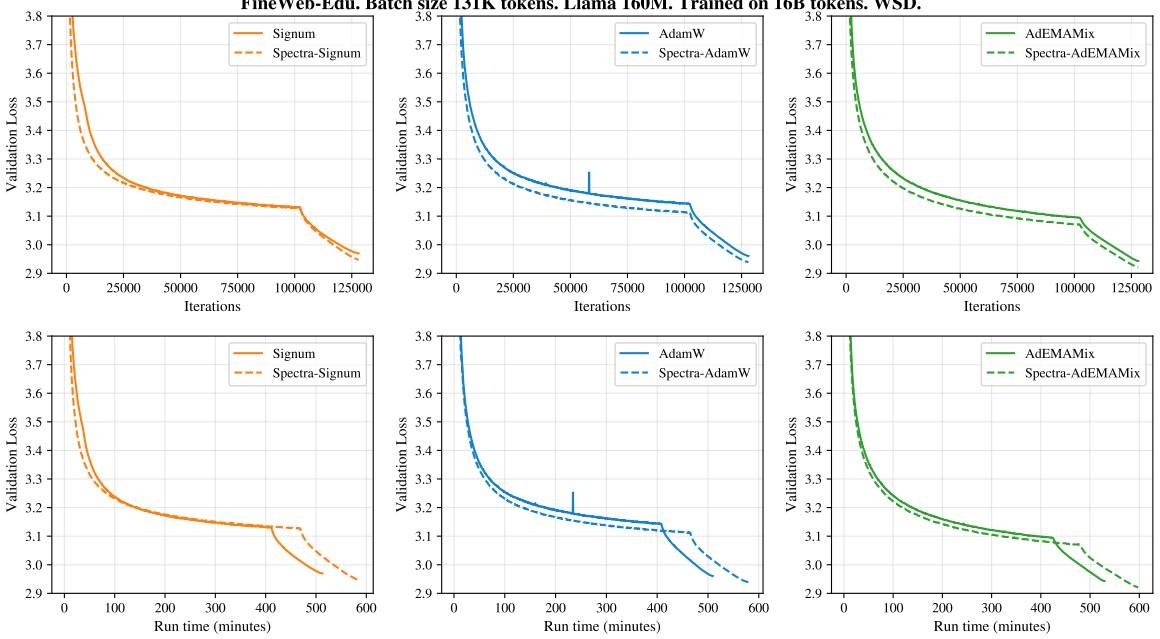

*Figure F.8.* Training dynamics of the 160M-parameter model. The plots display validation loss with respect to training iterations (up) and wall-clock time (down), conducted on 4 H100 GPUs. We use a micro batchsize of 64 with 1 accumulation step on each machine. The hyperparameters such as learning rate used for each method are reported in the tables in Section F.3.

**Discussion on pre-spectral clipping.** We conduct an ablation on pre-soft spectral clipping under noisier training conditions using SlimPajama with a small batch size of 8 (sequence length 512, 160M model, 128k iterations). As shown in Table F.4, pre-SSC improves SPECTRA-Signum with only post-SSC by 0.014 in this setting, compared to 0.005 on the cleaner FineWeb-Edu dataset with batch size 32, showing that pre-SSC could become more beneficial as gradient noise increases.

We further observe that a decaying threshold schedule (cosine from $c = 10$ to $c = 0.01$) outperforms a constant threshold ($c = 0.1$). This is expected: the spectral norm of both the large-batch gradient and the noise decreases over training (e.g, Figure F.9), and it also varies across layers, making a fixed $c$ suboptimal — it inevitably over-clips in later stages when gradient norms get smaller. Despite these improvements, we note that pre-spectral clipping provides a fundamentally more modest benefit than post-clipping, because post-clipping directly controls the spectral norm of the optimizer update, which stabilizes training and provides implicit weight regularization, whereas pre-clipping only safeguards against gradient spikes before the optimizer processes them. At large scale, the combined overhead of pre-clipping and post-clipping remains under approximately 4%. Investigating pre-clipping with adaptive, layer-wise thresholds and in high-noise settings such as reinforcement learning is an interesting direction for future work.

*Table F.4.* Pre-spectral clipping ablation on SlimPajama (160M, batch size 8, seq len 512, 128k iterations). Pre-spectral clipping improves the baseline SPECTRA by 0.014 nats under noisy conditions, and a decaying threshold schedule outperforms a constant threshold.

| Method | Val Loss |
|---|---|
| Signum (lr=$2 \times 10^{-4}$) | 4.955 |
| Signum (lr=$10^{-4}$) | 3.688 |
| SPECTRA-Signum (lr=$10^{-3}$) | 3.439 |
| + pre-clip (constant $c = 0.1$) | 3.431 |
| + pre-clip (cos decay, $c$: $10 \rightarrow 0.01$) | **3.425** |

## F.4. Evaluation of downstream Tasks.

We evaluate the pretrained models on in-context learning tasks using the `lm-evaluation-harness` package (Gao et al., 2024). We benchmark on the following standard datasets: ARC (Clark et al., 2018), BoolQ (Christopher et al., 2019), PIQA (Bisk et al., 2020), HellaSwag (Zellers et al., 2019), WinoGrande (win, 2019), OpenBookQA (Mihaylov et al., 2018), LambdaOpenAI (Radford et al., 2019b), SciQ (Johannes Welbl, 2017), and MMLU (Hendrycks et al., 2021b;a). All tasks are evaluated in a 5-shot setting, where the model is provided with five complete question-answer examples before the test question. For each task, the model scores each candidate answer by computing the log-probability of the answer text conditioned on the question, and predicts the choice with the highest log-probability. We report two accuracy metrics: the raw accuracy based on total log-probability, and the length-normalized accuracy, which divides each log-probability by the number of tokens in the corresponding choice to mitigate bias toward shorter answers. Following standard practice, we report length-normalized accuracy for HellaSwag, ARC, PIQA, OpenBookQA, and SciQ, and the standard accuracy for the remaining tasks.

*Table F.5.* Short training (8B tokens or 8k steps) of 1.5B muP models. The details and scripts can be found in `https://github.com/mlolab/llm-spectral-clipping/tree/main/scripts/1.5b-mup-models`.

| Optimizer | LR | Val Loss | Optimizer | LR | Val Loss |
|---|---|---|---|---|---|
| SOAP | $5 \times 10^{-3}$ | 2.653 | Spectra-Signum | $2 \times 10^{-3}$ | 2.642 |
| D-Muon | $2 \times 10^{-3}$ | 2.658 | | $3 \times 10^{-3}$ | 2.638 |
| MARS | $2 \times 10^{-3}$ | 2.643 | | $5 \times 10^{-3}$ | 2.643 |
| AdEMAMix | $2 \times 10^{-3}$ | 2.647 | Spectra-AdEMAMix | $2 \times 10^{-3}$ | 2.640 |
| | $3 \times 10^{-3}$ | 2.642 | | $3 \times 10^{-3}$ | **2.635** |

*Table F.6.* Hyperparameter configurations for **Spectra-AdamW**.

| Hyperparameter | 160M (wsd) | 160M (cos) | 780M (wsd) | 820M (wsd) |
|---|---|---|---|---|
| Learning Rate $\eta$ | 5e-4, **1e-3**, 2e-3 | 1e-3, **2e-3**, 3e-3 | 1e-3 | 1e-3 |
| Clipping Threshold $c$ | 5.0, **10.0** | 5.0 | 10.0 | 10.0 |
| Clipping Schedule | standard | standard | standard | standard |
| Others | | Same as in Table F.9 | | |

*Table F.7.* SOAP vs. SPECTRA-Signum at larger scales (final validation loss). SOAP is tuned over three learning rates. Despite SOAP requiring ∼19% more time per step at 780M (+8% at 820M) and nearly 2× more memory than SPECTRA-Signum, it achieves comparable results at 780M and worse results at 820M.

| Method (lr) | 780M | 820M |
|---|---|---|
| SOAP ($10^{-3}$) | 2.586 | 2.606 |
| SOAP ($3 \times 10^{-3}$) | **2.571** | 2.602 |
| SOAP ($5 \times 10^{-3}$) | 2.574 | 2.603 |
| SPECTRA-Signum ($10^{-3}$) | 2.573 | **2.594** |

*Table F.8.* Final validation loss comparison for the 160M Llama model trained on 16.8B tokens. Best results per schedule are **bolded**.

| Schedule | Method | Learning Rate | | | | |
|---|---|---|---|---|---|---|
| | | 1e-3 | 2e-3 | 3e-3 | 4e-3 | 5e-3 |
| wsd | AdamW-Post-Orth | 2.988 | 2.973 | 2.974 | – | – |
| wsd | Spectra-AdamW | **2.936** | – | – | – | – |
| cos | AdamW-Post-Orth | 3.009 | 2.966 | 2.955 | 2.953 | 2.956 |
| cos | Spectra-AdamW | – | **2.922** | – | – | – |

*Table F.9.* Hyperparameter configurations for **AdamW**.

| Hyperparameter | 160M (wsd) | 160M (cos) | 780M (wsd) | 820M (wsd) |
|---|---|---|---|---|
| Batch Size | 256 | 256 | 1012 | 992 |
| Sequence Length | 512 | 512 | 1024 | 1024 |
| Warmup Steps | 2000 | 2000 | 2000 | 2000 |
| Weight Decay $\lambda$ | 0.1 | 0.1 | 0.1 | 0.1 |
| Gradient Clipping | 0.5 | 0.5 | 0.1 | 0.1 |
| Momentum $\beta_1$ | 0.8 | 0.8 | 0.9 | 0.9 |
| Momentum $\beta_2$ | 0.999 | 0.999 | 0.999 | 0.999 |
| Learning Rate $\eta$ | 5e-4, **1e-3**, 2e-3 | 5e-4, 1e-3, **2e-3**, 3e-3 | 5e-4, **1e-3**, 2e-3 | 5e-4, **1e-3**, 2e-3 |

*Table F.10.* Hyperparameter configurations for **AdEMAMix**.

| Hyperparameter | 160M (wsd) | 160M (cos) | 780M (wsd) | 820M (wsd) |
|---|---|---|---|---|
| Batch Size | 256 | 256 | 1012 | 992 |
| Sequence Length | 512 | 512 | 1024 | 1024 |
| Warmup Steps | 2000 | 2000 | 2000 | 2000 |
| Weight Decay $\lambda$ | 0.1 | 0.1 | 0.1 | 0.1 |
| Gradient Clipping | 0.5 | 0.5 | 0.1 | 0.1 |
| Momentum $\beta_1$ | 0.9 | 0.9 | 0.9 | 0.9 |
| Momentum $\beta_2$ | 0.999 | 0.999 | 0.999 | 0.999 |
| Momentum $\beta_3$ | 0.999 | 0.999 | 0.999 | 0.999 |
| Coefficient $\alpha$ | 8 | 8 | 8 | 8 |
| Learning Rate $\eta$ | 5e-4, **1e-3**, 2e-3 | 1e-3, **2e-3**, 3e-3 | 2e-4, **5e-4**, 1e-3 | 5e-4, **1e-3**, 2e-3 |

*Table F.11.* Hyperparameter configurations for **Spectra-AdEMAMix**.

| Hyperparameter | 160M (wsd) | 160M (cos) | 780M (wsd) | 820M (wsd) |
|---|---|---|---|---|
| Learning Rate $\eta$ | 5e-4, **1e-3**, 2e-3 | 1e-3, **2e-3**, 3e-3 | 5e-4, **1e-3** | 1e-3 |
| Clipping Threshold $c$ | 5.0, **10.0** | 5.0 | 10.0 | 10.0 |
| Clipping Schedule | standard | standard | standard | standard |
| Others | | Same as in Table F.10 | | |

*Table F.12.* Hyperparameter configurations for **Signum**.

| Hyperparameter | 160M (wsd) | 160M (cos) | 780M (wsd) | 820M (wsd) |
|---|---|---|---|---|
| Batch Size | 256 | 256 | 1012 | 992 |
| Sequence Length | 512 | 512 | 1024 | 1024 |
| Warmup Steps | 8000 | 8000 | 2000 | 2000 |
| Weight Decay $\lambda$ | 0.1 | 0.1 | 0.1 | 0.1 |
| Gradient Clipping | 0.5 | 0.5 | 0.1 | 0.1 |
| Momentum $\beta$ | 0.95 | 0.95 | 0.95 | 0.95 |
| Nesterov momentum | True | True | True | True |
| Learning Rate $\eta$ | 2e-4, **5e-4**, 1e-3 | 5e-4, **1e-3**, 2e-3 | **2e-4**, 5e-4 | 2e-4 |

*Table F.13.* Hyperparameter configurations for **Spectra-Signum**.

| Hyperparameter | 160M (wsd) | 160M (cos) | 780M (wsd) | 820M (wsd) |
|---|---|---|---|---|
| Warmup Steps | 2000 | 2000 | 2000 | 2000 |
| Learning Rate $\eta$ | 5e-4, **1e-3**, 2e-3 | 1e-3, **2e-3**, 3e-3 | 1e-3 | 1e-3 |
| Clipping Threshold $c$ | 10.0 | 5.0 | 10.0 | 10.0 |
| Clipping Schedule | standard | standard | standard | standard |
| Others | | Same as in Table F.12 | | |

*Table F.14.* Hyperparameter configurations for **Spectra-Mars**.

| Hyperparameter | 160M (wsd) | 780M (wsd) | 820M (wsd) |
|---|---|---|---|
| Warmup Steps | 2000 | 2000 | 2000 |
| Learning Rate Mars | 3e-3 | 3e-3 | 3e-3 |
| Clipping Threshold $c$ | 10.0 | 10.0 | 10.0 |
| Clipping Schedule | standard | standard | standard |
| Others | | Same as in Table F.16 | |

*Table F.15.* Hyperparameter configurations for **D-Muon**.

| Hyperparameter | 160M (wsd) | 160M (cos) | 780M (wsd) | 820M (wsd) |
|---|---|---|---|---|
| Batch Size | 256 | 256 | 1012 | 992 |
| Sequence Length | 512 | 512 | 1024 | 1024 |
| Warmup Steps | 2000 | 2000 | 2000 | 2000 |
| Weight Decay $\lambda$ | 0.1 | 0.1 | 0.1 | 0.1 |
| Gradient Clipping | 0.5 | 0.5 | 0.1 | 0.1 |
| Momentum $\beta$ | 0.95 | 0.95 | 0.95 | 0.95 |
| AdamW $\beta_1$ (1D) | 0.8 | 0.8 | 0.9 | 0.9 |
| AdamW $\beta_2$ (1D) | 0.999 | 0.999 | 0.99 | 0.99 |
| Nesterov momentum | True | True | True | True |
| Learning Rate $\eta$ | 5e-4, **1e-3**, 2e-3 | 5e-4, **1e-3**, 2e-3 | 5e-4, **1e-3** | 1e-3 |

*Table F.16.* Hyperparameter configurations for **Mars (AdamW)**.

| Hyperparameter | 160M (wsd) | 160M (cos) | 780M (wsd) | 820M (wsd) |
|---|---|---|---|---|
| Batch Size | 256 | 256 | 1012 | 992 |
| Sequence Length | 512 | 512 | 1024 | 1024 |
| Warmup Steps | 2000 | 2000 | 2000 | 2000 |
| Weight Decay $\lambda$ | 0.1 | 0.1 | 0.1 | 0.1 |
| Gradient Clipping | 0.5 | 0.5 | 0.1 | 0.1 |
| Learning Rate AdamW | 5e-4, **1e-3** | 1e-3 | 1e-3 | 1e-3 |
| Learning Rate Mars | 3e-3 | 3e-3 | 1e-3, **3e-3** | 3e-3 |
| AdamW $\beta_1$ (1D) | 0.8 | 0.8 | 0.8 | 0.9 |
| AdamW $\beta_2$ (1D) | 0.999 | 0.999 | 0.999 | 0.999 |
| Mars $\beta_1$ | 0.95 | 0.95 | 0.95 | 0.95 |
| Mars $\beta_2$ | 0.99 | 0.99 | 0.99 | 0.99 |
| VR Scaling Factor $\eta$ | 0.025 | 0.025 | 0.025 | 0.025 |

*Table F.17.* Hyperparameter configurations for **SOAP**. (Due to the large memory footprint of SOAP, we need larger gradient accumulation steps to fit the model in memory, which largely extended the training runtime.

| Hyperparameter | 160M (wsd) | 160M (cos) | 780M (wsd) | 820M (wsd) |
|---|---|---|---|---|
| Batch Size | 256 | 256 | 1012 | 992 |
| Sequence Length | 512 | 512 | 1024 | 1024 |
| Warmup Steps | 2000 | 2000 | 2000 | 2000 |
| Weight Decay $\lambda$ | 0.1 | 0.1 | 0.1 | 0.1 |
| Gradient Clipping | 0.5 | 0.5 | 0.1 | 0.1 |
| Preconditioner dimension | 10000 | 10000 | 10000 | 10000 |
| Preconditioning frequency | 10 | 10 | 10 | 10 |
| SOAP $\beta_1$ | 0.9 | 0.9 | 0.95 | 0.95 |
| SOAP $\beta_2$ | 0.999 | 0.999 | 0.95 | 0.95 |
| Learning Rate $\eta$ | 5e-4, **1e-3**, 2e-3 | 1e-3, **2e-3**, 3e-3 | 1e-3, **3e-3**, 5e-3 | 1e-3, **3e-3**, 5e-3 |

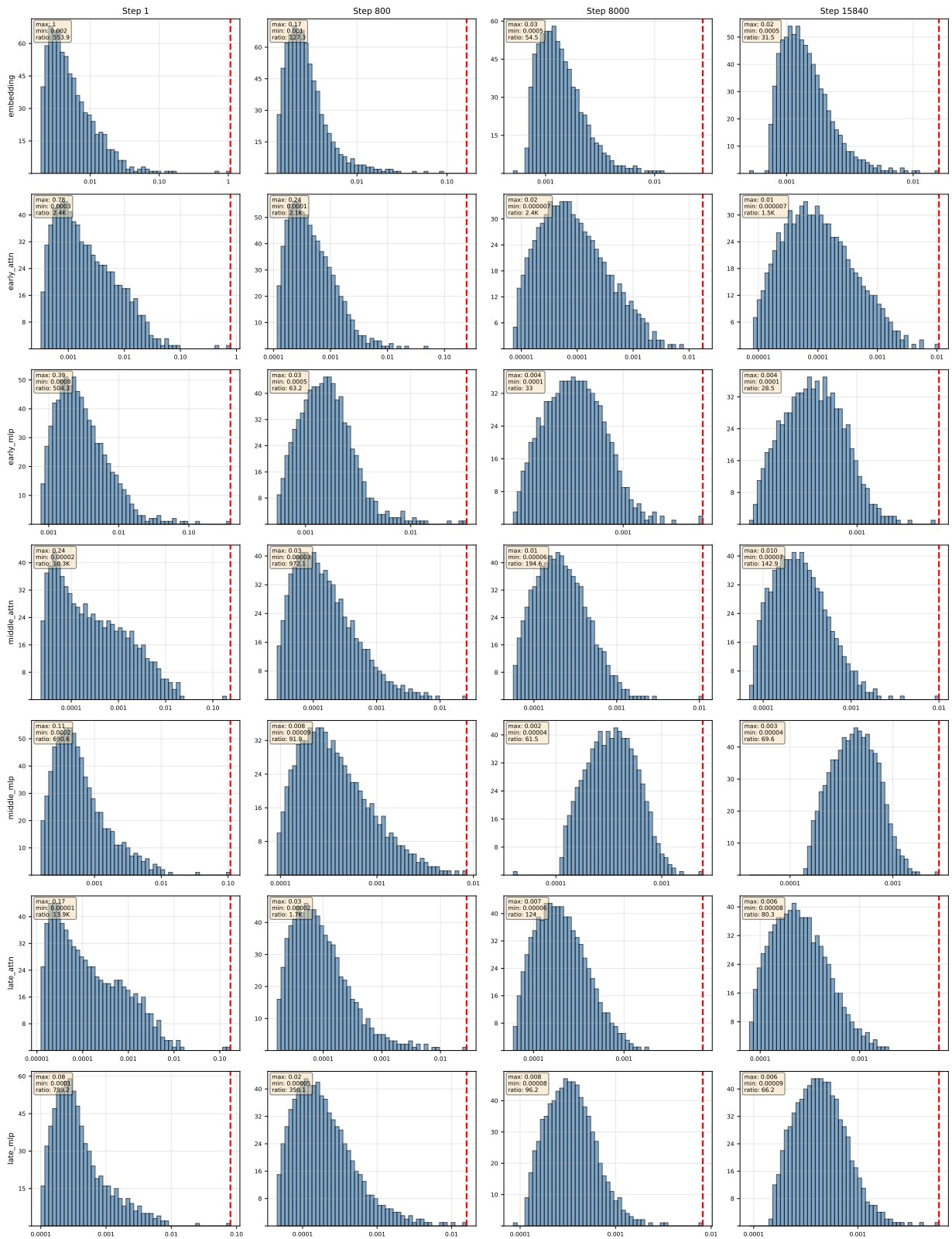

*Figure F.9.* Singular value distribution of raw stochastic gradients for the 124M-parameter model trained with AdamW. We visualize the spectrum at four training stages (0%, 5%, 50%, and 99%) for representative layers across the network depth. Each panel reports the maximum and minimum singular values along with their ratio (condition number). We observe that a small number of singular values are orders of magnitude larger than the rest, leading to high condition numbers.

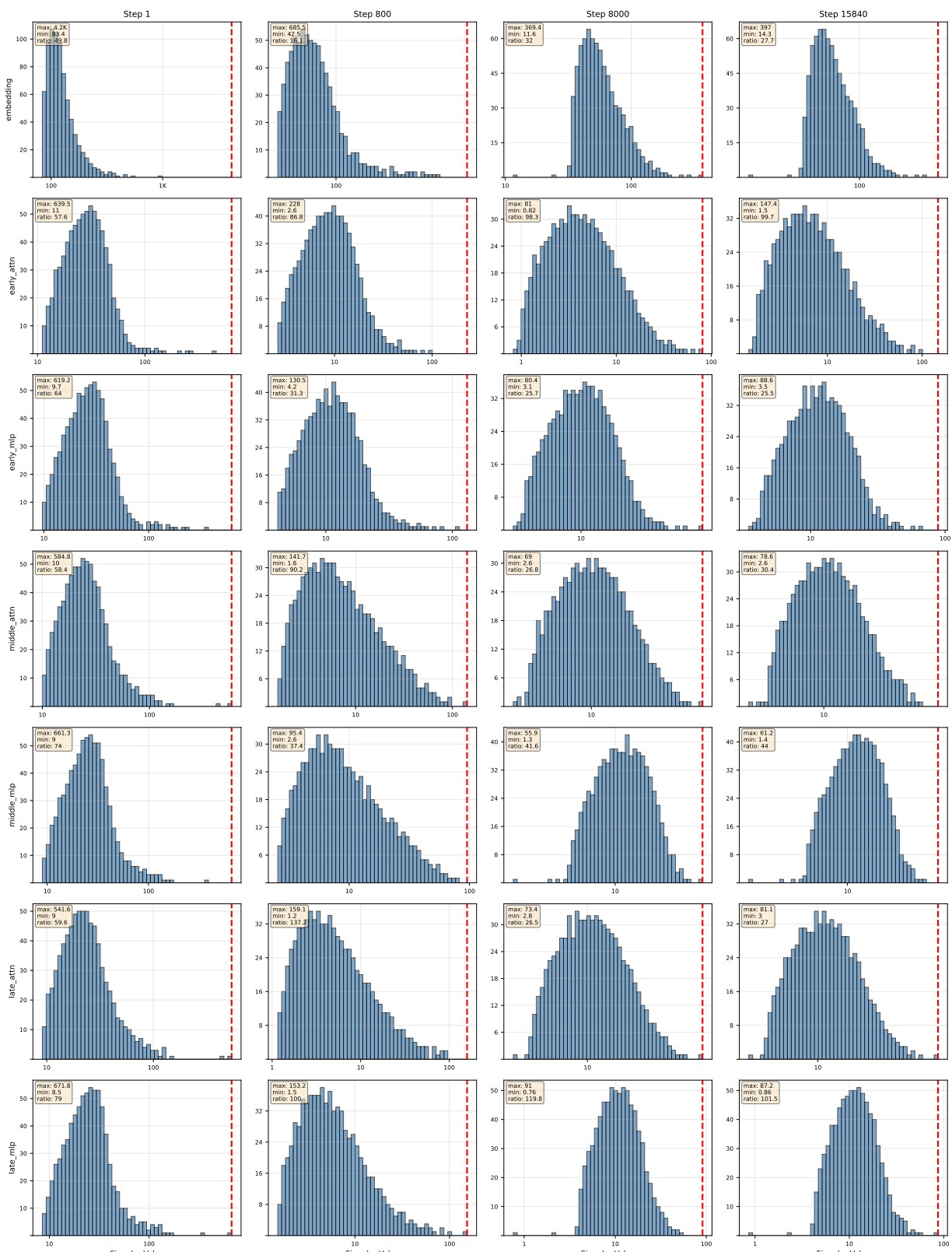

*Figure F.10.* Singular value distribution of update matrices ($\mathbf{U}_k$) trained with AdamW on the 124M-parameter model. We visualize the spectrum at four training stages (0%, 5%, 50%, and 99%) for representative layers across the network depth. Each panel reports the maximum and minimum singular values along with their ratio (condition number). AdamW updates show a more uniform spectral distribution compared to the raw gradient. However, they fail to impose a hard ceiling.

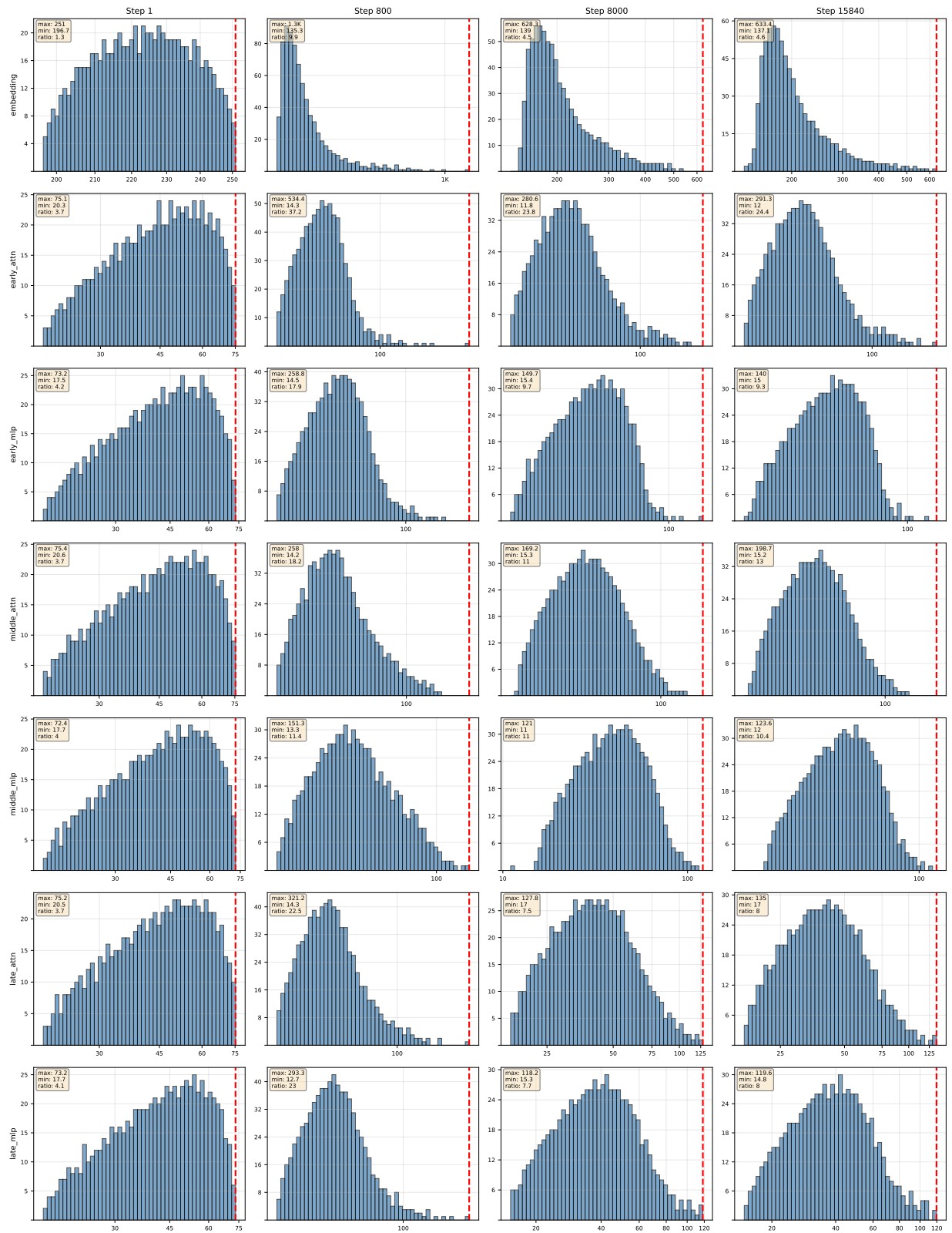

*Figure F.11.* Singular value distribution of update matrices ($\mathbf{U}_k$) trained with Signum on the 124M-parameter model. We visualize the spectrum at four training stages (0%, 5%, 50%, and 99%) for representative layers across the network depth. Each panel reports the maximum and minimum singular values along with their ratio (condition number). The spectral norm of the update matrices ranges from 70 to 1300.

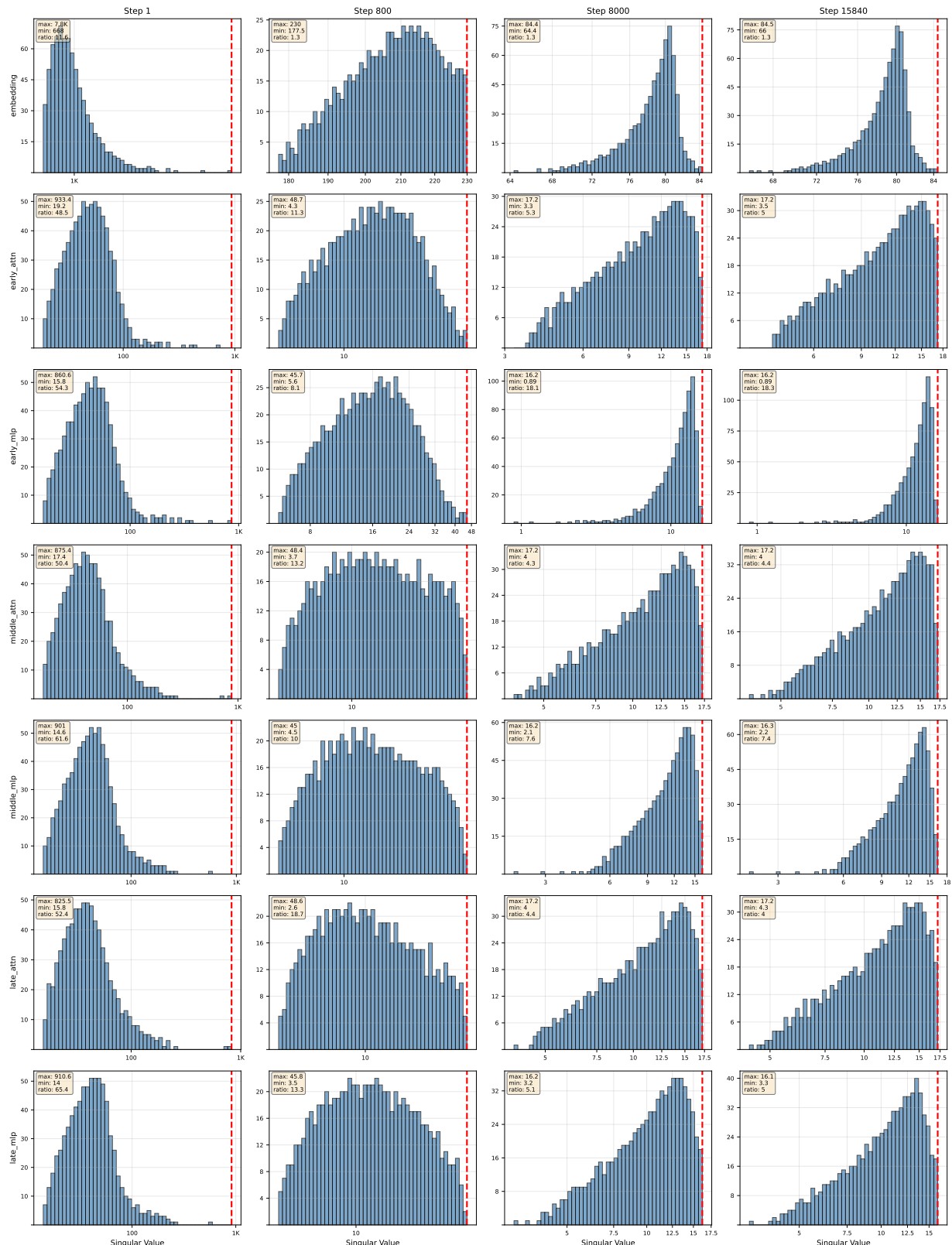

*Figure F.12.* Singular value distribution of update matrices $\mathbf{U}_k = \alpha \operatorname{clip}_{c_k}^{\mathrm{sp}}(\mathbf{U}_k^{\mathrm{AdamW}})$ trained with Spectra-AdamW on the 124M-parameter model. We visualize the spectrum at four training stages (0%, 5%, 50%, and 99%) for representative layers across the network depth. Each panel reports the maximum and minimum singular values along with their ratio (condition number). The spectral norms of the update matrices exhibit hard ceilings and the condition numbers are small.

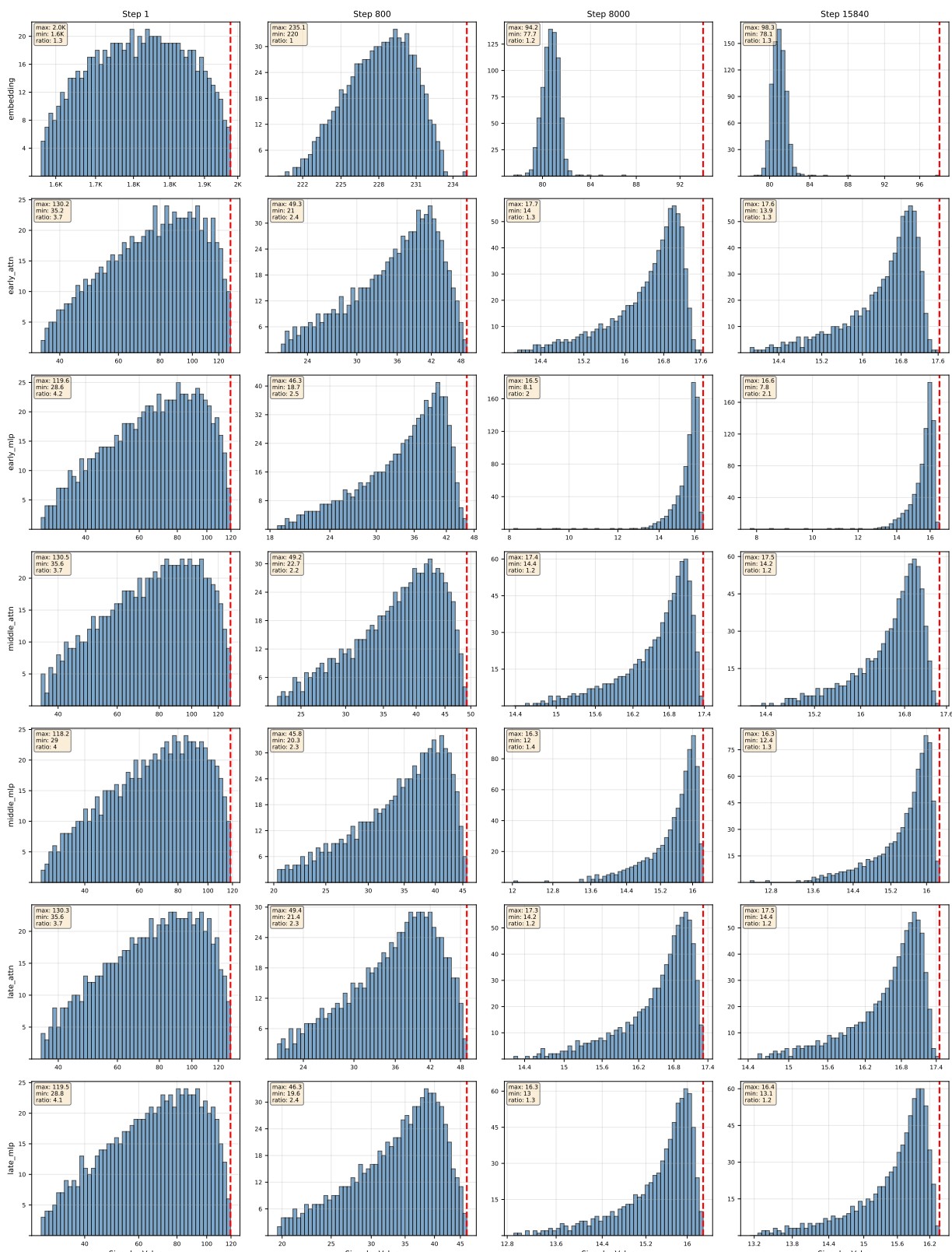

*Figure F.13.* Singular value distribution of update matrices $\mathbf{U}_k = \alpha \, \mathrm{clip}^{\mathrm{sp}}_{c_k}(\mathbf{U}^{\mathrm{Signum}}_k)$ trained with Spectra-Signum on the 124M-parameter model. We visualize the spectrum at four training stages (0%, 5%, 50%, and 99%) for representative layers across the network depth. Each panel reports the maximum and minimum singular values along with their ratio (condition number). The spectral norms of the update matrices exhibit hard ceilings and the condition numbers are small.

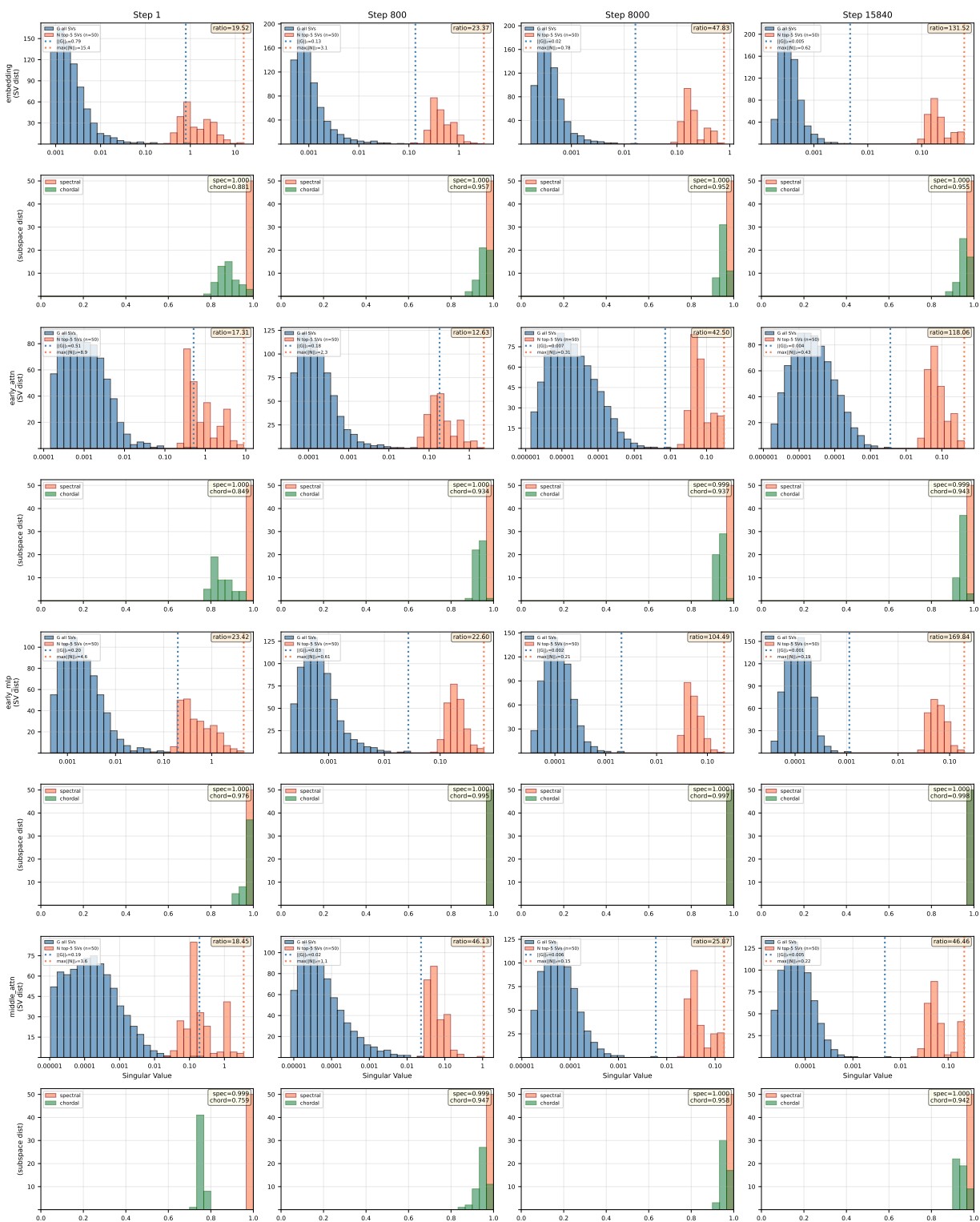

*Figure F.14.* For each layer, we report the singular value distribution of large-batch gradient **G** (blue bar) and top-5 singular value distribution of the noise matrices $\{\mathbf{N}_i\}_{i=1}^{50}$ (red) in the first row. In the second row, we report the distribution of both the spectral and chordal subspace distances of the top-5 principal directions between the noise matrices and the large-batch signal. Results are shown for representative layers of the 124M model (AdamW) at four training stages 0%, 5%, 50%, and 99%. (Part 1) This confirms the presence of "sparse spectral spikes" in the noise and indicates that the dominant noise spikes are nearly orthogonal to the principal directions of the true signal.

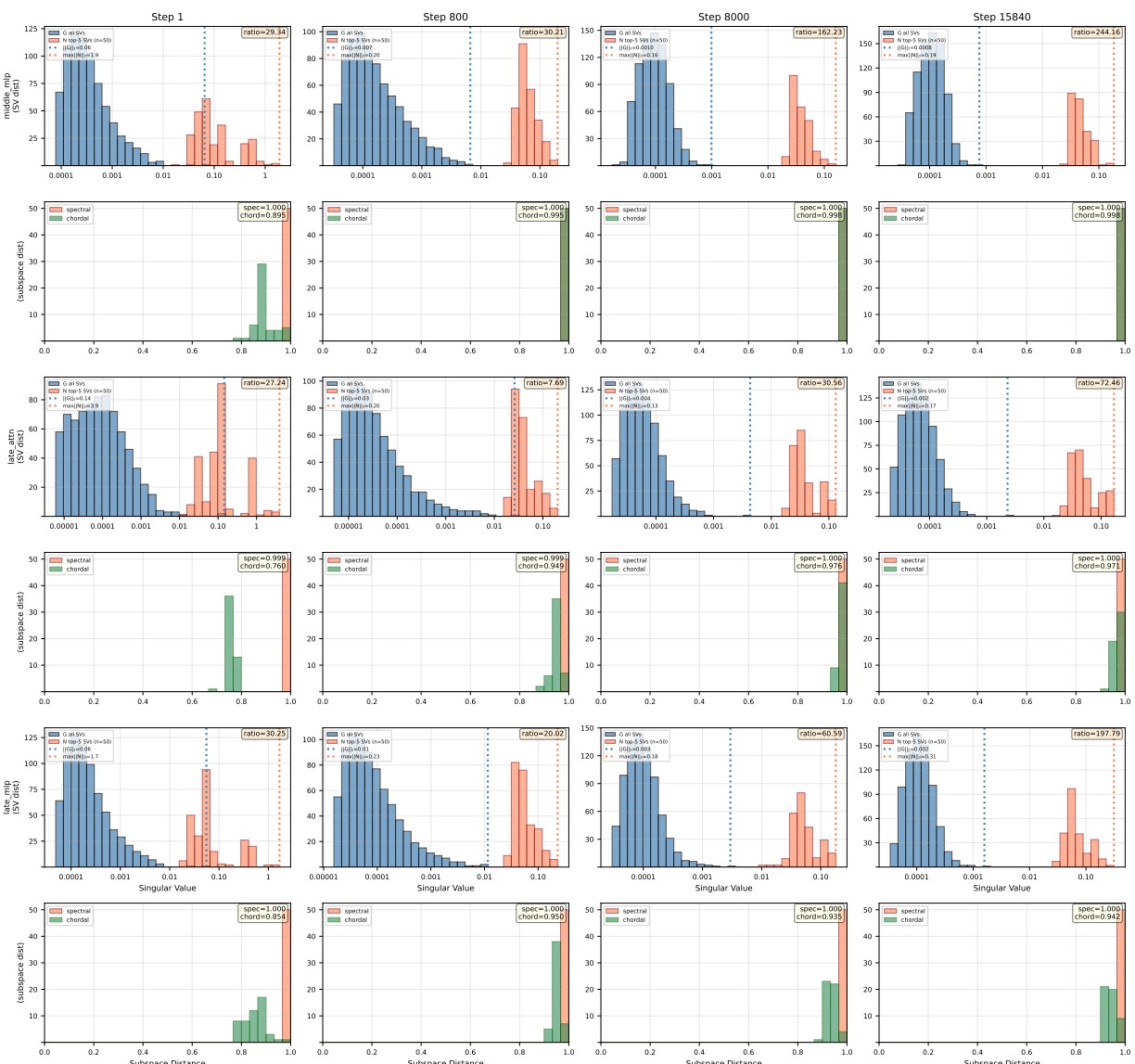

*Figure F.15.* Part 2 of Figure F.14. We observe a consistent phenomenon as in part 1.

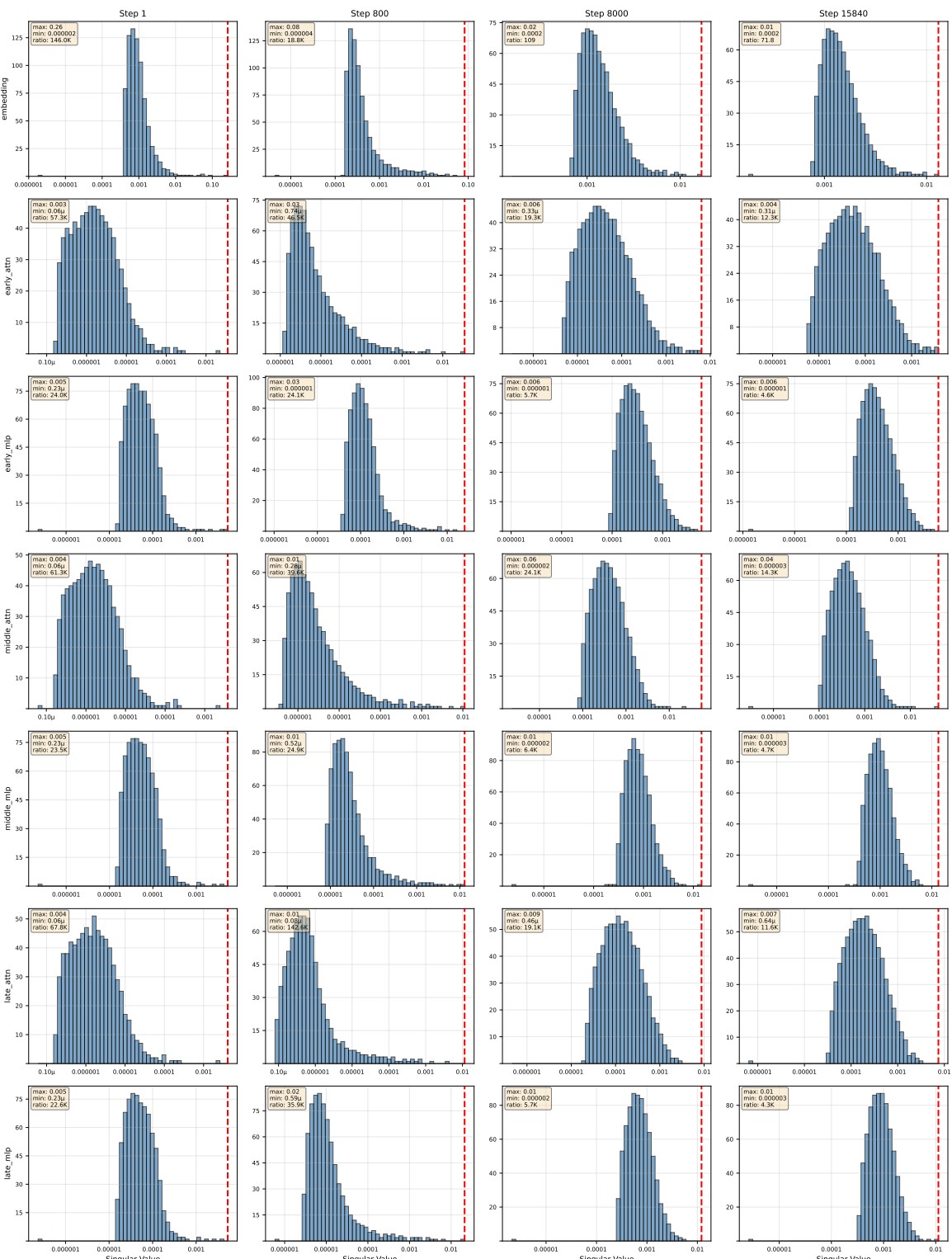

*Figure F.16.* Singular value distribution of raw stochastic gradients for the 124M-parameter GPT with muP trained with AdamW. The heavy-tailed 'spectral outlier' phenomenon is less significant than 124M-llama model without muP.

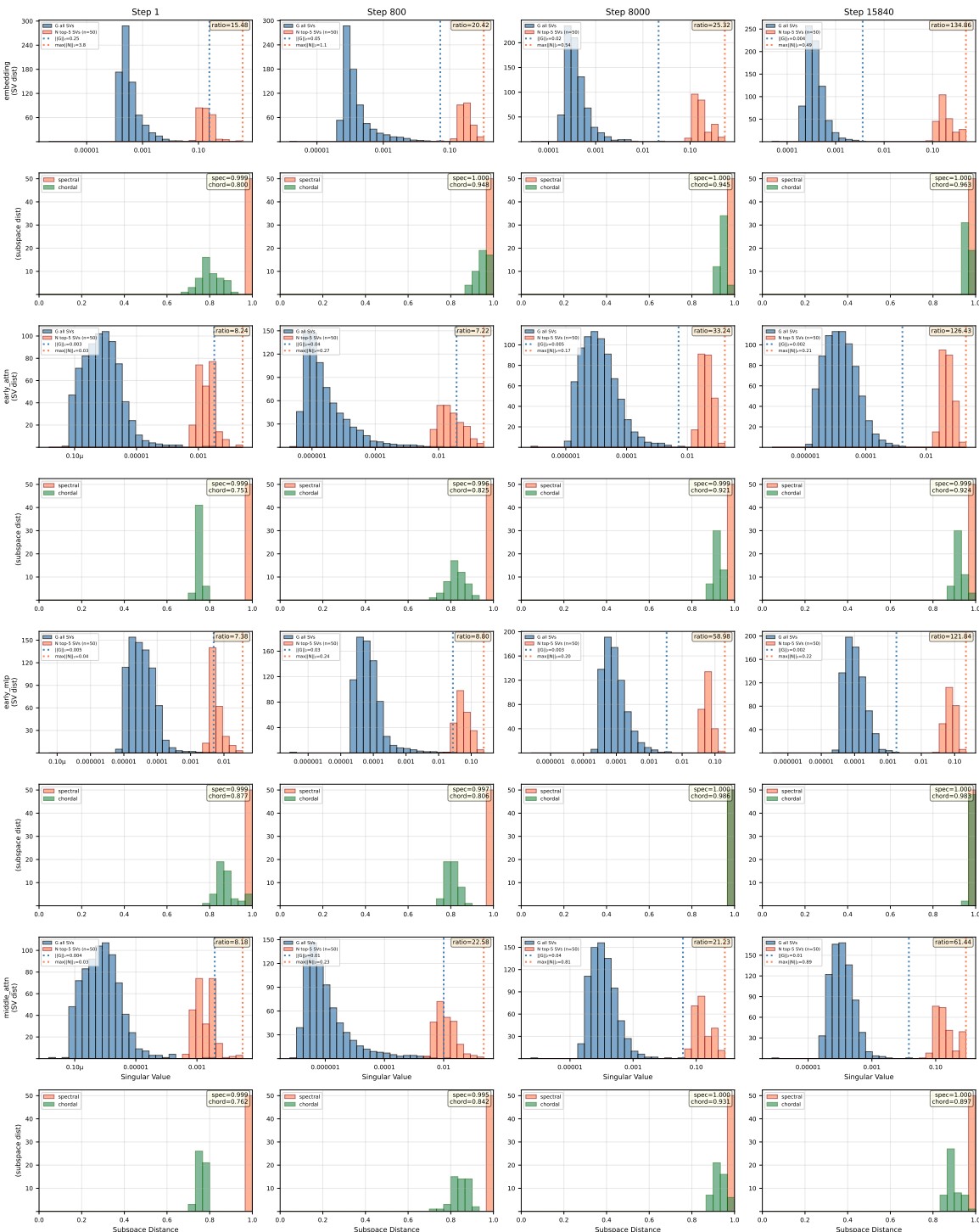

*Figure F.17.* For each layer, we report the singular value distribution of large-batch gradient **G** (blue bar) and top-5 singular value distribution of the noise matrices $\{\mathbf{N}_i\}_{i=1}^{50}$ (red) in the first row. In the second row, we report the distribution of both the spectral and chordal subspace distances of the top-5 principal directions between the noise matrices and the large-batch signal. Results are shown for representative layers of the 124M GPT model with muP (AdamW) at four training stages 0%, 5%, 50%, and 99%. The presence of "sparse spectral spikes" in the noise appears mostly only in the end of the training.

