# OpenReview forum: "Enhancing LLM Training via Spectral Clipping"
_ICML.cc/2026/Conference — ICML 2026 regular_

### Official Review · Reviewer_sSio · 2026-03-08

**Soundness:** 2
**Presentation:** 2
**Significance:** 2
**Originality:** 2
**Overall Recommendation:** 4
**Confidence:** 4

**Summary:**

AdamW and its variants are widely used for LLM pre-training but their coordinate-wise adaptivity does not take global spectral structure into account. This leads to two empirical issues: (i) large spectral norm of the updates which lead to instability of training and (ii) sparse spectral spikes where few dominant singular vectors have much larger singular values compared to the non-dominant counterparts. The authors propose a general framework called SPECTRA to address these issues which performs post-spectral clipping to constrain the spectral norm of the updates and pre-spectral clipping to filter out sparse spectral spikes. They analyze that post-spectral clipping in SPECTRA can be interpreted as a composite Frank-Wolfe method with spectral norm constraints and the weight norm regularization. The proposed strategy can be integrated into various Adam-type optimizers including AdamW, Signum, and AdEMAMix. The experiments on LLM pre-training show that SPECTRA consistently improves performance across different model sizes and optimizers.

**Compliance With Llm Reviewing Policy:**

Affirmed.

**Final Justification:**

During the rebuttal, the authors addressed my concerns regarding the runtime overhead and effectiveness of the pre-spectral clipping. The authors have also successfully clarified the differences between the paper and recent blog posts. Thus, I have updated my score from 3 weak reject to 4 weak accept.

**Key Questions For Authors:**

1. What are the main differences between SPECTRA and the similar prior works [1-3]? What are the differences in methodologies and experimental results?

2. Is SPECTRA sensitive to the clipping threshold $c$? Currently, the paper chooses $c \approx 10$ without a valid reason (in L375), which makes it unclear how this value was selected and if the proposed strategy works well with other values.

3. In Figure E.2, why is the runtime for SSC smaller compared to Muon at the same number of NS iterations?

4. Is SPECTRA sensitive to the number of Newton-Schultz iterations? As stated in the Weaknesses, the current version seems to lose its effectiveness in terms of runtime possibly due to the overhead for soft spectral clipping. Can we reduce this overhead by reducing the number of Newton-Schultz iterations with minimal performance drop?

5. Does SPECTRA also improve the performance for other optimizers? For example, can you provide the results for SOAP, D-Muon, and Mars (which are included in Figure 3)?

References

[1] Rohan Anil, https://x.com/_arohan_/status/1929945590366122037 \
[2] Jianlin Su, https://kexue.fm/archives/11006 \
[3] Franz Louis Cesista, https://leloykun.github.io/ponder/spectral-clipping/

**Limitations:**

No. While the authors implicitly acknowledge the lack of larger-scale experiments and more theoretical analysis as future work, they are encouraged to explicitly describe the potential limitations of the proposed strategy.

**Strengths And Weaknesses:**

## Strengths
-  Identification of sparse spectral spikes, to my knowledge, has not been explored in the literature. While the literature mainly reports the low-rankness of gradients for LLM pre-training [1], the authors identify that this low-rankness mainly stems from the noise rather than the signal.
- Interpretation of post-spectral clipping as a composite Frank-Wolfe method and the related convergence analysis provide insights into its mechanism.
- The proposed strategy SPECTRA improves the final validation loss for LLM pre-training.


## Weaknesses

### Originality of the method

It seems that the idea of spectral clipping has already been introduced in some X threads and blog posts [2-4]. The authors also acknowledge [4] as a related work in Appendix A. However, the paper lacks a detailed comparison between SPECTRA and these prior works in terms of methodology and experimental results.

### Ineffectiveness of the method in terms of runtime

The main text only reports the final loss and loss vs iterations. In terms of iterations, SPECTRA accelerates the convergence. However, as seen in Figure F.7 and Figure F.8 in the Appendix F, the effectiveness of SPECTRA seems to vanish in terms of runtime. Specifically, SPECTRA rather slows down the convergence for 160M models across all optimizers in Figure F.8. This issue is possibly due to the overhead for soft spectral clipping which requires 10 Newton-Schultz iterations (in L375).

### Ineffectiveness of pre-spectral clipping

The paper introduces pre-spectral clipping as one of the two main components of SPECTRA in the abstract and introduction (“SPECTRA is a … and (ii) optionally applies pre-spectral clipping to raw gradients to filter spectral noise spikes.”). However, as seen in Figure 2, the performance gain with pre-spectral clipping is only marginal which lead authors to simply discard this component (“we disable pre-spectral clipping for SPECTRA because the marginal performance gain does not justify the added computational cost”). Thus, it is unclear if pre-spectral clipping should be emphasized as a core component of the proposed strategy and why the strategy needs this component.

### Minor typo
It seems that L238 should change from “Compared with ~. Spectral ~” to  “Compared with ~, spectral ~.”
In L325, ’sparse spike’ should change to ‘sparse spike’ and batchsize should change to batch size.
In L364, practical should change to practice.

## References

[1] Zhao et al., Galore: Memory-efficient llm training by gradient low-rank projection, ICML 2024 \
[2] Rohan Anil, https://x.com/_arohan_/status/1929945590366122037 \
[3] Jianlin Su, https://kexue.fm/archives/11059 \
[4] Franz Louis Cesista, https://leloykun.github.io/ponder/spectral-clipping/

---

> ### Author Rebuttal · Authors · 2026-03-31
>
> We thank the reviewer for the insightful review! We did our best to reply to your constructive feedback as follows:
>
> **Link of extended analysis and experiments**: https://files.catbox.moe/65g13s.pdf, https://anonymous.4open.science/r/llm-spectral-clipping-2C74/rebuttal-spectra.pdf
>
> > Q3
>
> Let $m \leq n$, where $n/m \approx 3$ for typical layers. Muon's orthogonalization computes $(m \times n)(n \times m)$ and $(m \times m)(m \times n)$ every NS iteration — two non-square matmuls with $O(m^2 n)$. SSC computes $X X^T$ once, then each NS iteration uses only $(m \times m)(m \times m)$ square GEMMs at $O(m^3)$. With $n \approx 3m$, Muon's per-iteration cost is higher. Moreover, square GEMMs also achieve better tensor core utilization, increasing the gap as matrix dimensions grow.
>
> > W1, Q1
>
> Thanks for raising this concern.
>
> - [2] (X thread) poses the question 'why not clip singular values of the update?' with hard clipping ($\sigma > 1 \to 1$), asking community help for efficient implementation. No algorithm, theory, or experiments are provided.
> - [3] (blog) derives an formulation expressing spectral clip via matrix sign compositions, which requires nested NS iterations which cannot be paralleized. Moreover, as [4] shown, nested msign is numerically unstable.
> - [4] (blog) proposes to compute spectral clip using block-matrix iterations, which requires double the matrix dimension. No optimizer design and LLM training.
>
> Our novel SSP is designed for computational and memory efficiency, requiring no matrix enlargement or nesting.
>
> These discussions explore how to compute spectral clip as a primitive operation, but none provide a concrete optimizer method or LLM training results.
>
> On the other hand, we provide
>
> - concrete formulation of SPECTRA with weight decay
> - novel theoretical justification of Frank-Wolfe framework connecting SP to weight regularization
> - new explicit geometric interpretation of hyperparameter choices
> - new noise structure analysis during LLM training
> - novel spectral clipping analysis under spike
> - systematic LLM experiments across scales
> - novel SSP and SPECTRA implementation with pytorch
>
> Our contribution is the complete method with theory, analysis, and large-scale validation—not only primitive SP, which is indeed known.
>
> > W2
>
> We first measured **per-step timing** across three scales on 4 H100 GPUs (Table 1 in the link). SPECTRA's overhead comes from SSC, which shrinks dramatically from ~17–19\% at 160M (1 accstep, small model size) to only 1.6–2.5\% at 780M/820M, as the forward/backward cost (scaling with BS, accumulation steps, seq length, \# layers, dim) dominates at larger scale.
>
> **Fixed-time budget comparison**. At 780M, we resume from the pre-decay checkpoint and add baselines 5–20\% more steps (16800–19200 vs. SPECTRA's 16k) (Table 2 in the link). Even with 20\% more steps, baselines (except ademamix) remain worse than their SPECTRA counterparts at 16k steps. Since SPECTRA's overhead is $\sim 2$\%, yet baselines need >20-30\% more steps to match, SPECTRA is at least $\sim 18-28$\% faster in wall-clock time to reach the same loss in this case.
>
> > W3
>
> Pre-clipping is not a core component—it is described as an optional safeguard throughout the paper, which bounds the spectral norm of raw gradients, analogous to standard global pre-gradient clipping that is always activated as a safeguard. On clean data (FineWeb-Edu), it provides a modest improvement (+0.005 in Figure 2). We include it because it never hurts performance and expect larger benefits in settings prone to gradient spikes (e.g., RL).
>
> > Q2
>
> We ablate over $c \in \\{1, 5, 10, 20, 50\\}$ at 160M (Table 4 in the link). Performance is flat from $c=5$ to $c=50$ (all within 0.004 of the optimum). Only $c=1$ degrades meaningfully due to over-aggressive clipping. The wide plateau suggests that the method is not sensitive to c in practice. We use $c=10$ as a robust default across all model scales, which performs well. Geometrically, $c/\lambda$ controls the radius of the spectral norm ball. The learning rate has a much larger impact on performance than $c$.
>
> > Q4
>
> The NS iteration converges superlinearly, so yes, fewer iterations may suffice depending on the conditioning of the matrix. We ablate over $ns \in \\{5, 10, 15\\}$ at 160M (Table 5 in the link): results show almost no difference in final loss, suggesting $ns=5$ is a viable choice for reducing overhead. We use $ns=10$ as a conservative default to ensure convergence across all training phases (e.g., early warmup where conditioning can be poor).
>
> > Q5
>
> SOAP and D-Muon already incorporate spectral operations internally—D-Muon orthogonalizes updates, and further clipping is unnecessary; SOAP's preconditioner serves a similar role. MARS is a coordinate-wise method without spectral operations. SPECTRA can be directly applied. Results are shown in Table 6 in the link, where we see consistent improvements.
>
> We thank you again for your strong help in improving the paper!

---

> > ### Author Rebuttal · Reviewer_sSio · 2026-04-01
> >
> > I thank the authors for the detailed response to my questions. Most of my concerns have been addressed and I will increase my score to 4. I have additional questions regarding W3.
> >
> > [W3] I understand that pre-clipping is an optional component. However, the improvement of 0.005 seems to be marginal especially when considering the overhead of SSC. Do you have any experimental results where pre-clipping introduces larger benefits (e.g., RL)?
> >
> > For others, my concerns have been fully addressed.
> >
> > [W1] Thanks for the comparison to [2-4]. I encourage authors to add this in the revised version.
> >
> > [W2] Thanks for the clarification and the additional experiments. While SPECTRA may lose its benefit in small-scale settings where the relative overhead is large, I think its scalability to larger models (due to smaller overhead) makes the method practical to use.
> >
> > [Q2-Q5] Thanks for the additional experiments. The inclusion of these results in the revised version will strengthen the paper.

---

> > > ### Author Response · Authors · 2026-04-04
> > >
> > > We thank the reviewer for the thoughtful follow-up.
> > >
> > > > [W3] Pre-clipping
> > >
> > > We conducted additional experiments to investigate pre-clipping under noisier conditions. Using SlimPajama with a small batch size of 8 (seq_len=512, 160M model, 128k iterations) to amplify gradient noise:
> > > | Method | Val Loss |
> > > |--------|----------|
> > > | Signum (lr=2e-4) | 4.955 |
> > > | Signum (lr=1e-4) | 3.688 |
> > > | SPECTRA-Signum (no pre-clip, lr=1e-3) | 3.439 |
> > > | SPECTRA-Signum + pre-clip (constant c=0.1) | 3.431 |
> > > | SPECTRA-Signum + pre-clip (cos decay, c: 10→0.01) | **3.425** |
> > >
> > > Pre-clipping improves SPECTRA by 0.014 in this noisier setting (vs. 0.005 on FineWeb-Edu with 32 batchsize), confirming its benefit. We also found that a decaying threshold schedule outperforms a constant c, because both the 'true'(large batch) gradient and noise spectral norms decrease over training (as shown in Figures F.15/F.16) — a fixed c inevitably over-clips late or under-clips in the early phase.
> > >
> > > That being said, we do not expect pre-clipping to match the impact or give the same strong improvement as post-clipping: post-clipping directly controls the update spectral norm, which stabilizes training and provides implicit weight regularization, while pre-clipping can only serve as a safeguard against gradient spikes. At a large scale, the combined overhead of pre-clip and post-clip remains under around 4%, which is moderate.
> > >
> > > We agree that investigating pre-clipping in RL and other high-noise settings is an interesting direction for future work. We will improve the discussion of the pre-clipping's safeguard role in the revised manuscript and include all additional experimental results.
> > >
> > > Thank you again for the constructive feedback.

---

### Official Review · Reviewer_4uZa · 2026-03-12

**Soundness:** 3
**Presentation:** 3
**Significance:** 2
**Originality:** 2
**Overall Recommendation:** 3
**Confidence:** 2

**Summary:**

This paper studies training instability in large language models through the spectrum of matrix-shaped optimizer updates. The proposed method, SPECTRA, augments a base optimizer with post-spectral clipping of the final update matrix and optional pre-spectral clipping of stochastic gradients, with the goal of controlling unusually large update singular values and suppressing sparse spectral spikes in the gradient. The paper also gives a theoretical interpretation of post-clipping through a composite Frank-Wolfe / spectrally regularized optimization view, and introduces Soft Spectral Clipping based on Newton-Schulz iterations to reduce the cost of exact SVD-based clipping. Empirically, the method is evaluated on 124M-820M LLM pretraining runs and is positioned in the broader optimizer landscape that includes AdEMAMix[1], SOAP[2], MARS[3], and recent Muon-style spectral optimization work[4].

References

[1] Matteo Pagliardini et al. *The AdEMAMix Optimizer: Better, Faster, Older*. ICLR 2025.

[2] Nikhil Vyas et al. *SOAP: Improving and Stabilizing Shampoo using Adam*. ICLR 2025.

[3] *MARS: Unleashing the Power of Variance Reduction for Training Large Models*. ICML 2025.

[4] Zhiqi Bu et al. *Towards Understanding of Orthogonalization in Muon*. ICML 2025.

**Compliance With Llm Reviewing Policy:**

Affirmed.

**Key Questions For Authors:**

1. The paper already compares against accepted SOTA optimizers such as SOAP[1], MARS[2], and Muon-style baselines[3], which I appreciate. Can you strengthen this with more complete matched-budget analysis, especially larger-scale SOAP results and a clearer head-to-head discussion of where SPECTRA wins or loses relative to MARS and Muon-style orthogonalization? A convincing answer here would materially increase confidence in the paper's significance and originality.
2. For a fixed wall-clock or token budget, when does SPECTRA's better step efficiency actually overcome its extra per-step overhead relative to strong accepted baselines such as AdEMAMix[4] or SOAP[1]? I would like a more explicit compute-normalized analysis.
3. How robust are the sparse spectral spike assumptions and the empirical benefits across different architectures, datasets, and scaling setups, and how should practitioners think about SPECTRA alongside modern scaling frameworks such as u-muP[5]?

References:

[1] Nikhil Vyas et al. *SOAP: Improving and Stabilizing Shampoo using Adam*. ICLR 2025.

[2] *MARS: Unleashing the Power of Variance Reduction for Training Large Models*. ICML 2025.

[3] Zhiqi Bu et al. *Towards Understanding of Orthogonalization in Muon*. ICML 2025.

[4] Matteo Pagliardini et al. *The AdEMAMix Optimizer: Better, Faster, Older*. ICLR 2025.

[5] *u-muP: The Unit-Scaled Maximal Update Parametrization*. ICLR 2025.

**Limitations:**

yes

**Strengths And Weaknesses:**

Strengths: The paper addresses an important and timely optimization problem, the proposed framework is modular enough to sit on top of multiple base optimizers, and the theory is a real contribution rather than a superficial add-on. The empirical story is broadly positive: the paper shows consistent gains in stability and validation loss across several settings, and the claim that SPECTRA can enable more aggressive learning rates is one of the most convincing parts of the submission. In a fast-moving optimizer landscape that already includes strong accepted baselines such as SOAP[1], MARS[2], and AdEMAMix[3], it is valuable that the paper aims to combine a practical training recipe with a principled spectral interpretation.

Weaknesses: First, the evaluation scope is still somewhat narrow for the strength of the paper's claims: most evidence is on decoder-only pretraining, on limited data settings, and up to 820M parameters. Second, several important practical questions remain under-analyzed, especially the separate contributions of pre-clipping versus post-clipping, sensitivity to the clipping threshold and scaling schedule, and the real wall-clock trade-off once the extra per-step cost is accounted for. Third, while the theory is interesting, some assumptions are stylized relative to modern LLM training practice, so I would like more direct diagnostics connecting the assumptions to the observed empirical gains.

Concerns: The paper does include empirical comparisons to SOAP[1], MARS[2], and Muon-style methods[4], which is a meaningful strength. However, the comparison story still feels incomplete: SOAP appears only in smaller-scale settings, while the larger-model SOAP runs are explicitly left in progress; for MARS and Muon-style baselines, I would like clearer matched-budget summaries and more direct analysis of when spectral clipping is preferable to orthogonalization or matrix-whitening style updates. I would also like a more explicit discussion of what compute regime favors SPECTRA and how the method interacts with modern scaling rules such as u-muP[6].

References:

[1] Nikhil Vyas et al. *SOAP: Improving and Stabilizing Shampoo using Adam*. ICLR 2025.

[2] *MARS: Unleashing the Power of Variance Reduction for Training Large Models*. ICML 2025.

[3] Matteo Pagliardini et al. *The AdEMAMix Optimizer: Better, Faster, Older*. ICLR 2025.

[4] Zhiqi Bu et al. *Towards Understanding of Orthogonalization in Muon*. ICML 2025.

[5] Chen Fan, Mark Schmidt, Christos Thrampoulidis. *Implicit Bias of Spectral Descent and Muon on Multiclass Separable Data*. NeurIPS 2025 Spotlight.

[6] *u-muP: The Unit-Scaled Maximal Update Parametrization*. ICLR 2025.

---

> ### Author Rebuttal · Authors · 2026-03-31
>
> We thank the reviewer for the insightful review! We did our best to reply to your constructive feedback as follows:
>
> **Link of extended analysis and experiments**: https://files.catbox.moe/65g13s.pdf, https://anonymous.4open.science/r/llm-spectral-clipping-2C74/rebuttal-spectra.pdf
>
> > ## C1, C2, Q1, Q2
>
> Thanks for these important points!
>
> We first measured **per-step timing** across three scales on 4 H100 GPUs (Table 1 in the link). SPECTRA's overhead comes from SSC, which shrinks dramatically with scale: from ~17–19\% at 160M to only ~1.6–2.5\% at 780M/820M, as the forward/backward cost (scaling with batch size, accumulation steps, seq length, \# layers, dim) dominates at larger scale.
>
> SOAP incurs much higher overhead at all scales (e.g., +21\% at 780M vs. AdamW) due to its larger memory footprint (Table F.2), which forces more accumulation steps, plus periodic eigendecomposition and per-step gradient projection. D-Muon has comparable overhead to SPECTRA; SSC is actually cheaper than orthogonalization at the same NS iterations (see our answer to Reviewer sSio Q3).
>
> **Fixed-time budget comparison**. At 780M, we resume from the pre-decay checkpoint and add baselines 5–20\% more steps (16800–19200 vs. SPECTRA's 16k) (Table 2 in the link). Even with 20\% more steps, baselines (except ademamix) remain worse than their SPECTRA counterparts at 16k steps. Since SPECTRA's overhead is $\sim 2$\%, yet baselines need >20-30\% more steps to match, SPECTRA is at least $\sim 18-28$\% faster in wall-clock time to reach target accuracy in this case.
>
> For Mars, with 20\% more steps, it (2.579) cannot match SPECTRA-Signum (2.573) at 16000 steps. Actually, Mars is coordinate-wise and also benefits from SPECTRA wrapping (Table 6 in the link).
>
> D-Muon has comparable per-step cost to SPECTRA. With 20\% more steps, D-Muon matches SPECTRA-Signum at 16000 steps. We also applied post-orthogonalization to AdamW (Table F.4), which is less effective than SSC.
>
> SOAP. Larger-scale experiments completed after submission. Despite SOAP's higher per-step cost, its final loss remains comparable or slightly worse than the best SPECTRA variant (Table 3 in the link).
>
> > ## W1, C3, Q3
>
> We provided experiments on a 1.5B-parameter muP model trained with a BS of 1M tokens, and 8,000 iterations (Table 8, Figure 1, 2 in the link), where we can observe SPECTRA variants show superior performance among the considered optimizers.
>
> We have verified that SPECTRA's benefits hold across diverse settings: 160M, 780M (deep, narrow), 820M (shallow, wide), and 1.5B with muP. We also conducted noise structure analysis on a MuP-GPT model (Figure 3, 4), confirming spectral spikes in stochastic gradients, though less frequent and with smaller magnitude.
>
> We use FineWeb-Edu throughout, which is a well-curated, standard pretraining dataset. We believe this is the right evaluation setting: if SP helps on clean data, it validates the method's core mechanism (controlling the spectral norm of updates) rather than relying on data noise to amplify the effect. In fact, if spectral spikes are more pronounced with noisier gradients, we expect SPECTRA's benefit to be even larger on less curated datasets.
>
> > ## W2
>
> We ablate over $c \in \\{1, 5, 10, 20, 50 \\}$ at 160M (Table 4 in the link). Performance is flat from $c=5$ to $c=50$ (all within 0.004 of the optimum). Only $c=1$ degrades meaningfully due to over-aggressive clipping. The wide plateau suggests that the method is not sensitive to c in practice. We use $c=10$ as a robust default across all model scales, which performs well. The learning rate has a much larger impact on performance than $c$.
>
> Pre-spectral clipping serves a different role: it bounds the spectral norm of raw gradients, analogous to standard pre-global gradient clipping. In Figure 2, pre-clipping provides a modest improvement (+0.005) with a clean dataset. We expect larger benefits on noisier data or in settings prone to gradient spikes (e.g., reinforcement learning). Since it never hurts performance and adds the same overhead as post-clipping, we include it as an optional safeguard. We did not activate it in most experiments to isolate the contribution of post-clipping.
>
> > ## W3
>
> While Theorem 3.4 assumes convexity and Lipschitz gradients, the FW formulation itself is assumption-free: SP + weight decay structurally corresponds to composite FW under a spectral norm ball constraint, characterizing how $c$, $\eta$, and $\lambda$ control weight norms. This applies directly to LLM training, as verified empirically (e.g., $\ell_\infty$ norms are smaller under SPECTRA with same lr). The empirical gains connect to the theory through: (1) SSC bounds the spectral norm of updates, stabilizing training, (2) the implicit weight regularization potentially improves generalization, and (3) anisotropic spectral spikes as components in gradient noise are addressed by Lemma 4.2, independent of convexity.
>
> We thank you again for your strong help in improving the paper!

---

### Official Review · Reviewer_CpTK · 2026-03-12

**Soundness:** 3
**Presentation:** 3
**Significance:** 3
**Originality:** 3
**Overall Recommendation:** 5
**Confidence:** 2

**Summary:**

This paper investigates the spectral structure of gradients and update matrices in large language model (LLM) training and proposes a Spectral Clipping framework to stabilize training and improve generalization. The authors identify two empirical issues: (i) optimizer updates often exhibit large spectral norms, which can destabilize training and harm generalization; (ii) stochastic gradient noise displays sparse spectral spikes, where a few singular values dominate and are significantly larger than the rest. To address these, they introduce the SPECTRA framework, which incorporates post-spectral clipping to constrain the spectral norm of updates and pre-spectral clipping to eliminate spectral spikes in gradients. Theoretically, they demonstrate that spectral clipping is equivalent to the Composite Frank-Wolfe algorithm, corresponding to spectral-norm-constrained optimization with implicit regularization. Empirically, applying spectral clipping across various optimizers leads to reductions in validation loss, confirming its effectiveness.

**Compliance With Llm Reviewing Policy:**

Affirmed.

**Final Justification:**

The authors have addressed my concerns well, so I have decided to maintain my positive score.

**Key Questions For Authors:**

1. The paper treats spectral spikes as noise rather than signal. Could it be possible that these spikes arise from limited model capacity, causing them to align orthogonally to the true gradient signal? If so, would clipping such spikes inadvertently discard important directional information and potentially impair training performance?
2. Regarding scalability, do the authors expect the proposed spectral clipping technique to remain effective when applied to larger-scale language models? Further discussion or preliminary evidence on this point would be valuable.

**Limitations:**

The authors have appropriately acknowledged the primary limitations of their work in the discussion for future research. However, the paper would be further strengthened by addressing the questions raised in the "Key Questions For Authors" section.

**Strengths And Weaknesses:**

Strength:
1. This paper introduces a simple post-spectral clipping operation, denoted as $U_k\rightarrow\text{clip}_{c_k}^{sp}(U_k)$, which is integrated into each iteration to effectively control the spectral norm of the updates. This straightforward mechanism enhances training stability and generalization.
2. Through empirical analysis, the authors identify that the stochastic gradient spectrum is heavy-tailed and that the gradient noise contains low-rank spikes. They demonstrate that conventional global clipping tends to distort the signal while clipping noise, whereas the proposed spectral clipping removes noise spikes while preserving the signal subspace, thereby offering a significant advantage.
3. From an optimization perspective, the authors prove that the spectral clipping update is equivalent to the Composite Frank–Wolfe algorithm, which corresponds to solving a spectral-norm-constrained optimization problem with Frobenius norm regularization. They also provide a convergence bound for the proposed method, further substantiating its theoretical soundness.

Weakness:
1. The theoretical analysis relies on the assumptions that the loss function is convex and the gradients are Lipschitz continuous. While these assumptions may not hold in LLM training, they are common in optimization literature and do not severely undermine the contribution. However, it would strengthen the paper to discuss how the theoretical insights might extend to non-convex settings.
2. The paper distinguishes spectral normalization (a model architecture regularization) from spectral clipping (an optimizer-level stability mechanism). However, it lacks theoretical justification for why spectral clipping is superior to spectral normalization. A theoretical comparison or intuitive explanation of the advantages would be beneficial. Nevertheless, this is not a fatal flaw.
3. The proposed soft spectral clipping algorithm employs the Newton–Schulz iteration for efficient computation, which is also used in Muon. Despite the computational similarity, the paper does not provide a theoretical argument for why soft spectral clipping outperforms Muon's orthogonalization approach.

---

> ### Author Rebuttal · Authors · 2026-03-31
>
> We thank the reviewer for the insightful review!  We did our best to reply to your constructive feedback as follows:
>
> **Link of extended analysis and experiments**: https://files.catbox.moe/65g13s.pdf, https://anonymous.4open.science/r/llm-spectral-clipping-2C74/rebuttal-spectra.pdf
>
> > ## W1
>
> Thanks for the nice suggestion. The convexity assumption can be removed in Theorem 3.4: instead of showing convergence of function residual, we can show convergence of the Frank-Wolfe gap to a stationary point, which is the standard measure in constrained non-convex optimization [1].
>
> The most interesting theoretical insights that guide practice do not depend on convexity: (1) the geometric characterization of how $c$, $\lambda$, $\eta$ determine the size of spectral norm ball and implicit weight regularization (Section 5.2), and (2) Lemma 4.2 showing that spectral clipping mitigates spectral spikes — both hold in general. As an example, we use this lemma to prove convergence of SGD under non-convexity (Theorem 4.6), demonstrating that it serves as a useful building block for future non-convex analysis under more realistic noise assumptions. We thank the reviewer for this suggestion and will include this discussion in the revision.
>
> [1] Convergence Rate of Frank-Wolfe for Non-Convex Objectives, arxiv 2016.
>
> > ## W2 & W3
>
> We believe the reviewer refers to the distinction between spectral clipping and normalization, both applied at the optimizer level. We discussed this before Section 3.1 (line 207): Muon's orthogonalization is a special case of SPECTRA where $c \to 0$ (equivalently $\alpha \to \infty$), which normalizes all singular values to 1. The main theoretical difference: orthogonalization solves the constrained problem without regularization ($b=0$), whereas SPECTRA with finite $c$ corresponds to a composite Frank-Wolfe step with implicit weight regularization, potentially improving generalization. Intuitively, normalization discards the relative magnitude structure of the update. Clipping preserves singular values below $c$ and only attenuates those above, retaining fine-grained directional information. We also compare empirically post-orthogonalization vs. SSC at 160M model. (Table F.4) where the former is less effective.
>
> > ## Q1
>
> Great question. Yes, it is possible that spectral spikes carry signal rather than noise. Our noise structure analysis (Figures in Section F.2) shows that the largest noise singular values can be comparable to the true gradient signal, while the true gradient itself has large condition number. In this regime, aggressive clipping would indeed discard important directional information.
> Our ablation on $c$ (Table 4 in the link) confirms this: $c=1$ (aggressive) hurts performance, while $c \in [5, 50]$ performs equally well — the threshold is set large enough to clip only extreme spikes without interfering with informative spectral components. For pre-clipping, we treat it as an optional safeguard with a conservatively large threshold for the same reason. We will make this point clear in the manuscript.
>
> > ## Q2
>
> Indeed, evidence of effectiveness to larger scale is important. In the link above, we provided experiments on a 1.5B-parameter muP model (embed dim: 1536, nb of heads: 24, nb of layers: 48) trained with a batch size of 1M tokens, and 8,000 iterations.  where we can observe SPECTRA variants show superior performance among the considered optimizers.
>
> We thank you again for your strong help in improving the paper, and we appreciate your support.

---

> > ### Author Rebuttal · Reviewer_CpTK · 2026-04-02
> >
> > Thank authors for their thorough response, which has satisfactorily addressed all of my concerns.

---

### Official Review · Reviewer_m2z4 · 2026-03-13

**Soundness:** 4
**Presentation:** 3
**Significance:** 2
**Originality:** 3
**Overall Recommendation:** 5
**Confidence:** 4

**Summary:**

This paper proposes a spectral clipping method (SPECTRA) by (i) post-spectral clipping of updates to enforce spectral-norm constraints (ii) optional pre-spectral clipping of gradients to suppress spectral noise spikes. This method mitigates sparse spectral spikes observed in existing  optimizers. Theoretical results show an equivalence relationship between the proposed method and a Composite Frank-Wolfe algorithm. Extensive experiments show that SPECTRA improves validation loss for various optimizers.

**Compliance With Llm Reviewing Policy:**

Affirmed.

**Final Justification:**

I appreciate the authors' efforts for addressing my quesitons and concerns. I maintain my assessment as accept.

**Key Questions For Authors:**

See above.

**Limitations:**

yes

**Strengths And Weaknesses:**

Strengths:

1. The motivation for the algorithm is well established.

2. The authors provide a proof that applying post-spectral clipping to the standard update rule with decoupled weight decay is equivalent to a Composite Frank-Wolfe algorithm, which implies SPECTRA minimizes a spectral-norm constrained  objective with implicit weight regularization.

3. Empirical verification. This paper provides extensive experimental verification for both the observed phenomena and proposed algorithm.


Weaknesses:

1. In section 5.1, it is mentioned the computational overhead is moderate—typically 2-10 percent of the total runtime. How is this overhead compared with performance improvement? The cost-benefit tradeoff is not sufficiently clarified.

2. Results on larger models would make the empirical findings more convincing.

3. The clipping threshold is set as $c=10$ or $5$, but it lacks a systematic sensitivity analysis of this hyperparameter. In particular, there is no dedicated ablation showing how performance and stability vary across a wider range of $c$ values, nor a clear study of how the best $c$ depends on model scale, optimizer choice, or training schedule. As a result, it is difficult to assess how robust the method really is to the choice of $c$.

4. It could be helpful to include additional discussion of related preconditioning methods from standard optimization theory that also modify the update in the spectral domain, in order to better clarify the conceptual relationship between spectral clipping and classical preconditioning.

5. While there is a connection between the proposed SPECTRA and Muon, it is still not clear why truncating singular values performs better than fully normalizing them to one.

---

> ### Author Rebuttal · Authors · 2026-03-31
>
> We thank the reviewer for the insightful review! We did our best to reply to your constructive feedback as follows:
>
> **Link of extended analysis and experiments**: https://files.catbox.moe/65g13s.pdf, https://anonymous.4open.science/r/llm-spectral-clipping-2C74/rebuttal-spectra.pdf
>
> > ## W1
>
> Thanks for raising this important question.
>
> We first measured **per-step timing** across three scales on 4 H100 GPUs (Table 1 in the link). SPECTRA's overhead comes entirely from SSC, which shrinks dramatically with scale: from ~17–19\% at 160M to only ~1.6–2.5\% at 780M/820M, as the forward/backward cost (scaling with batch size, accumulation steps, sequence length, \# layers, dim) dominates at larger scale.
>
> SOAP incurs much higher overhead at all scales (e.g., +21\% at 780M vs. AdamW) due to its larger memory footprint (Table F.2), which forces more accumulation steps, plus periodic eigendecomposition and per-step gradient projection. D-Muon has comparable overhead to SPECTRA; SSC is actually cheaper than orthogonalization at the same NS iterations (see our answer to Reviewer sSio Q3).
>
> **Fixed-time budget comparison**. At 780M, we resume from the pre-decay checkpoint and add baselines 5–20\% more steps (16800–19200 vs. SPECTRA's 16k) (Table 2 in the link). Even with 20\% more steps, baselines (except ademamix) remain worse than their SPECTRA counterparts at 16k steps.
> Since SPECTRA's overhead is $\sim$2\%, yet baselines need 20-30\% more steps to match, SPECTRA is at least $\sim 18-28$\% faster in wall-clock time to reach the same loss in this case. SPECTRA is not only sample-efficient but also compute-efficient.
>
> For Mars, with 20\% more steps, it (2.579) cannot match SPECTRA-Signum (2.573) at 16000 steps. Actually Mars is coordinate-wise and also benefits from SPECTRA wrapping (Table 6 in the link).
>
> D-Muon has comparable per-step cost to SPECTRA. With 20\% more steps, D-Muon matches SPECTRA-Signum at 16000 steps. We also applied post-orthogonalization to AdamW (Table F.4), which is less effective than SSC.
>
> SOAP. Larger-scale experiments completed after submission. Despite SOAP's higher per-step cost, its final loss remains comparable or slightly worse than the best SPECTRA variant (Table 3 in the link).
>
> > ## W2
>
> In the link above, we provided experiments on a 1.5B-parameter muP model (embed dim: 1536, nb of heads: 24, nb of layers: 48) trained with a batch size of 1M tokens, and 8,000 iterations.  where we can observe SPECTRA variants show superior performance among the considered optimizers.
>
> > ## W3
>
> Thanks for the question.
> We ablate over $c \in \\{1, 5, 10, 20, 50 \\}$ at 160M with WSD schedule (Table 4 in the link). Performance is flat from $c=5$ to $c=50$ (all within 0.004 of the optimum). Only $c=1$ degrades meaningfully due to over-aggressive clipping.
> The wide plateau  suggests that the method is not sensitive to c in practice.
> We use $c=10$ as a robust default across all model scales which performs well. Geometrically, $c/\lambda$ controls the radius of the spectral norm ball. The learning rate on the other hand has a much larger impact on performance than $c$.
>
> > ## W4
>
> Thanks for the nice suggestion. The most related preconditioning method is Shampoo, which applies two-sided preconditioning using accumulated gradient statistics:
> $$X_{k+1} = (1-\lambda \eta_k)X_k - \alpha \eta_k L_k^{-1/4} M_k R_k^{-1/4}, \quad L_k = L_{k-1} + G_k G_k^T,  R_k = R_{k-1} + G_k^T G_k.$$
> Muon simplifies this by replacing accumulated statistics with the current momentum:
> $$X_{k+1}
> = (1-\lambda \eta_k)X_k - \alpha \eta_k (M_k M_k^T)^{-1/4} M_k (M_k^T M_k)^{-1/4}
> = (1-\lambda \eta_k)X_k - \alpha \eta_k (M_k M_k^T)^{-1/2} M_k.$$
> SPECTRA can indeed be viewed as a cheap one-sided preconditioning:
> $$X_{k+1} = (1-\lambda \eta_k)X_k - \alpha \eta_k (I + U_kU_k^T/c_k^2)^{-1/2} U_k.$$
> However, the conceptual role differs: Shampoo's preconditioning is closer to a matrix version of AdaGrad (adaptive stepsize), while Muon and SPECTRA are more naturally interpreted as Frank-Wolfe methods with spectral norm constraints. We will add this discussion to the related work.
>
> > ## W5
>
> Great question. Muon's orthogonalization is a special case of SPECTRA where $c \to 0$ (equivalently $\alpha \to \infty$), which normalizes all singular values to 1. The main theoretical difference: orthogonalization solves the constrained problem without regularization ($b=0$), whereas SPECTRA with finite $c$ corresponds to a composite Frank-Wolfe step with implicit weight regularization, potentially improving generalization. Intuitively, normalization discards the relative magnitude structure of the update. Clipping preserves singular values below $c$ and only attenuates those above, retaining fine-grained directional information. We also compare empirically post-orthogonalization vs. SSC at 160M model. (Table F.4) where the former is less effective.
>
> We thank you again for your strong help in improving the paper and appreciate your support.

---

> > ### Author Rebuttal · Reviewer_m2z4 · 2026-04-03
> >
> > Thank the authors for the detailed response. My questions have been addressed.

---

### Decision · Program_Chairs · 2026-04-30

**Decision:**

Accept (regular)

**Comment:**

This paper proposes **SPECTRA**, a simple singular value clipping framework for LLM pretraining that can be combined with optimizers. It is clearly motivated, and has broadly consistent empirical gains in validation and training stability.

The main concerns are the scope of the evaluation, additional overhead, how helpful the pre-clipping is, and the gap between theory and modern LLM training practice. During the authors’ response stage, the authors addressed these concerns sufficiently through additional experiments, further analysis, and clarifications. I therefore recommend acceptance.

For the final version, the paper would benefit from incorporating the added large-scale and compute-normalized results more directly, and from clarifying that how helpful the pre-clipping is.